# SAMPLE-EFFICIENT ACTOR-CRITIC ALGORITHMS WITH AN ETIQUETTE FOR ZERO-SUM MARKOV GAMES

## ABSTRACT

We introduce algorithms based on natural actor-critic and analyze their sample complexity for solving two player zero-sum Markov games in the tabular case. Our results improve the best-known sample complexities of policy gradient/actor-critic methods for convergence to Nash equilibrium in the multi-agent setting. We use the error propagation scheme in approximate dynamic programming, recent advances for global convergence of policy gradient methods, temporal difference learning, and techniques from stochastic primal-dual optimization. Our algorithms feature two stages, requiring agents to agree on an etiquette before starting their interactions, which is feasible for instance in self-play. However, the agents only access to joint reward and joint next state and not to each other's actions or policies. Our complexity results match the best-known results for global convergence of policy gradient algorithms for single agent RL. We provide numerical verification of our methods for a two player bandit environment and a two player game, Alesia. We observe improved empirical performance as compared to the recently proposed optimistic gradient descent-ascent variant for Markov games.

## 1 INTRODUCTION

We study two-player zero-sum Markov game framework which is a key model with broad applications in competitive reinforcement learning (RL), robust RL, and many others Zhang et al. (2019a; 2021). This framework is introduced by Shapley (1953) as *stochastic games* and popularized in RL with Littman (1994). In its basic form, two agents with competing interests interact in an environment where the reward and the state transition depend on the actions of both players. Even with this simplicity, such systems achieved impressive success in game-playing and robotics (Kober et al., 2013; Silver et al., 2017; Mnih et al., 2015; Vinyals et al., 2019; Brown & Sandholm, 2019).

While value-based methods offer near-optimal guarantees (Sidford et al., 2020; Bai et al., 2020; Bai & Jin, 2020; Xie et al., 2020; Tian et al., 2020), policy gradient (PG) methods, including actor-critic (AC) and their *natural* counterparts natural PG (NPG) (Kakade, 2001) and natural AC (NAC) (Peters & Schaal, 2008), only have limited guarantees, despite their model-free and easy-to-implement structure, flexibility and generality (Schulman et al., 2015; 2017; Wang et al., 2016).

The PG methods (Kakade, 2001; Sutton et al., 2000) directly optimize the value function in the policy space— a non-convex optimization problem even in the basic single agent, tabular setting. Intriguingly, recent results demonstrate globally optimal convergence of PG methods by identifying a hidden convexity structure for single agent RL (Agarwal et al., 2020; Cen et al., 2020; Mei et al., 2020; Bhandari & Russo, 2019; 2021; Xu et al., 2020b; Lan, 2021; Khodadadian et al., 2021b; Hong et al., 2020; Xu et al., 2020a; Khodadadian et al., 2021b), and multi-agent RL (MARL) (Daskalakis et al., 2020; Wei et al., 2021; Zhao et al., 2021).

The existing results on PG methods for tabular two-player zero-sum Markov games mostly focus on decentralized algorithms with sample complexities $\tilde{\mathcal{O}}(\epsilon^{-12.5})$ (Daskalakis et al., 2020), $\tilde{\mathcal{O}}(\epsilon^{-8})$, and even $\tilde{\mathcal{O}}(\epsilon^{-4})$, yet with some limitations (Wei et al., 2021); see Section 1.1 for the details. With function approximation, Zhao et al. (2021) obtains $\tilde{\mathcal{O}}(\epsilon^{-6})$ sample complexity when given access to unbiased sampling oracles of the value functions.

On the other hand, the best-known sample complexity for global optimality for single agent problem is $\tilde{\mathcal{O}}(\epsilon^{-2})$ in the tabular case (Lan, 2021). As this complexity is achieved by value-based/model-based methods in the multi-agent setting (Sidford et al., 2020; Zhang et al., 2020), one expects a similar complexity for policy-based methods. Our work precisely bridges this gap and develops policy gradient methods whose performance for MARL is closer to their single agent counterparts.

**Contributions.** We propose algorithms based on natural actor-critic (NAC) framework for solving two-player zero-sum Markov games in the tabular case. Our sample complexity results match the best-known ones for global optimality in the single agent setting (Lan, 2021; Khodadadian et al., 2021b; Hong et al., 2020; Xu et al., 2020b). In particular, we show that by using inner loops for policy evaluation and a carefully designed algorithm, the sample complexity to get an $\epsilon$-approximate Nash equilibrium, is $\tilde{\mathcal{O}}(\epsilon^{-2})$, by assuming a uniform lower bound on the policies. Without this assumption, we show $\tilde{\mathcal{O}}(\epsilon^{-4})$ complexity.[1]

Surprisingly, we achieve these results—to our knowledge, for the first time with policy gradient methods—mostly by a careful adaptation of the recent results for policy gradient methods in single agent setting, temporal difference learning, two-stage error propagation framework of policy iteration (Perolat et al., 2015), and by employing techniques from stochastic primal-dual optimization.

These developments require a careful algorithm design and analysis. In particular, two-stage nature of the algorithm incurs biases between the stages that we have to control carefully. Obtaining $\tilde{\mathcal{O}}(\epsilon^{-2})$ complexity requires a tighter analysis for both stages of the algorithm, with strict control on the aforementioned bias. Therefore, it requires more advanced techniques and algorithms, inspired from the stochastic primal-dual optimization literature. We explicitly highlight our important new techniques as *insights* in the sequel. The full proofs are included in the appendices.

## 1.1 RELATED WORKS

**Policy gradient methods.** There is growing interest in global convergence of PG methods in the single agent setting. Several works showed convergence rates of natural policy gradient (NPG) in the tabular setting by assuming access to exact value function oracle (Agarwal et al., 2020; Cen et al., 2020; Mei et al., 2020; Bhandari & Russo, 2019; 2021) or when value functions are estimated from the data (Shani et al., 2020; Xu et al., 2020b; Lan, 2021; Khodadadian et al., 2021b; Hong et al., 2020; Xu et al., 2020a; Khodadadian et al., 2021b). To our knowledge, the best sample complexity for NPG methods with inner loop for policy evaluation (which we refer to as *NAC*) is $\tilde{\mathcal{O}}(\epsilon^{-2})$ and is due to (Lan, 2021). For single loop NAC with online policy evaluation, the best sample complexity is $\tilde{\mathcal{O}}(\epsilon^{-4})$ as obtained in (Khodadadian et al., 2021b; Hong et al., 2020; Xu et al., 2020b) (see (Khodadadian et al., 2021a, Table 1)). For a general overview of results in MARL we refer to Zhang et al. (2019a).

**Policy gradient methods for two-player zero-sum Markov games.** With the positive results on global convergence of PG methods, translating these results to the competitive MARL has been the goal of many recent works. In particular, independent policy gradient methods with the agents interacting symmetrically has been considered in Daskalakis et al. (2020); Wei et al. (2021). The work of Daskalakis et al. (2020) built on Agarwal et al. (2020) by using REINFORCE estimator (Williams, 1992) and obtained sample complexity of $\tilde{\mathcal{O}}(\epsilon^{-12.5})$ for reaching to one-sided Nash equilibrium.

The algorithm of Wei et al. (2021) built on optimistic gradient descent-ascent (OGDA) method combined with a running estimate of the value function, obtaining $\tilde{\mathcal{O}}(\epsilon^{-8})$ sample complexity for finding a policy pair with small duality gap. In addition, Wei et al. (2021) showed improved complexity $\tilde{\mathcal{O}}(\epsilon^{-4})$ when restricted to Euclidean projections onto the simplex with metric subregularity assumption. There are two subtleties about this result: First, as pointed out in Daskalakis et al. (2020), metric subregularity constant can be arbitrarily small, resulting in degradation of the rate. Second, as also pointed out by Wei et al. (2021), this result is limited to Euclidean setting and cannot be extended to the NPG with softmax policy update, which requires projection with KL divergence. The algorithm can be seen similar to the gradient ascent algorithm in Agarwal et al. (2020). As

---

[1]In Appendix E, we design an algorithm based on *single loop* NAC with the complexity of $\tilde{\mathcal{O}}(\epsilon^{-4})$ (and $\mathcal{O}(\epsilon^{-7})$ without assuming lower bounded policies). Our results on this algorithm is, to our knowledge, the first finite-sample analysis of single loop NAC-based methods for two-player zero-sum Markov games.

| | Assumption | Complexity |
|---|---|---|
| Daskalakis et al. (2020) | $\zeta_{s,a,b} \geq \underline{\zeta} > 0.^*$ | $\tilde{\mathcal{O}}(\epsilon^{-12.5})$ |
| Wei et al. (2021) | $\max_{x,y,s,s'} T_{x,y}^{s \to s'} = \frac{1}{\mu}$ where $\mu > 0.^\dagger$ | $\tilde{\mathcal{O}}(\epsilon^{-8}), \tilde{\mathcal{O}}(\epsilon^{-4})^\ddagger$ |
| Section 3.1 | Assumption 1 | $\tilde{\mathcal{O}}(\epsilon^{-4})$ |
| Section 3.1 | Assumption 1, 2 | $\tilde{\mathcal{O}}(\epsilon^{-2})$ |

Table 1: $^*$In a game with finite steps, $\zeta_{s,a,b}$ is the probability that the game will end at state $s$, after taking actions $a, b$ (Daskalakis et al., 2020, Section 2). $^\dagger T_{x,y}^{s \to s'}$ is the time that it takes to go from state $s$ to state $s'$ by using policy pair $x, y$ (Wei et al., 2021, Assumption 1). $^\ddagger$This $\tilde{\mathcal{O}}(\epsilon^{-4})$ complexity by Wei et al. (2021) requires using Euclidean projections onto the simplex instead of softmax updates and depends on the metric subregularity constant. Hence it is not applicable to NPG.

shown in Agarwal et al. (2020) for single agent problems, NPG methods have much better convergence properties than Euclidean projected gradient ascent methods. For comparison with the works in Daskalakis et al. (2020); Wei et al. (2021), we also refer to Remark 2.1 and Table 1.

Another very related work to ours is by Zhao et al. (2021) which considered *(i)* tabular setting with exact value functions and *(ii)* online setting with function approximation, also using the error propagation scheme of Perolat et al. (2015). Building on Agarwal et al. (2020), this work showed $\tilde{\mathcal{O}}(\epsilon^{-6})$ sample complexity with function approximation, with access to unbiased samples of the value functions. In contrast, we focus on the tabular setting and we do not assume access to unbiased value function oracles. Indeed, lack of unbiased samples for value functions required us to use new *insights* described in the sequel, to derive the tighter complexities $\tilde{\mathcal{O}}(\epsilon^{-2})$ and $\tilde{\mathcal{O}}(\epsilon^{-4})$.

**Policy gradient methods for linear quadratic regulator (LQR).** For zero-sum LQR, Zhang et al. (2019b); Bu et al. (2019) showed global convergence of PG with exact value function oracles. These methods have a nested structure where one player computes best-response and the other does policy gradient updates. Recently, Zhang et al. (2021) built on Zhang et al. (2019b) to derive sample complexities when value functions are estimated from data.

## 2 PRELIMINARIES

**Notation.** We consider the tabular setting with finite state and action spaces denoted by $S$, $A$, $B$ and the discount factor $\gamma < 1$. The policy of the min agent is $x$ and the max agent is $y$, with action sets $A, B$, respectively. At state $s$, both agents take actions independent of each other: $a \sim x(\cdot|s)$ and $b \sim y(\cdot|s)$. Based on the actions, the environment transitions to the next state $s' \sim P(\cdot|s, a, b)$ and the agents receive reward $|r(s, a, b)| \leq 1$. Given a policy pair $x, y$, we denote the induced steady-state distribution as $\rho^{x,y}$. Let $\mathcal{U}$ denote the uniform distribution for states that we also take as the initial state distribution for simplicity. We denote the probability simplex as $\Delta$. Given a policy $x$, we sometimes use the notation $x^s$ for $x(\cdot|s)$ in the proofs. We use $e(s_t) \in \mathbb{R}^{|\mathcal{S}|}$ to denote the vector such that $e(s) = 1$ if $s = s_t$ and $e(s) = 0$, if $s \neq s_t$. We use the same notation for $e(s_t, a_t)$. The value function for state $s$ is defined as

$$V^{x,y}(s) = \mathbb{E}_{x,y}\left[\sum_{t=0}^{\infty} \gamma^t r(s_t, a_t, b_t) | s_0 = s\right],$$

where $\mathbb{E}_{x,y}$ is over random variables $s_t, a_t, b_t$ for all $t \geq 0$ as $a_t \sim x(\cdot|s_t)$, $b_t \sim y(\cdot|s_t)$ and $s_{t+1} \sim P(\cdot|s_t, a_t, b_t)$. Similarly, the action value function is defined as $Q^{x,y}(s, a, b) = \mathbb{E}_{x,y}\left[\sum_{t=0}^{\infty} \gamma^t r(s_t, a_t, b_t) | s_0 = s, a_0 = a, b_0 = b\right]$. With these definitions, we can state the formal problem. For all $s \in S$, we aim to solve

$$\min_{x(\cdot|s) \in \Delta} \max_{y(\cdot|s) \in \Delta} V^{x,y}(s).$$

We denote the information needed in algorithms as *oracles*. We provide the background on NPG, NAC, TD(0) in Appendix A.

**Nash equilibrium.** We assume the existence of a pair of policies $x^\star, y^\star$ that are Nash equilibrium, namely, for all $s$, $V^{x^\star,y}(s) \leq V^\star(s) := V^{x^\star,y^\star}(s) \leq V^{x,y^\star}(s)$. We are interested in finding a one-sided Nash equilibrium, similar to Daskalakis et al. (2020); Zhao et al. (2021); Zhang et al. (2019b);

Bu et al. (2019). As mentioned in Daskalakis et al. (2020), for the other player, one can re-run the algorithm by switching roles to have the guarantee for both players. In particular, for the initial state distribution $\mathcal{U}$, we seek for $x_{out}$ such that

$$\mathbb{E}_{s_0 \sim \mathcal{U}}[\max_y V^{x_{out}, y}(s_0) - V^\star(s_0)] \leq \epsilon.$$

It is easy to prove that this quantity on the LHS is $0$ if and only if $x_{out}$ is a Nash equilibrium.

**Interaction procedure.** We use the interactions of the agents with the environment to estimate the value functions and related oracles for the running of the algorithm. At each interaction, agents have access to $(s_i, a_i, r(s_i, a_i, b_i), s_{i+1})$ and $(s_i, b_i, r(s_i, a_i, b_i), s_{i+1})$, respectively. In terms of access of agents, our oracle model is similar to Daskalakis et al. (2020); Wei et al. (2021). However, one difference is that we require a *game etiquette*: Our algorithms have two stages where the agents have to behave differently. As long as this etiquette is respected by the agents (for example embedded to players in the beginning of the game), they do not need further communication.

**Softmax update rule/NAC.** Given Kullback-Leibler divergence KL and action-value function $Q^{x_t}$,

$$x_{t+1}(\cdot|s) = \mathcal{P}^{\mathrm{KL}}(x_t(\cdot|s), Q^{x_t}(s, \cdot)) := \arg \min_{x(\cdot|s) \in \Delta} \langle Q^{x_t}(s, \cdot), x(\cdot|s) \rangle + \mathrm{KL}(x(\cdot|s), x_t(\cdot|s)), \quad (1)$$

is known as NPG with softmax parameterization (Agarwal et al., 2020, Lemma 5.1). When there is a *critic* estimating $Q^{x_t}$, along with *actor* updating $x_t$ with NPG, this algorithmic framework is called natural actor-critic, in short, NAC. We focus on KL divergence for simplicity and its wide use. Our developments also hold for more general *Bregman divergences* as Zhan et al. (2021).

**Assumption 1.** *There exists $\rho$ such that, for any policy iterate pair $x_t, y_t$, for any state $s$, it holds that $\rho^{x_t, y_t}(s) \geq \underline{\rho} > 0$, where $\rho^{x_t, y_t}$ is the stationary state distribution induced by the policy pair.*

**Assumption 2.** *There exist $\underline{x}, \underline{y}$ such that, for any policy iterate pair $x_t, y_t$, for any state action tuple $s, a, b$, it holds that $x_t(a|s) \geq \underline{x} > 0, y_t(b|s) \geq \underline{y} > 0$.*

**Our rationale on the assumptions.** Assumption 1 and 2 essentially mean positive definiteness of the sampling matrices in policy evaluation (see eqs. (30), (34) and (42)). To our knowledge, some form of this assumption is required in most of the existing work on temporal difference (TD) (including TD(0)) methods for policy evaluation (Bhandari et al., 2018; Xu et al., 2020b; Khodadadian et al., 2021a; Lan, 2021; Hong et al., 2020; Xu et al., 2020a; Wu et al., 2020; Zou et al., 2019) (see App. A). The complexity $\tilde{\mathcal{O}}(\epsilon^{-2})$ requires Assumption 2 even for single agent problems (see (Lan, 2021, Rem. 1, Sec. 5.2)).

**Remark 2.1.** *As summarized in Table 1, similar assumptions to Assumption 1 are used in Daskalakis et al. (2020); Wei et al. (2021). In particular, each of these assumptions ensure that all action-state pairs are observed with nonzero probability throughout the game. Moreover, by additionally requiring Assumption 2, we can obtain the complexity $\tilde{\mathcal{O}}(\epsilon^{-2})$, matching the single-agent counterpart.*

**Markovian bias.** For simplicity, we assume that we sample from the steady state distribution of a given policy pair. During normal interaction with the environment, this is not the case and we obtain a single stream of data. Hence, TD(0) update is biased—commonly referred to as the Markovian bias. A large body of literature in the single agent literature showed that the effect of this bias in TD(0) update is essentially additive and can be handled by assuming uniform mixing of the induced Markov chain (Wu et al., 2020; Bhandari et al., 2018; Zou et al., 2019; Khodadadian et al., 2021b; Xu et al., 2020b;a). These analyses apply to our policy evaluation routines, extending them to the Markovian setting. For simplicity, we show our techniques with i.i.d. assumption and then illustrate how the extension with Markovian data follows with the uniform mixing assumption in Appendix D.

**Error propagation for approximate dynamic programming.** Perolat et al. (2015) proposed error propagation analysis for approximate version of generalized policy iteration for zero-sum Markov games (see Appendix C). The authors showed that the following two-stage algorithm converges:

• *Stage 1:* Given a fixed value function $V_{k-1}$, find the policy pair which is an $\epsilon$-equilibrium.

$$\min_{x^s \in \Delta} \max_{y^s \in \Delta} \sum_{a,b} x(a|s) y(a|s) Q_{k-1}(s, a, b) =: x^s Q_{k-1}^s y^s, \quad (2)$$

---

**Algorithm 1** Reflected NAC with a game etiquette and $\zeta$-greedy exploration

---

**Require:** $\mathcal{P}^{\mathrm{KL}}$ defined in (1) in Sec. 2. Exploration parameter $\zeta \geq 0$ (with equality if Assumption 2 holds). Subroutines `Policy-Eval-V`, `Policy-Eval-`$\theta$, `Policy-Eval-`$\nu$ (see Alg. 2, 3, 4 and note $\beta_n^\nu$, $\beta_n^\theta$, $\beta_n^\omega$ are potential step sizes for this routine). Initial policies $x_0, y_0, \bar{y}_0$. $x^k, y^k$ denote outer, $x_t, y_t$ denote inner loop's iterate.

1: **for** $k = 0, 1, \ldots$ **do**
2:     **Stage 1** // Approximately solve a matrix game
3:     **for** $t = 0, 1, \ldots, T - 1$ **do**
4:         $[\hat{V}_{k-1}^x, \hat{V}_{k-1}^y] = [\texttt{Policy-Eval-}V(x^{k-1}, y^{k-1}, N, \beta_n^\omega), \texttt{Policy-Eval-}V(x^{k-1}, y^{k-1}, N, \beta_n^\omega)]$
             // Both players compute their own estimations of $V_{k-1}$, denoted as $\hat{V}^x$ and $\hat{V}^y$
5:         $[\hat{\theta}_{t+1}^x, \hat{\theta}_{t+1}^y] = [\texttt{Policy-Eval-}\theta(x_t, y_t, N, \hat{V}_{k-1}^x, \beta_n^\theta), \texttt{Policy-Eval-}\theta(x_t, y_t, N, \hat{V}_{k-1}^y, \beta_n^\theta)]$

6:         $x_{t+1}(\cdot|s) = \mathcal{P}^{\mathrm{KL}}\left(x_t(\cdot|s), \eta\left(2\hat{\theta}_{t+1}^x(s, \cdot) - \hat{\theta}_t^x(s, \cdot)\right)\right)$

7:         $y_{t+1}(\cdot|s) = \mathcal{P}^{\mathrm{KL}}\left(y_t(\cdot|s), -\eta\left(2\hat{\theta}_{t+1}^y(s, \cdot) - \hat{\theta}_t^y(s, \cdot)\right)\right)$

8:     **Output** $x^k = \frac{1}{T}\sum_{t=1}^T x_t$.
9:     **Stage 2** // Approximately find best response
10:     **for** $t = 0, 1, \ldots, T - 1$ **do**
11:         $\hat{\nu}_{t+1} = \texttt{Policy-Eval-}\nu(x^k, \bar{y}_t, N, \beta = \beta_n^\nu)$
12:         $\bar{y}_{t+1}(\cdot|s) = \mathcal{P}^{\mathrm{KL}}(\bar{y}_t(\cdot|s), -\eta\hat{\nu}_{t+1}(s, \cdot))$
13:     **Output** $y^k = \bar{y}_{\hat{t}}$, where $\hat{t} \in [T]$ is selected uniformly at random.

---

**Algorithm 2** $V_N = \texttt{Policy-Eval-}V\texttt{(x, y, N, }\beta\texttt{)}$

---

**Require:** Policy pair $x, y$, iteration counter $N$, step size $\beta$, initial value function estimate $V_0$.
    $\hat{x}(\cdot|s) = (1-\zeta)x(\cdot|s) + \frac{\zeta}{|A|}, \hat{y}(\cdot|s) = (1-\zeta)y(\cdot|s) + \frac{\zeta}{|B|}$.
1: **for** $n = 0, 1, \ldots, N - 1$ **do**
2:     Sample $s_n \sim \rho^{\hat{x},\hat{y}}(\cdot), a_n \sim \hat{x}(\cdot|s_n), b_n \sim \hat{y}(\cdot|s_n), s_{n+1} \sim P(\cdot|s_n, a_n, b_n)$.
3:     $V_{n+1} = V_n - \beta_n e(s_n)\left(V_n(s_n) - r(s_n, a_n, b_n) - \gamma V_n(s_{n+1})\right)$

---

where $Q_{k-1}(s, a, b) = r(s, a, b) + \gamma \sum_{s'} P(s'|s, a, b)V_{k-1}(s')$. When it is clear from the context, we drop the subscript of $Q_{k-1}$. This is a matrix game and is the sample-complexity bottleneck (Perolat et al., 2015). Let $\epsilon_g$ denote the accuracy and $x^k$ the output of this stage at iteration $k$:

$$\mathbb{E}[\mathbb{E}_{s\sim\mathcal{U}}[\max_{y^s\in\Delta}(x^k)^s Q_{k-1}^s y^s - \min_{x^s\in\Delta}\max_{y^s\in\Delta} x^s Q_{k-1}^s y^s]] = \epsilon_1^k, \tag{3}$$

where the outer expectation is over the randomness of the algorithm used to generate $x^k$.

• *Stage 2:* This step finds an approximate best response. The fixed policy ($x^k$), can be viewed as a part of the environment. Denote $y^k$ as the approximate best-response computed in this stage, at iteration $k$. The resulting value function $V_k = V^{x^k, y^k}$ is fed to stage 1 in the next iteration. Let $\epsilon_e$ be the accuracy for this stage, $y^k$ the output of this stage:

$$\mathbb{E}[\mathbb{E}_{s\sim\mathcal{U}}[\max_y V^{x^k, y}(s) - V^{x^k, y^k}(s)]] = \epsilon_2^k, \tag{4}$$

where the outer expectation is over the randomness of the algorithm used to generate $y^k$. Then, Perolat et al. (2015, Theorem 1), Zhao et al. (2021) show that the following holds (see also Appendix C).

$$\mathbb{E}[\mathbb{E}_{s\sim\mathcal{U}}[\max_{y\in\Delta} V^{x^K, y}(s) - V^\star(s)]] \leq \frac{|S|K}{1-\gamma}\tilde{\mathcal{O}}\left(\sup_{k\leq K}\epsilon_1^k + \sup_{k\leq K}\epsilon_2^k\right) + \mathcal{O}\left(\frac{|S|\gamma^K}{1-\gamma}\right). \tag{5}$$

## 3 REFLECTED NAC ALGORITHM WITH A GAME ETIQUETTE

**Our approach.** We introduce NAC-based algorithms (Konda & Tsitsiklis, 2000; Peters & Schaal, 2008) to solve these two stages in an alternating fashion to obtain an approximate Nash equilibrium, in view of (5). We leverage primal-dual algorithms to solve the matrix game in Stage 1

---

**Algorithm 3** $\hat{\theta}_N$ = `Policy-Eval-`$\theta$`(x, y, N, `$\hat{V}$`, `$\beta$`) for player` $y$

---

**Require:** $\hat{x}(\cdot|s) = (1 - \zeta)x(\cdot|s) + \frac{\zeta}{|A|}, \hat{y}(\cdot|s) = (1 - \zeta)y(\cdot|s) + \frac{\zeta}{|B|}$.

1: **for** $n = 0, 1, \ldots, N - 1$ **do**
2: Sample $s_n \sim \rho^{\hat{x},\hat{y}}(\cdot), a_n \sim \hat{x}(\cdot|s_n), b_n \sim \hat{y}(\cdot|s_n), s_{n+1} \sim P(\cdot|s_n, a_n, b_n)$.
3: $\hat{\theta}_{n+1} = \hat{\theta}_n - \beta_n e(s_n, b_n)\big(\hat{\theta}_n(s_n, b_n) - r(s_n, a_n, b_n) - \gamma\hat{V}(s_{n+1})\big)$

---

**Algorithm 4** $\hat{\nu}_N$ = `Policy-Eval-`$\nu$`(x, y, N, `$\beta$`) for player` $y$

---

**Require:** $\hat{x}(\cdot|s) = (1 - \zeta)x(\cdot|s) + \frac{\zeta}{|A|}, \hat{y}(\cdot|s) = (1 - \zeta)y(\cdot|s) + \frac{\zeta}{|B|}$.

1: **for** $n = 0, 1, \ldots, N - 1$ **do**
2: Sample $s_n \sim \rho^{\hat{x},\hat{y}}(\cdot), a_n \sim \hat{x}(\cdot|s_n), b_n \sim \hat{y}(\cdot|s_n), s_{n+1} \sim P(\cdot|s_n, a_n, b_n), b_{n+1} \sim y(\cdot|s_{n+1})$.
3: $\hat{\nu}_{n+1} = \hat{\nu}_n - \beta_n e(s_n, b_n)\big(\hat{\nu}_n(s_n, b_n) - r(s_n, a_n, b_n) - \gamma\hat{\nu}_n(s_{n+1}, b_{n+1})\big)$

---

efficiently (Malitsky & Tam, 2020; Nemirovski et al., 2009) and NPG for the single agent problem in Stage 2. For estimating value functions that are used as oracles in these algorithms, we employ TD(0) (Tsitsiklis & Van Roy, 1997; Bhandari et al., 2018; Sutton, 1988). To our knowledge, the best complexities in the single agent setting are obtained with this approach (Lan, 2021).

**Stage 1.** In this step, at iteration $k$, we compute an approximate equilibrium of the matrix game (2). As $V_{k-1}$ is fixed, this is a standard matrix game and throughout this loop, we omit the dependence of $Q$ on $k$ and leave it implicit. Our discussion here is for $x$ player and it would be symmetric for the $y$ player. Due to its simplicity and generality, we use forward-reflected-backward (FoRB) algorithm (Malitsky & Tam, 2020).[2] FoRB takes a reflected step rather than using $\mathbb{E}_{b\sim y_t}Q(s, \cdot, b)$ directly, which is the case in gradient descent-ascent (GDA). The FoRB update is

$$x_{t+1}(\cdot|s) = \mathcal{P}^{\text{KL}}\big(x_t(\cdot|s), \eta\,\big(2\mathbb{E}_{b\sim y_t}Q(s, \cdot, b) - \mathbb{E}_{b\sim y_{t-1}}Q(s, \cdot, b)\big)\big).$$

In standard matrix game notation, we can view $\theta^x_{\star,t} = \mathbb{E}_{b\sim y_t}Q(s, \cdot, b)$ as the matrix-vector multiplication between the *dual vector* $y_t$ and *game matrix* $Q$. To get this oracle, we have to solve a linear equation by sampling the policies $x_t, y_t$. The linear equation is obtained by using the definition of $Q$, given after (2). However, the difficulty is that we do not have access to $V_{k-1}$. Given that $V_{k-1}$ is the value function of policies $x^{k-1}, y^{k-1}$, in Step 4, we use TD(0) to learn this value function and obtain a *biased* estimation $\hat{V}_{k-1}$ **(see Algorithm 2)**. Using this estimation, we can then proceed to solve the linear equation by sampling the policies $x_t, y_t$ to find an estimate for $\theta^x_{\star,t} = \mathbb{E}_{b\sim y_t}Q(s, \cdot, b)$, in Step 5 **(see Algorithm 3)**. This step is similar to stochastic approximation/SGD approaches (Nemirovski et al., 2009; Lan, 2021). Using the oracle, we perform one FoRB step for each player, in Steps 6 and 7.

As we detail in App. F, for Steps 4, 5 and corresponding policy evaluation routine Alg. 3, $x$ agent only accesses $s_t, r(s_t, a_t, b_t), s_{t+1}$ and its own action $a_t$ to form the stochastic oracle and $y$ accesses $s_t, r(s_t, a_t, b_t), s_{t+1}$ and its own action $b_t$. We have additional bias coming from the approximation of $V_{k-1}$ by $\hat{V}_{k-1}$, the estimation of which is important for getting our complexity results. We take special care for the stochastic dependency to make sure to decompose bias and variance of $\hat{V}_{k-1}$ estimate (See Insight 1). Markovian data would bring additional bias as mentioned before.

**Remark 3.1.** *For the best complexity, we use fresh estimates of $\hat{V}_{k-1}$ at every iteration (see Algorithm 1 and Insight 4). This gives a tight bound for the bias to get the $\tilde{\mathcal{O}}(\epsilon^{-2})$ complexity. This insight is in contrast to the black box view of Perolat et al. (2015), which uses an estimate of $V_{k-1}$ from the stage 2 within the stage 1. Our analysis behooves both agents to remember the output policies of stage 2 instead, so that they can recompute $V_{k-1}$ with a lower bias in the stage 1.*

**Stage 2.** In this step, at iteration $k$, $x$ player fixes its policy and $y$ computes an approximate best response by solving the single agent problem in (4) by using NPG in (Lan, 2021; Agarwal et al., 2020). Value function $V_{k-1}$ that was used in stage 1 is precisely the value function of the policies outputted at this stage in iteration $k - 1$. For NPG update, $y$ player needs $\mathbb{E}_{a\sim x^k(\cdot|s)}Q^{x^k, \bar{y}_t}(\cdot, a, \cdot)$.

---

[2] In principle, this part can be replaced with mirror-prox (Nemirovski, 2004) or OGDA.

This is the joint $Q$ function after taking the expectation over the actions of player $x$. Since $x$ player's policy is fixed, we only use policy $x^k$ in this loop, whereas $y$ player continue to update its policy $\bar{y}_t$.

We can write Bellman equation for learning this oracle and then use TD(0) as in Bhandari et al. (2018) **(see Algorithm 4)**, which is similar to learning a Q-function. In particular, as long as $s_t, a_t, b_t, s_{t+1}$ are sampled using the interaction procedure described earlier, there is no need for $\bar{y}_t$ update to see the actions or policy of $x^k$. A similar formulation for policy evaluation in MARL is considered in Perolat et al. (2018) in a slightly different setting.

**Greedy exploration.** If Assumption 2 does not hold, one way to lower bound the policies is to use $\zeta$-greedy exploration which we incorporate into our algorithm. The idea of the analysis will be to pick the exploration parameters depending on the final accuracy. Since the bounds have inverse dependence with this parameter, using greedy exploration will result in a worse complexity compared to what we get when Assumption 1 holds. In the former case, we can take $\zeta = 0$.

## 3.1 MAIN RESULT

**Theorem 3.2.** *(Overall performance bound with Assumption 1, 2) For Alg. 1 with $\zeta = 0$,*

$$\mathbb{E}[\mathbb{E}_{s_0 \sim \mathcal{U}}[\max_y V^{x_k, y}(s_0) - V^\star(s_0)]] \le \mathcal{O}\left(\frac{K|S|}{T(1-\gamma)^3}\right)$$

$$+ \mathcal{O}\left(\frac{K|S|^2(|A| \vee |B|)}{N(1-\gamma)^5(\rho(1-\gamma))^2}\left(\frac{1}{(\rho \min\{\underline{x}, \underline{y}\})^2} \vee \frac{1}{(\rho \underline{y}(1-\gamma))^2}\right)\right) + \mathcal{O}\left(\gamma^K\right).$$

*In particular, the overall complexity is $\tilde{\mathcal{O}}(|S|^3(|A| \vee |B|)\epsilon^{-2}(1-\gamma)^{-12}\underline{\rho}^{-6}\underline{y}^{-2}(\min\{\underline{x}, \underline{y}\})^{-2})$.*

In this bound, the first term is for solving the outer loops in stages 1 and 2. The second term is due to the bias and the variance of the stochastic oracles that we got by sampling the policies. The final term is due to approximating generalized policy iteration in the outermost loop (Perolat et al., 2015).

**Theorem 3.3.** *(Overall performance bound with Assumption 1, see App. G.2.2) Let Assumption 1 hold. Let $\zeta = \frac{(1-\gamma)^2 \epsilon}{|A| \vee |B|}$ in Alg. 1.*

$$\mathbb{E}[\mathbb{E}_{s_0 \sim \mathcal{U}}[\max_y V^{x_k, y}(s_0) - V^\star(s_0)]] \le \mathcal{O}\left(\frac{K|S|}{T(1-\gamma)^3}\right) + \mathcal{O}\left(\frac{K|S|^3(|A| \vee |B|)^6}{N(1-\gamma)^{12}\underline{\rho}^2 \epsilon^2}\right) + \mathcal{O}\left(\gamma^K\right).$$

*Consequently, the sample complexity is $\tilde{\mathcal{O}}(|S|^4(|A| \vee |B|)^6(1-\gamma)^{-15}\epsilon^{-4}\underline{\rho}^{-2})$.*

As summarized in Table 1, Theorem 3.3 matches the best-known existing complexity from Wei et al. (2021), without using the metric subregularity constant and without restricting to Euclidean projections. For natural policy gradient, this result improves $\tilde{\mathcal{O}}(\epsilon^{-8})$ in Wei et al. (2021). We refer to Remark G.17 for a more detailed comparison. Moreover, Theorem 3.2 matches the best-known complexity in single agent RL under Assumption 1, 2.

## 3.2 CONVERGENCE ANALYSIS

**Proof sketch.** Next, we show how to obtain the required bounds for this final guarantee, in view of eq. (3), (4), (5). Our strategy is to characterize the error of each stage by using the outer/inner structure given in the algorithm. The innermost algorithms (see Alg. 2) are estimating the required oracles depending on value functions, by sampling the policies, and applying either SGD or TD(0).

As per (5), the next lemma will characterize the error of stage 1 (see (3)): solving the matrix game. A critical point to derive the fastest rate as observed by Lan (2021) in the single agent setting is to characterize the bias and variance separately. As the algorithm in Lan (2021) is akin to gradient descent, we extend the ideas there to the more complicated FoRB algorithm.

> **Insight 1.** *The existing analyses for stochastic FoRB are not suitable for us. In the stochastic variant in Malitsky & Tam (2020), deterministic oracle is computed at each iteration. Böhm et al. (2020) uses unbiased oracles with bounded variance and decreasing step size. In our case, we*

*have biased oracles and we use inner loops to decrease bias and variance. Next, we develop an analysis with constant step size and characterization of the bias and variance explicitly.*

**Lemma 3.4.** *(Bound for stage 1) Let Assumption 1, 2 hold. Denote $x_{out} = \frac{1}{T}\sum_{t=1}^{T} x_t$ and $y_{out} = \frac{1}{T}\sum_{t=1}^{T} y_t$ and let $\eta = \frac{1-\gamma}{8}$. Then, it holds that*

$$\mathbb{E}\left[\mathbb{E}_{s\sim\mathcal{U}}\left[\max_{x^s,y^s} x_{out}^s Q^s y^s - x^s Q^s y_{out}\right]\right] = \tilde{\mathcal{O}}\left(\frac{1}{\eta T}\right) + \mathcal{O}\left(\frac{1}{T}\sum_{t=1}^{T}\mathbb{E}\|\mathbb{E}[\theta_{t+1}|x_t] - \theta_{\star,t}\|\right)$$

$$+ \mathcal{O}\left(\frac{\eta}{T}\sum_{t=1}^{T}\mathbb{E}\|\theta_{t+1} - \theta_{\star,t}\|^2 + \mathbb{E}\|\theta_t - \theta_{\star,t-1}\|^2\right) + \frac{1}{T\eta}\mathbb{E}[\mathbb{E}_{s\sim\mathcal{U}}\max_{z=(x,y)}\sum_{t=1}^{T}[\mathcal{E}_{1,t}(z) + \mathcal{E}_{2,t}(z)]].$$

**Remark 3.5.** *When bias and variance are 0, this reduced to $1/T$ rate as in Zhao et al. (2021). Bounding $\epsilon_1^k$ (see (3)) is via bounding LHS of Lem. 3.4. This allows to bound the suboptimality thanks to (5). To this end, we bound the second term in RHS of Lem. 3.4 in Lem. 3.7 and the third term in Lem. 3.6.*

We drop the superscripts from $\theta, \hat{V}_{k-1}$ (see Alg. 1) as estimations are symmetric. For a free variable $z = (x, y)$, define $\mathcal{E}_{1,t}(z) + \mathcal{E}_{2,t}(z) = \eta\langle\theta_{t+1}(\cdot|s) - \mathbb{E}[\theta_{t+1}(\cdot|s)|x_t], x(\cdot|s) - x_t(\cdot|s)\rangle - \eta\langle\theta_{t+1}^y(\cdot|s) - \mathbb{E}[\theta_{t+1}^y(\cdot|s)|y_t], y(\cdot|s) - y_t(\cdot|s)\rangle$.

**Insight 2.** *The last error term in the lemma involving $\mathcal{E}_{1,t}, \mathcal{E}_{2,t}$ is due to the coupling between the free variables $x^s, y^s$ and randomness of the algorithm. For this error, we adapt the "ghost iterate" trick from Nemirovski et al. (2009) for stochastic primal-dual algorithms (see Lemma F.6).*

This lemma analyzes the behavior of FoRB for solving the matrix game with biased oracles. The bound therefore reflects the bias and variance of these oracles. For simplicity, we suppress some dependencies in the following bounds, however we include them in Thm. 3.2 and in the appendices.

Next is the the variance estimation, which is similar to Lan (2021), except handling the error term coming from $\hat{V}_{k-1}$ as in Insight 3. Apart from that subtlety, this part is similar to SGD-type analysis with a biased oracle, where we measure the squared distance of the iterate to the solution.

**Lemma 3.6.** *(Variance estimation for step 5) Let Assumption 1, 2 hold. Let $\beta_n^\theta = \frac{2}{\rho\min\{\underline{y},\underline{x}\}(n+n_0)}$ for $n_0 \geq 1$. Then, for Algorithms 1 and 3,*

$$\mathbb{E}\|\theta_N - \theta_{\star,t}\|_2^2 \leq \mathcal{O}\left(\frac{1}{N^2} + \frac{1}{(\rho\min\{\underline{y},\underline{x}\})^2}\left(\frac{1}{N} + \mathbb{E}\|\hat{V}_{k-1} - V_{k-1}\|_\infty^2\right)\right).$$

**Insight 3.** *Different from the standard critic analyses (Hong et al., 2020; Khodadadian et al., 2021b), we account for the additional bias coming from having $\hat{V}_{k-1}$ instead of real $V_{k-1}$ (see (2), Step 4). We exploit structure of the underlying problem to make sure the error term appears as $\mathbb{E}\|\hat{V}_{k-1} - V_{k-1}\|_\infty^2$ in the bound instead of $\mathbb{E}\|\hat{V}_{k-1} - V_{k-1}\|_\infty$, which would deteriorate the rate.*

The next estimation is critical for obtaining the complexity result. In particular, we bound the bias of $\theta_{t+1}$. Since in Lemma 3.4, we need a tight bound for $\|\mathbb{E}[\theta_{t+1}|x_t] - \theta_{\star,t}\| = \|\mathbb{E}[\theta_N|x_t] - \theta_{\star,t}\|$, we have to be careful with the additional bias from $\hat{V}_{k-1}$. This part is similar to SGD-type analysis with a biased oracle, where we measure the distance of the expectation of the iterate to the solution.

**Lemma 3.7.** *(Bias estimation for step 5)) Let Assumption 1, 2 hold, $\beta_n^\theta = \frac{2}{\rho\min\{\underline{x},\underline{y}\}(n+n_0)}$, $n_0 = \mathcal{O}\left(\frac{1}{(\rho\min\{\underline{x},\underline{y}\})^2}\right)$. For Algorithms 1 and 3*

$$\|\mathbb{E}[\theta_N|x_t] - \theta_{\star,t}\|^2 \leq \mathcal{O}\left(\frac{1}{N^2} + \frac{1}{(\rho\min\{\underline{x},\underline{y}\})^2}\|\mathbb{E}[\hat{V}_{k-1}|x_t] - V_{k-1}\|_\infty^2\right). \quad (6)$$

**Insight 4.** *The reason to use fresh estimates for $\hat{V}_{k-1}$ at each $t$ as in Algorithm 1 is the result of this lemma (see Remark 3.1). Since the bias term in the algorithm's analysis is $\|\mathbb{E}[\theta_N|x_t] - \theta_{\star,t}\|$ in Lemma 3.4, we take the square root of the result of Lemma 3.7. If $\hat{V}_{k-1}$ is estimated before $x_t$, then we will have $\mathbb{E}\|\hat{V}_{k-1} - V_{k-1}\|$ in the bound of Lemma 3.4, which will have the rate*

$\mathcal{O}(1/\sqrt{N})$. *On the other hand, if we estimate $\hat{V}_{k-1}$ freshly as in Algorithm 1, then we will be able to use the improved bias bound $\|\mathbb{E}[\hat{V}_{k-1}|x_t] - V_{k-1}\| \leq \mathcal{O}(1/N)$ as in the next lemma.*

The next lemma is for the estimation of the value function $V_{k-1}$ using the policies $x^{k-1}, y^{k-1}$ with TD(0). Therefore, this is an analysis for TD(0), similar to Lan (2021).

**Lemma 3.8.** *(Variance/Bias estimation for step 4)) Let Assumption 1, 2 hold and $\beta_n^\omega = \frac{2}{\rho(1-\gamma)(n+n_0)}$, with $n_0 = \mathcal{O}\left(\frac{1}{(\rho(1-\gamma))^2}\right)$. The variance and bias of $\hat{V}_{k-1}$, computed as in Algorithm 1 satisfies*

$$\|\mathbb{E}[\hat{V}_{k-1}|x_t, y_t] - V_{k-1}\|_2^2 \leq \mathcal{O}\left(N^{-2}\right), \quad \mathbb{E}\|\hat{V}_{k-1} - V_{k-1}\|_2^2 \leq \mathcal{O}\left(N^{-1}(\rho(1-\gamma))^{-2}\right).$$

Unlike stage 1, the stage 2 (finding the best response) mirrors the single agent analysis closely. Due to space constraints, we defer the details to App. F. Combining Lemma 3.4 with the result of stage 2 (which is of the same order) in (5) gives Theorem 3.2. The main idea of Thm. 3.3 is to use $\zeta$-greedy exploration to replace the policy values $\underline{x}, \underline{y}$ (see Theorem 3.2, Lemmas 3.6, 3.7, 3.8), which might be 0 without Assumption 2 (see App.G.2.2).

## 4 NUMERICAL VERIFICATION

We validate our algorithm in tabular domains comparing against OGDA (Wei et al., 2021) and REINFORCE (Daskalakis et al., 2020). More results , implementation details and environment description are given in Appendix H. We emphasize that our main contribution is theoretical; the preliminary computational results are for verification purposes.

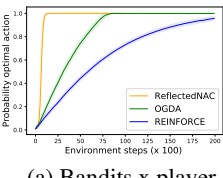
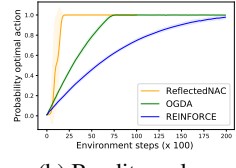
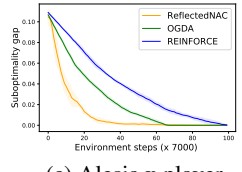
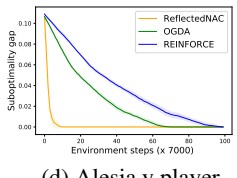

| (a) Bandits x player | (b) Bandits y player | (c) Alesia x player | (d) Alesia y player |

Figure 1: **Bandits** We plot the probability of action $a^*$ for the $x$ policy and $b^*$ for the $y$ policy. Results are averaged over 10 seeds. **Alesia**: Experiments in a Alesia with length $L = 3$ and coin budget $C = 6$. The suboptimality gap on the vertical axis is $\max_y V^{x,y}(s_0) - V^*(s_0)$ for $x$ and $|\min_x V^{x,y}(s_0) - V^*(s_0)|$ for $y$ where $s_0$ is the initial state that is deterministic in Alesia. Results are averaged over 5 seeds.

**Observations.** Both the domains challenge theoretical assumptions. Therefore, our best complexity result $\tilde{\mathcal{O}}(\epsilon^{-2})$ from Theorem 3.2 does not apply in this setting. Similarly, the assumptions in Table 1 do not hold. The assumption of (Wei et al., 2021) and our Assumption 1 do not hold because in Alesia the players can only lose coins. Therefore the initial state cannot be reached in finite time from any state. Finally the assumption of Daskalakis et al. (2020) does not hold since in Alesia the game ends only when either one player wins or both players finish their coin budget. The game ends with probability 0 in all other cases. Thus, it follows that we cannot lower bound the termination probability at any state. Nevertheless, we observe that all the algorithms converge. Figures 1a,1b shows the value of the bandit player policies evaluated at the NE actions $(a^*, b^*)$, i.e. $x(a^*)$ and $y(b^*)$. Reflected NAC converges faster than OGDA (Wei et al., 2021) and REINFORCE (Daskalakis et al., 2020). Similar conclusions arise from Figures 1c, 1d where we plot the suboptimality gap in Alesia.

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

CONTENTS

## A  BACKGROUND ON NPG AND NAC

**Natural policy gradient and natural actor-critic.**     As we work in the tabular setting, in this paper, we focused on the natural policy gradient (Kakade, 2001) in softmax parameterization which admits a simple update rule. In particular, the update rule for NPG in single agent setting is (Agarwal et al., 2020, Lemma 5.1)

$$\pi_{t+1}(\cdot|s) \propto \pi_t(\cdot|s) \exp(\eta Q^{\pi_t}(s, \cdot)),$$

which is the closed form solution of the update in (1). To get a sample-based version of this algorithm, one needs to learn $Q^{\pi_t}$ typically in an inner policy evaluation loop as in Lan (2021). This is also called natural actor-critic (NAC) since the actor updates the policy $\pi_{t+1}$ and critic learns the value function $Q^{\pi_t}$.

Note that the update rule in (Agarwal et al., 2020, Lemma 5.1) is written with the advantage function, however, due to softmax parameterization, it is equivalent to the form we give.

We can also generalize (1), by using Bregman distances instead of the KL divergence

$$x_{t+1}(\cdot|s) = \mathcal{P}(x_t(\cdot|s), Q^{x_t}(s, \cdot)) = \arg \min_{x(\cdot|s) \in \Delta} \langle Q^{x_t}(s, \cdot), x(\cdot|s) \rangle + D(x(\cdot|s), x_t(\cdot|s)), \quad (7)$$

Finally, for the formal setup of Bregman distances, we refer to Tseng (2008); Nemirovski et al. (2009). Throughout the paper, we focus on the case when $D$ is KL divergence, so that the update rule corresponds to NPG rule. This choice corresponds to the distance generating function of $D$ being strongly convex in $\ell_1$ norm which gives the standard inequality

$$D(x, y) \geq \frac{1}{2}\|x - y\|_1^2, \quad (8)$$

that is used frequently throughout the proofs.

**Single loop NAC.**     Unlike the previous case, single loop NAC (Hong et al., 2020; Khodadadian et al., 2021b) does not have an inner loop for computing $Q^{\pi_t}$ at iteration $t$. In contrast, single loop NAC keeps a running estimate for this oracle (which corresponds to one iteration of policy evaluation) and due to its two time-scale nature, still converges. In the tabular case, the simplest single loop NAC update takes the form

$$\theta_{t+1} = \theta_t - \beta_t e(s_t, a_t) \left(\theta_t(s_t, a_t) - r(s_t, a_t) - \gamma \theta_t(s_{t+1}, a_{t+1})\right) \quad (9)$$
$$\pi_{t+1}(\cdot|s) \propto \pi_t(\cdot|s) \exp(\eta_t \theta_{t+1}(s, \cdot)),$$

with properly selected $\beta_t, \eta_t$, generally with $\eta_t/\beta_t \to 0$.

**Temporal difference learning.**     For constructing state or action value functions from samples, we will use temporal difference learning and in particular TD(0) (Sutton, 1988; Bhandari et al., 2018; Tsitsiklis & Van Roy, 1997). This algorithm can be seen as a stochastic approximation scheme for solving a linear equation (Tsitsiklis & Van Roy, 1997; Lan, 2021). In particular, by denoting the stationary state distribution under $\pi$ as $\rho^\pi$, we define

$$F^\pi(\theta)(s, a) = \rho^\pi(s)\pi(a|s)\left(\theta(s, a) - r(s, a) - \gamma \sum_{s', a'} P(s'|s, a)\pi(a'|s')\theta(s', a')\right).$$

First, we note that $F^\pi(\theta^\star) = 0$ where $\theta^\star = Q^\pi$. Under Assumption 1, 2 $F^\pi$ is strongly monotone (see (Bhandari et al., 2018, Lemma 3), (Lan, 2021, Section 5.2). The main tools to show this are Assumption 1, 2 and Bellman operator being $\gamma$-contraction. Then, one can use for example (Bauschke et al., 2011, Example 22.6, Example 20.7).

One can sample $s_t \sim \rho^\pi$, $a_t \sim \pi(\cdot|s_t)$ and $s_{t+1} \sim P(\cdot|s_t, a_t)$, $a_{t+1} \sim \pi(\cdot|s_{t+1})$ and one step of TD(0) corresponds to (9). Note that under i.i.d. assumption, the update in (9) is an unbiased estimate of the update that we would get by using the true operator $F^\pi$. The results for TD(0) can be extended to Markovian setting without the i.i.d. assumption by using a uniform mixing assumption (Bhandari et al., 2018) (see also Appendix D).

## B  BASIC RESULTS ON RL AND OPTIMIZATION

**Some notation.**  We say that an operator $\theta \mapsto F(\theta)$ is $\lambda_{\min}$-strongly monotone if $\langle F(\theta_1) - F(\theta_2), \theta_1 - \theta_2 \rangle \geq \lambda_{\min} \|\theta_1 - \theta_2\|_2^2$ and $\lambda_{\max}$-Lipschitz if $\|F(\theta_1) - F(\theta_2)\|_2 \leq \lambda_{\max} \|\theta_1 - \theta_2\|_2$. These conditions can be defined with other norms, but we stick to $\ell_2$-norm for simplicity.

**Lemma B.1.** *Define $\theta_t$ recursively as $\theta_{t+1} = \theta_t - \beta_t \tilde{F}(\theta_t, \xi_t)$ where $r(s, a, b) \leq 1$ and $\tilde{F}(\theta_t, \xi_t) = e(s', a')(\theta_t(s', a') - r(s, a, b) - \gamma \theta_t(s'', a''))$ and recall the definition of $Q_k(s, a, b) = r(s, a, b) + \gamma \sum_{s'} P(s'|s, a, b) V_k(s')$. Then, it follows for any $t, k$*

$$\|\theta_t\|_\infty \leq \frac{1}{1 - \gamma},$$

$$\|\theta_t\|_2 \leq \frac{\sqrt{|S||A|}}{1 - \gamma},$$

$$\|\hat{V}_{k-1}\|_\infty \leq \frac{1}{1 - \gamma},$$

$$\|\tilde{F}(\theta_t, \xi_t)\|_2 \leq \frac{3}{1 - \gamma},$$

$$\|Q_k(s, a, b)\|_\infty \leq \frac{2}{1 - \gamma}.$$

*Proof.* The first inequality is proven by induction, for example see (Khodadadian et al., 2021b, Lemma C.10). Following inequalities are either basic consequences of the first inequality or directly follow from definition. □

A classical result that we use frequently in the proofs is performance difference lemma by Kakade & Langford (2002). The statement of the lemma is slightly different due to multi agent setting, but since one policy is held fixed while changing the other one, the original proof of the lemma extends straightforwardly. The proof for this case is given in Daskalakis et al. (2020).

**Lemma B.2** (Performance difference lemma. See Kakade & Langford (2002); Daskalakis et al. (2020))**.** *For any policies $x, y_1, y_2$ and any state $s_0$*

$$V^{x, y_1}(s_0) - V^{x, y_2}(s_0) = \frac{1}{1 - \gamma} \mathbb{E}_{s \sim d_{s_0}^{x, y_1}} \langle \mathbb{E}_{a \sim x(\cdot|s)} Q^{x, y_2}(s, a, \cdot), y_1(\cdot|s) - y_2(\cdot|s) \rangle$$

A standard result that we use is Lipschitzness of $y \mapsto V^{x, y}(s_0)$. For example, see (Hong et al., 2020, Lemma 7). We provide proofs as we use the precise constants and use them slightly differently than Hong et al. (2020).

**Lemma B.3.** *For any policies $x, y_1, y_2$,*

$$\|V^{x, y_1} - V^{x, y_2}\|_\infty \leq \frac{2}{(1 - \gamma)^2} \max_s \|y_1(\cdot|s) - y_2(\cdot|s)\|_1.$$

*Proof.* By the performance difference lemma Kakade & Langford (2002) and Cauchy-Schwarz inequality, for any $s_0$,

$$V^{x, y_1}(s_0) - V^{x, y_2}(s_0) = \frac{1}{1 - \gamma} \mathbb{E}_{s \sim d_{s_0}^{x, y_1}} \langle \mathbb{E}_{a \sim x(\cdot|s)} Q^{x, y_2}(s, a, \cdot), y_1(\cdot|s) - y_2(\cdot|s) \rangle$$

$$\leq \frac{1}{1 - \gamma} \mathbb{E}_{s \sim d_{s_0}^{x, y_1}} \|\mathbb{E}_{a \sim x(\cdot|s)} Q^{x, y_2}(s, a, \cdot)\|_\infty \|y_1(\cdot|s) - y_2(\cdot|s)\|_1.$$

Next, we are going to further upper bound the right hand side using Lemma B.1

$$V^{x, y_1}(s_0) - V^{x, y_2}(s_0) \leq \frac{1}{1 - \gamma} \max_s \|\mathbb{E}_{a \sim x(\cdot|s)} Q^{x, y_2}(s, a, \cdot)\|_\infty \|y_1(\cdot|s) - y_2(\cdot|s)\|_1$$

$$\leq \frac{2}{(1 - \gamma)^2} \max_s \|y_1(\cdot|s) - y_2(\cdot|s)\|_1.$$

We take maximum over $s_0$ to conclude. □

**Lemma B.4.** *We have*

$$\|Q^{x,y_1} - Q^{x,y_2}\|_\infty \le \frac{2\gamma}{(1-\gamma)^2} \max_s \|y_1(\cdot|s) - y_2(\cdot|s)\|_1.$$

*Proof.* We note that by the definition of $Q^{x,y_1}$ it follows that for all $s, a, b$

$$|Q^{x,y_1}(s,a,b) - Q^{x,y_2}(s,a,b)| = \gamma \left| \sum_{s'} P(s'|s,a,b)\left(V^{x,y_1}(s') - V^{x,y_2}(s')\right) \right|.$$

Jensen's inequality, and the previous lemma gives the result. $\qquad\square$

**Lemma B.5.** *Let $z_t$ be defined as for all $s$,*

$$z_{t+1}(\cdot|s) = P(z_t(\cdot|s), \eta_t \theta_{t+1}(s, \cdot)),$$

*where $\mathcal{P}$ is defined in (1). In particular,*

$$\mathcal{P}(z_t(\cdot|s), \theta_{t+1}(s, \cdot)) = \arg\min_{z(\cdot|s)\in\Delta} \langle \eta_t \theta_{t+1}(s, \cdot), z(\cdot|s) \rangle + D(z(\cdot|s), z_t(\cdot|s)).$$

*Then, it holds that for all $s$,*

$$\|z_{t+1}(\cdot|s) - z_t(\cdot|s)\|_1 \le \eta_t \|\theta_{t+1}\|_\infty.$$

*Proof.* By the update rule of $z_t$, it holds for all $z$ that (for example, see (Tseng, 2008, Property 1))

$$\langle \nabla D(z_{t+1}(\cdot|s), z_t(\cdot|s)) + \eta_t \theta_{t+1}(s, \cdot), z(\cdot|s) - z_{t+1}(\cdot|s) \rangle \ge 0.$$

By plugging in $z = z_t$ and using three point identity gives

$$D(z_{t+1}(\cdot|s), z_t(\cdot|s)) + D(z_t(\cdot|s), z_{t+1}(\cdot|s)) \le \eta_t \langle \theta_{t+1}(s, \cdot), z_t(\cdot|s) - z_{t+1}(\cdot|s) \rangle.$$

By (8) and Cauchy-Schwarz inequality gives

$$\|z_{t+1}(\cdot|s) - z_t(\cdot|s)\|_1 \le \eta_t \|\theta_{t+1}(s, \cdot)\|_\infty.$$

The result follows by $\|\theta_{t+1}(s, \cdot)\|_\infty \le \|\theta_{t+1}\|_\infty$. $\qquad\square$

**Lemma B.6.** *We have that $\max_y V^{x_k,y} - V^{x_\star,y_\star} = 0$ iff $x_k$ is in the set of Nash equilibrium points.*

*Proof.* Recall that we say that $(x^\star, y^\star)$ is a Nash equilibrium if for any $x, y$

$$x_\star Q^{x_\star,y} y \le x_\star Q^{x_\star,y_\star} y_\star \le x Q^{x,y_\star} y_\star.$$

In particular, it is true when we plug in $x = x_k, y = y_k$.

Next, by definition of min operation, one can bound $\min_x x Q^{x,y_k} y_k$ and by the definition of max operation, $x_k Q^{x_k,y_\star} y_\star \le \max_y x_k Q^{x_k,y} y$. In sum, we have

$$\min_x x Q^{x,y_k} y_k \le x_\star Q^{x_\star,y_k} y_k \le x_\star Q^{x_\star,y_\star} y_\star \le x_k Q^{x_k,y_\star} y_\star \le \max_y x_k Q^{x_k,y} y \qquad (10)$$

Of course, by definition, for any $x, y$, $V^{x,y} = x Q^{x,y} y$.

This is easy to see: let $x_k = x_\star$ for any $x_\star$ which is a Nash equilibrium, then $\max_y V^{x_\star,y} = V^{x_\star,y_\star}$. Now assume that $\max_y V^{x_k,y} = V^{x_\star,y_\star}$, then by (10) it must be that $\max_y V^{x_k,y} = V^{x_k,y_\star} = V^{x_\star,y_\star}$. Hence $x_k = x_\star$. $\qquad\square$

## C  ERROR PROPAGATION FRAMEWORK

Error propagation of generalized policy iteration for Markov games is given in Perolat et al. (2015).

**Notation.**  We define the Bellman operators following Perolat et al. (2015)

$$T_{x,y}V(s) = \sum_{a,b} x(a|s)y(b|s)r(s,a,b) + \gamma \sum_{s',a,b} x(a|s)y(b|s)P(s'|s,a,b)V(s')$$

$$T_x V(s) = \max_y T_{x,y}V(s)$$

$$TV(s) = \min_x \max_y T_{x,y}V(s).$$

It is easy to derive that these operators are contractions in $\ell_\infty$ norm with constant $\gamma$. See also Perolat et al. (2015); Zhao et al. (2021).

We find the one-sided Nash equilibrium as Daskalakis et al. (2020) and Zhao et al. (2021).

$$\max_y V^{x,y}(s) - V^\star(s) \leq \epsilon.$$

This is in contrast to Wei et al. (2021) that shows the rate in the duality gap. Two phases are characterized in Perolat et al. (2015) as

• Phase 1: $T_{x_k}V_{k-1} \approx TV_{k-1}$. By using the definitions of $T, T_x$ this corresponds to

$$\max_y T_{x_k,y}V_{k-1} - \min_x \max_y T_{x,y}V_{k-1}, \tag{11}$$

where

$$T_{x,y}V_{k-1}(s) = \sum_{a,b} x(a|s)y(b|s)r(s,a,b) + \gamma \sum_{s',a,b} x(a|s)y(b|s)P(s'|s,a,b)V_{k-1}(s').$$

As $V_{k-1}$ is fixed, this will give a standard matrix game for all $s$. By using a stochastic algorithm, we are going to make the output $x_k$ to be an approximate solution in expectation, therefore we write

$$\mathbb{E} \max_y x_k^s Q_{k-1}^s y^s - \min_x \max_y x^s Q_{k-1}^s y^s = \epsilon_1^k(s), \tag{12}$$

where the expectation is over the randomness of the specific algorithm used to generate $x_k$.

In our analysis, we bound the stronger quantity, which is called duality gap

$$\mathbb{E}\Big[\max_y x_k^s Q_{k-1}^s y^s - \min_x x^s Q_{k-1}^s y_k^s\Big] \geq \epsilon_1^k(s),$$

by the definition of a Nash equilibrium since $\min_x x^s Q_{k-1}^s y_k^s \leq \min_x \max_y x^s Q_{k-1}^s y^s$.

• Phase 2: $V_k \approx (T_{x_k})^m V_{k-1}$. Since $T_{x_k}$ is a contraction Perolat et al. (2015); Zhao et al. (2021), as $m \to \infty$, for any $V$, $T_{x_k}V \to \max_y V^{x_k,y}$. Let us denote the best response as $y_k^*$ and the approximate best response as $y_k$. We want to bound

$$V^{x_k,y_k^*}(s) - \mathbb{E}V^{x_k,y_k}(s) = \epsilon_2^k(s), \tag{13}$$

where the expectation is over the randomness of the algorithm used to generate $y_k$.

Then (Perolat et al., 2015, Theorem 1) states that (where $\max_y x_k^s Q_{k-1}^s y^s - \min_x \max_y x^s Q_{k-1}^s y^s = \epsilon'_k(s)$, $V^{x_k,y_k^*}(s) - V^{x_k,y_k}(s) = \epsilon_k(s)$)

$$\sum_s \mu(s)\left(\max_y V^{x_k,y}(s) - V^\star(s)\right) \leq \frac{2(\gamma - \gamma^k)}{(1-\gamma)^2} C_\infty^{1,k,0} \sup_{j \in [0,k-1]} \|\epsilon_j\|_{1,\sigma}^2$$

$$+ \frac{(1-\gamma^k)C_\infty^{0,k,0}}{(1-\gamma)^2} \sup_{j \in [1,k]} \|\epsilon'_j\|_{1,\sigma} + \frac{2\gamma^k}{1-\gamma} C_\infty^{k,k+1,0} \min(\|d_0\|_{1,\sigma}, \|b_0\|_{1,\sigma}),$$

where

$$C_q^{l,k,d} = \frac{(1-\gamma)^2}{\gamma^l - \gamma^k} \sum_{i=l}^{k-1} \sum_{j=i}^\infty \gamma^j c_q(j+d)$$

with

$$c_q(j) = \sup_{x_1,y_1,\ldots,x_j,y_j} \left\| \frac{d(\mu P_{x_1,y_1}\ldots P_{x_j,y_j})}{d\sigma} \right\|_{q,\sigma}$$

For given state distributions $\mu, \sigma$, let us define the concentrability coefficient Perolat et al. (2015):

$$\sup_j \sup_{x_1,y_1,\ldots,x_j,y_j} \left\| \frac{\mu P_{x_1,y_1}\ldots P_{x_j,y_j}}{\sigma} \right\|_\infty =: C_{\mu,\sigma} < +\infty.$$

In particular, by upper bounding $c_q(j) \leq C_{\mu,\sigma}$ for simplicity (one can also use the tighter bounds; we use the loose upper bounds for simplicity as they only affect the final bound slightly), the bound becomes

$$\sum_s \mu(s) \left( \max_y V^{x_k,y}(s) - V^\star(s) \right) \leq \frac{2kC_{\mu,\sigma}}{1-\gamma} \sup_{j\in[0,k-1]} \|\epsilon_j\|_{1,\sigma}^2$$

$$+ \frac{kC_{\mu,\sigma}}{(1-\gamma)} \sup_{j\in[1,k]} \|\epsilon'_j\|_{1,\sigma} + \frac{2\gamma^k C_{\mu,\sigma}}{1-\gamma},$$

where we also used an estimation from (Zhao et al., 2021, Lemma 2).

One important point here is that we will be making sure the inequalities in these stages hold in expectation. This is also pointed out in Zhao et al. (2021) with a short explanation. We describe here the details needed to ensure that these bounds hold in expectation. For this, we have to track the analysis in Perolat et al. (2015) and in Scherrer et al. (2012) where the derivations in Perolat et al. (2015) build on. In particular, the relations in (Perolat et al., 2015, Lemma 1) are linear and therefore, would also hold in expectation. Then, in derivation of (Perolat et al., 2015, Theorem 1), the arguments in (Scherrer et al., 2012, Lemma 2, Lemma 3) are used. Tracking (Scherrer et al., 2012, Lemma 3), we see that after taking the total expectation, the bounds become

$$\sum_s \mu(s) \left( \mathbb{E} \max_y V^{x_k,y}(s) - V^\star(s) \right) \leq \frac{2kC_{\mu,\sigma}}{1-\gamma} \sup_{j\in[0,k-1]} \|\epsilon_1^j\|_{1,\sigma}^2$$

$$+ \frac{kC_{\mu,\sigma}}{(1-\gamma)} \sup_{j\in[1,k]} \|\epsilon_2^j\|_{1,\sigma} + \frac{2\gamma^k C_{\mu,\sigma}}{1-\gamma}, \quad (14)$$

where $\epsilon_{g,j}$ and $\epsilon_{e,j}$ are as defined in (12), (13).

**Remark C.1.** *For simplicity, throughout the paper we take $\sigma, \mu$ to be the uniform distribution and hence replaced $C_{\mu,\sigma}$ by its worst case value $|S|$. As mentioned in Munos (2003), this value can be much smaller in general.*

## D  MARKOVIAN BIAS

As mentioned before, in this setting, the Markovian error is essentially additive in our arguments for policy evaluation steps, and can be bounded by using uniform mixing assumption. In particular, this assumption holds when the induced Markov Chain over the states, for any policy pair is aperiodic and irreducible (Lan, 2021; Khodadadian et al., 2021b). In this chapter, we give an informal explanation to illustrate how Markovian sampling can be incorporated into our proofs with the uniform mixing assumption. The main references for this kind of analysis is Lan (2021); Bhandari et al. (2018) for Algorithm 1 and Khodadadian et al. (2021b); Zou et al. (2019) for single loop NAC.

We are going to sketch the arguments for Appendix F.1 which will be applicable also to other policy evaluation routines.

Recall that by using the oracle for stage 1, we can write

$$F(\theta_n)(s,a) = \rho^{x_t,y_t}(s)x_t(a|s)\Big(\theta_n(s,a) - \sum_b y_t(b|s)r(s,a,b)$$

$$- \gamma \sum_{s',b} y_t(b|s)P(s'|s,a,b)V_{k-1}(s')\Big), \quad (15)$$

and

$$\tilde{F}(\theta_n, \xi_n) = e(s_n, a_n) \left( \theta_n(s_n, a_n) - r(s_n, a_n, b_n) - \gamma \hat{V}_{k-1}(s_{n+1}) \right).$$

This time we have a Markovian data stream and we denote $\xi_n = (s_n, a_n, b_n, s_{n+1})$. Unlike the i.i.d. case $\mathbb{E}_{\xi_n} \tilde{F}(\theta_n, \xi_n) \neq F(\theta_n)$.

Now we inspect the place in the proof of Lemma 3.6 where we used this estimation. Recall that the term we have

$$-\langle \tilde{F}(\theta_n, \xi_n), \theta_n - \theta_\star \rangle,$$

where we take conditional expectation in Lemma 3.6. As we can no longer compute the expectation, we are going to identify the error term

$$-\langle \tilde{F}(\theta_n, \xi_n), \theta_n - \theta_\star \rangle = -\langle F(\theta_n), \theta_n - \theta_\star \rangle \underbrace{-\langle \tilde{F}(\theta_n, \xi_n) - F(\theta_n), \theta_n - \theta_\star \rangle}_{err(n)}.$$

We now separate the error of $\hat{V}_{k-1}$ and identify the Markovian error, since we separately handled the error due to $\hat{V}_{k-1}$ in Lemma 3.6. Let us define $\check{F}(\theta_n, \xi_n) = e(s_n, a_n) (\theta_n(s_n, a_n) - r(s_n, a_n, b_n) - \gamma V_{k-1}(s_{n+1}))$

$$err(n) = \underbrace{-\langle \check{F}(\theta_n, \xi_n) - F(\theta_n), \theta_n - \theta_\star \rangle}_{\zeta(\theta_n, \xi_n)} -\langle \tilde{F}(\theta_n, \xi_n) - \check{F}(\theta_n, \xi_n), \theta_n - \theta_\star \rangle.$$

We notice that the last term in the above bound is simply $-\gamma \langle e(s_n, a_n)(V_{k-1}(s_{n+1}) - \hat{V}_{k-1}(s_{n+1})), \theta_n - \theta_\star \rangle$ which can be bounded as in Lemma 3.6. Therefore we focus on the Markovian error which is defined as $\zeta(\theta_n, \xi_n)$.

We will argue as in Bhandari et al. (2018). First, it is easy to see that as in Bhandari et al. (2018), $\theta_n \mapsto \check{F}(\theta_n, \xi_n)$, $\theta_n \mapsto F(\theta_n)$ and $\theta \mapsto \theta_n - \theta_\star$ are all Lipschitz. Therefore, it follows for some constant $C_1$ that

$$|\zeta(\theta_n, \xi_n) - \zeta(\theta_{n-\tau}, \xi_n)| \leq C_1 \|\theta_n - \theta_{n-\tau}\|_2.$$

By triangle inequality and using the update rule $\theta_{n+1} = \theta_n - \beta_n \tilde{F}(\theta_n, \xi_n)$ along with Lemma B.1 give

$$\zeta(\theta_n, \xi_n) \leq \zeta(\theta_{n-\tau}, \xi_n) + \frac{3C_1}{1-\gamma} \sum_{i=n-\tau}^{n-1} \beta_i.$$

Next, we bound $\mathbb{E}\zeta(\theta_{n-\tau}, \xi_n)$ as in Lan (2021). Let $\mathcal{F}_{n-1}$ be the filtration generated by $\xi_0, \ldots, \xi_{n-1}$ and note that $\theta_n$ depends on the same randomness as $\mathcal{F}_{n-1}$ for all $n$. In particular, by tower property,

$$\begin{aligned}
\mathbb{E}\zeta(\theta_{n-\tau}, \xi_n) &= \mathbb{E}\langle F(\theta_{n-\tau}) - \check{F}(\theta_{n-\tau}, \xi_n), \theta_{n-\tau} - \theta_\star \rangle \\
&= \mathbb{E}\langle F(\theta_{n-\tau}) - \mathbb{E}[\check{F}(\theta_{n-\tau}, \xi_n)|\mathcal{F}_{n-\tau-1}], \theta_{n-\tau} - \theta_\star \rangle \\
&\leq 2\mathbb{E}\|F(\theta_{n-\tau}) - \mathbb{E}[\check{F}(\theta_{n-\tau}, \xi_n)|\mathcal{F}_{n-\tau-1}]\| \\
&\leq C\rho^\tau,
\end{aligned}$$

for some $C$, where the last bound can be derived the same as (Lan, 2021, Lemma 16) under the assumption that the induced Markov chain is aperiodic and irreducible.

Then, one can use the arguments in Bhandari et al. (2018) by picking $\tau \sim \tau^{mix} \sim \frac{\log(1/\epsilon)}{\log(1/\rho)}$ if $n > \tau^{mix}$ and $\tau = n$ if $n \leq \tau^{mix}$ and using the step size rule of $\beta_n$ which decays as $1/n$, the same as Bhandari et al. (2018); Lan (2021) which will only add logarithmic spurious terms to the final complexity.

In the case of single loop NAC, the arguments are slightly more involved, however, they are well-studied. In particular, Zou et al. (2019) introduced the technique to handle Markovian noise for SARSA. These arguments are used for single loop NAC in Khodadadian et al. (2021b); Wu et al. (2020); Xu et al. (2020a), which also applies in our setting for Algorithm 5, similar to how the above arguments apply in our setting for Algorithm 1.

---

**Algorithm 5** Single loop NAC with a game etiquette

---

**Require:** $\hat{V}_0$ such that $\|\hat{V}_0 - V^{x_0,y_0}\|_\infty^2 \leq \epsilon$
  **for** $k = 1, 2, \ldots$ **do**
    **Stage 1**
    **for** $t = 0, 1, \ldots, T - 1$ **do**
      Sample $(s_t, a_t, b_t, s_{t+1})$, with policy pair $x_t, y_t$ observe $s_t, b_t, r(s_t, a_t, b_t), s_{t+1}$
      $\theta_{t+1}^x = \theta_t^x - \beta_t e(s_t, a_t)\big(\theta_t^x(s_t, a_t) - r(s_t, a_t, b_t) - \gamma \hat{V}_{k-1}(s_{t+1})\big)$
      $\theta_{t+1}^y = \theta_t^y - \beta_t e(s_t, b_t)\big(\theta_t^x(s_t, b_t) - r(s_t, a_t, b_t) - \gamma \hat{V}_{k-1}(s_{t+1})\big)$
      $[x_{t+1}(\cdot|s),\ y_{t+1}(\cdot|s)] = [\mathcal{P}(x_t(\cdot|s), \eta\theta_{t+1}^x(s, \cdot)),\ \mathcal{P}(y_t(\cdot|s), -\eta\theta_{t+1}^y(s, \cdot))]$
    Output $x_k = \frac{1}{T}\sum_{t=1}^T x_t$.
    **Stage 2**
    **for** $t = 0, 1, \ldots, T - 1$ **do**
      Sample $(s_t, a_t, b_t, s_{t+1}, b_{t+1})$ with policy pair $x_k, \bar{y}_t$, observe $s_t, b_t, r(s_t, a_t, b_t), s_{t+1}, b_{t+1}$
      $\nu_{t+1} = \nu_t - \beta_t^\nu e(s_t, b_t)\left(\nu_t(s_t, b_t) - r(s_t, a_t, b_t) - \gamma\nu_t(s_{t+1}, b_{t+1})\right)$
      $\omega_{t+1} = \omega_t - \beta_t^\omega e(s_t)\left(\omega_t(s_t) - r(s_t, a_t, b_t) - \gamma\omega_t(s_{t+1})\right)$
      $\bar{y}_{t+1}(\cdot|s) = \mathcal{P}(\bar{y}_t(\cdot|s), -\eta\theta_{t+1}(s, \cdot))$
    Output $y_k = \bar{y}_{\hat{t}}$, $\hat{V}_k = \omega_{\hat{t}+1}$, where $\hat{t} \in [T]$ is selected uniformly at random and $V_k = V^{x_k,y_k}$.

---

# E   Single loop NAC with etiquette

**Theorem E.1.** *Let Assumption 1, 2 hold and $\mu$ be a state distribution. For Algorithm 5, for the output of $x$-player*

$$\mathbb{EE}_{s_0\sim\mu}[\max_y V^{x_k,y}(s_0)] - V^\star(s_0) \leq \frac{kC_{\mu,\sigma}}{T^{1/4}(1-\gamma)}\left\{\frac{\sqrt{|S|(|A|\vee|B|)}}{(1-\gamma)^2} + \frac{1}{\lambda_{\min}^\theta(1-\gamma)}\right.$$
$$+ \frac{|S|\sqrt{|A|\vee|B|}}{(1-\gamma)\lambda_{\min}^\theta} + \frac{\sqrt{|S|(|A|\vee|B|)}}{\lambda_{\min}^\omega\lambda_{\min}^\theta(1-\gamma)} + \frac{|S|\sqrt{|A|\vee|B|}}{(1-\gamma)^3\lambda_{\min}^\theta}$$
$$+ \frac{\sqrt{|S||B|}}{(1-\gamma)^2} + \frac{1}{\lambda_{\min}^\nu(1-\gamma)^2} + \left.\frac{\sqrt{|S||B|}}{(1-\gamma)^4}\right\} + \mathcal{O}\left(\frac{\gamma^k C_{\mu,\sigma}}{1-\gamma}\right)$$

*which gives $\tilde{\mathcal{O}}\left(\frac{C_G^4|S|^4(|A|\vee|B|)^2}{\epsilon^4(1-\gamma)^{20}(\lambda_{\min}^\theta\lambda_{\min}^\omega\lambda_{\min}^\nu)^4}\right)$ sample complexity.*

*Proof.* Inserting the results of Lemmas E.7, E.11 and E.14 to (14) gives the result. $\square$

## E.1   Proofs for stage 1 of single loop NAC with etiquette in

In this part, we are going to formulate and present the results for solving stage 1 with single loop NAC. Unlike Zhao et al. (2021), we do not assume to have an unbiased access to $V_{k-1}$. Therefore, we have a stochastic oracle involving $\hat{V}_{k-1}$ which is an estimate of $V_{k-1}$. We characterize the error from this term and note that the goal of stage 2 will be to provide this oracle $\hat{V}_{k-1}$ with small error. Therefore, the results in this part will contain the error term $\mathbb{E}\|\hat{V}_{k-1} - V_{k-1}\|_\infty^2$

### E.1.1   Formulation

**Notation.** The problem is for all $s$

$$\min_{x(\cdot|s)}\max_{y(\cdot|s)}\sum_{a,b} x(a|s)y(b|s)Q_{k-1}(s,a,b) =: x^s Q_{k-1}^s y^s,$$

$$\text{where } Q_{k-1}(s,a,b) = r(s,a,b) + \gamma\sum_{s'} P(s'|s,a,b)V_{k-1}(s'),$$

where we also defined $x^s, y^s, Q^s$.

Here, $V_{k-1}$ is fixed (independent of $x, y$ and iteration counter $t$), therefore the problem is standard matrix game, with a restricted access to game matrix $Q_{k-1}(s, a, b)$. In particular, $V_{k-1} \approx V^{x_{k-1}, y_{k-1}}$. For all $s$, the equilibrium condition is for all $x, y$

$$x_\star^s Q_{k-1}^s y^s \leq x_\star^s Q_{k-1}^s y_\star^s \leq x^s Q_{k-1}^s y_\star^s.$$

For lighter notation, we refer to $Q_{k-1}$ as $Q$ since it is fixed during the loop.

At iteration $t$ of solving this matrix game, we will need the oracles $\mathbb{E}_{b \sim y_t(\cdot|s)} Q(s, a, b)$ and $\mathbb{E}_{a \sim x_t(\cdot|s)} Q(s, a, b)$ for the $x$ player and $y$ player, respectively. This part is symmetric.

Let us write the oracle for $x$ variable by using the definition of $Q(s, a, b) = Q_{k-1}(s, a, b)$:

$$\theta_{\star,t}^x(s, a) = \mathbb{E}_{b \sim y_t(\cdot|s)} Q(s, a, b) \tag{16}$$

$$= \sum_b y_t(b|s) r(s, a, b) + \gamma \sum_{s', b} y_t(b|s) P(s'|s, a, b) V_{k-1}(s').$$

Given the sampling matrix $\text{diag}(x_t) \otimes \text{diag}(\rho^{x_t, y_t})$ as in (Lan, 2021, Sec. 5.2), where $\rho^{x_t, y_t}$ is the steady state distribution under $x_t, y_t$; for the critic, define the operator

$$F_t^x(\theta)(s, a) = \rho^{x_t, y_t}(s) x_t(a|s) \Big[ \theta(s, a) - \sum_b y_t(b|s) r(s, a, b)$$

$$- \gamma \sum_{s', b} y_t(b|s) P(s'|s, a, b) V_{k-1}(s') \Big], \quad (17)$$

which, by Assumption 1, 2 is strongly monotone with $\lambda_{\min}^\theta$ and we would like to find $\theta_{\star,t}$ such that $F_t^x(\theta_{\star,t}^x) = 0$. By (16), $\theta_{\star,t}^x(s, a) = \mathbb{E}_{b \sim y_t(\cdot|s)} Q(s, a, b)$. Since sampling matrix is positive definite due to abovementioned assumptions, the solution of (17) is unique.

As we do not have access to true $V_{k-1}$, we have the stochastic operator with the estimate $\hat{V}_{k-1}$ and by sampling $\xi_t = (s_t, a_t, b_t, s_{t+1})$, $s_t \sim \rho^{x_t, y_t}$, $a_t \sim x_t(\cdot|s_t)$, $b_t \sim y_t(\cdot|s_t)$, $s_{t+1} \sim P(\cdot|s_t, a_t, b_t)$

$$\tilde{F}_t(\theta_t, \xi_t) = e(s_t, a_t) \left( \theta_t(s_t, a_t) - r(s_t, a_t, b_t) - \gamma \hat{V}_{k-1}(s_{t+1}) \right).$$

By assuming we can sample from the stationary state distributions $\rho^{x_t, y_t}$,

$$\mathbb{E}_{\xi_t} \left[ \tilde{F}_t(\theta_t, \xi_t) + e(s_t, a_t) \gamma \left( V_{k-1}(s_{t+1}) - V_{k-1}(s_{t+1}) \right) \right]$$

$$= F_t(\theta_t) + \sum_{s, a, b, s'} \rho^{x_t, y_t}(s) x_t(a|s) y_t(b|s) P(s'|s, a, b) e(s, a) \left( V_{k-1}(s') - \hat{V}_{k-1}(s') \right)$$

$$=: F(\theta_t) + \gamma P_{x_t, y_t}(V_{k-1} - \hat{V}_{k-1}) = F(\theta_t) + \delta_{v,t}, \quad (18)$$

where we defined the matrix $P_{x_t, y_t}$.

Above, the first equality is due to

$$\mathbb{E}_{\xi_t} \left[ e(s_t, a_t) \left( \theta_t(s_t, a_t) - r(s_t, a_t, b_t) - \gamma V_{k-1}(s_{t+1}) \right) \right]$$

$$= \sum_{s, a, b, s'} \Pr(s, a, b, s') e(s, a) \left( \theta_t(s, a) - r(s, a, b) - \gamma V_{k-1}(s') \right)$$

$$= \sum_{s, a, b, s'} \rho^{x_t, y_t}(s) x_t(a|s) y_t(b|s) P(s'|s, a, b) e(s, a) \left( \theta_t(s, a) - r(s, a, b) - \gamma V_{k-1}(s') \right)$$

$$= \sum_{s, a} \rho^{x_t, y_t}(s) x_t(a|s) e(s, a) \left( \theta_t(s, a) - \sum_b y_t(b|s) r(s, a, b) - \gamma \sum_{s', b} y_t(b|s) P(s'|s, a, b) V_{k-1}(s') \right),$$

where the last line is due to $\sum_{b, s'} y_t(b|s) P(s'|s, a, b) \theta_t(s, a) = \theta_t(s, a) \sum_{b, s'} \Pr(s', b|s, a) = \theta_t(s, a)$ and $\sum_{b, s'} y_t(b|s) P(s'|s, a, b) r(s, a, b) = \sum_b y_t(b|s) r(s, a, b) \sum_{s'} P(s'|s, a, b) = \sum_b y_t(b|s) r(s, a, b)$.

### E.1.2 Proofs

We will drop the superscript on $\theta^x, \theta^y$ as the algorithms are symmetric, so we will only analyze one case.

**Lemma E.2.** *Under Assumption 1, 2, let $\beta_t = \frac{1}{\lambda_{\min}^\theta t^{1/2}}$ and $\eta_t = \frac{1}{t^{3/4}}$. Then for the critic computed in stage 1 of Algorithm 5,*

$$\frac{1}{T}\sum_{t=1}^{T}\mathbb{E}\|\theta_{t+1} - \theta_{\star,t}\|_2^2 \leq \frac{2}{\sqrt{T}}\|\theta_1 - \theta_{\star,t}\|_2^2 + \frac{2(1 + \log T)}{(\lambda_{\min}^\theta)^2(1-\gamma)^2\sqrt{T}}$$
$$+ \frac{16|S||A|(1 + \log T)}{\sqrt{T}(1-\gamma)^4} + \frac{4\gamma^2|S||A|}{(\lambda_{\min}^\theta)^2}\mathbb{E}\|\hat{V}_{k-1} - V_{k-1}\|_\infty^2.$$

**Corollary E.3.** *Extracting only the dependence on $\lambda_{\min}, |S|, |A|, \gamma, T$ gives*

$$\frac{1}{T}\sum_{t=1}^{T}\mathbb{E}\|\theta_{t+1} - \theta_{\star,t}\|_2^2 \leq \frac{1}{\sqrt{T}}\left\{\mathcal{O}\left(\frac{|S||A|}{(1-\gamma)^2}\right) + \tilde{\mathcal{O}}\left(\frac{1}{(\lambda_{\min}^\theta)^2(1-\gamma)^2}\right) + \tilde{\mathcal{O}}\left(\frac{|S||A|}{(1-\gamma)^4}\right)\right\}$$
$$+ \mathbb{E}\|\hat{V}_{k-1} - V_{k-1}\|_\infty^2\mathcal{O}\left(\frac{\gamma^2|S||A|}{(\lambda_{\min}^\theta)^2}\right).$$

**Remark E.4.** *Different from the standard critic analyses, we have to account for the additional bias coming from only having $\hat{V}_{k-1}$ instead of real $V_{k-1}$.*

**Remark E.5.** *Important point here is to exploit strong monotonicity of $F_t$ defined in (17), to make the error term $\mathbb{E}\|\hat{V}_{k-1} - V_{k-1}\|_\infty^2$ appear instead of the worse term $\mathbb{E}\|\hat{V}_{k-1} - V_{k-1}\|_\infty$ which would deteriorate the complexity.*

**Remark E.6.** *With some extra work, we can obtain a step size $\beta_t$ not depending on $\lambda_{\min}^\theta$, similar to Khodadadian et al. (2021b). We do not pursue this for brevity and keeping the analysis simple.*

*Proof.* Let us recall $\theta_{\star,t}(s,a) = \mathbb{E}_{b\sim y_t(\cdot|s)}Q(s,a,b)$ (16) and that $F_t(\theta_{\star,t}) = 0$ by the definition of $F_t$ in (17). Moreover, $\theta_{t+1} = \theta_t - \beta_t\tilde{F}_t(\theta_t, \xi_t)$. Analyzing the update rule of critic in the standard way (for example see (Hong et al., 2020, Proof of Thm. 3)), gives

$$\|\theta_{t+1} - \theta_{\star,t}\|_2^2 = \|\theta_t - \theta_{\star,t}\|_2^2 - 2\beta_t\langle\tilde{F}_t(\theta_t, \xi_t), \theta_t - \theta_{\star,t}\rangle + \beta_t^2\|\tilde{F}_t(\theta_t, \xi_t)\|_2^2. \tag{19}$$

We will take expectation w.r.t. to the sample $\xi_t = (s_t, a_t, b_t, s_{t+1})$, conditioned on $\theta_t, x_t, y_t$, and therefore on $\theta_{\star,t}$ and use $s_t \sim \rho^{x_t, y_t}$. We also note (18) to separate the error due to $\hat{V}_{k-1}$ and derive

$$-\beta_t\mathbb{E}_{\xi_t}\langle\tilde{F}_t(\theta_t, \xi_t), \theta_t - \theta_{\star,t}\rangle = -\beta_t\langle F_t(\theta_t), \theta_t - \theta_{\star,t}\rangle - \beta_t\langle\delta_{v,t}, \theta_t - \theta_{\star,t}\rangle$$
$$\leq -\beta_t\langle F_t(\theta_t), \theta_t - \theta_{\star,t}\rangle + \frac{\beta_t\gamma^2}{2\lambda_{\min}^\theta}\|P_{x_t, y_t}(\hat{V}_{k-1} - V_{k-1})\|^2 + \frac{\beta_t\lambda_{\min}^\theta}{2}\|\theta_t - \theta_{\star,t}\|^2,$$

where $P_{x_t, y_t}$ is the matrix denoting the probability matrix multiplying $\hat{V}_{k-1} - V_{k-1}$ in (18) and $\delta_{v,t} = \gamma P_{x_t, y_t}(V_{k-1} - \hat{V}_{k-1})$ and we used Cauchy-Schwarz and Young's inequalities. We can use standard inequalities to estimate $\|P_{x_t, y_t}(\hat{V}_{k-1} - V_{k-1})\|_2^2 \leq |S||A|\|P_{x_t, y_t}(\hat{V}_{k-1} - V_{k-1})\|_\infty^2 \leq |S||A|\|\hat{V}_{k-1} - V_{k-1}\|_\infty^2$ and take $\mathbb{E}_{\xi_t}$ in (19) by using the two estimations above to get

$$\mathbb{E}_{\xi_t}\|\theta_{t+1} - \theta_{\star,t}\|_2^2 = \mathbb{E}_{\xi_t}\|\theta_t - \theta_{\star,t}\|_2^2 - 2\beta_t\langle F_t(\theta_t), \theta_t - \theta_{\star,t}\rangle + \frac{\beta_t\gamma^2|S||A|}{\lambda_{\min}^\theta}\|\hat{V}_{k-1} - V_{k-1}\|_\infty^2$$
$$+ \beta_t\lambda_{\min}^\theta\|\theta_t - \theta_{\star,t}\|_2^2 + \beta_t^2\mathbb{E}_{\xi_t}\|\tilde{F}_t(\theta_t, \xi_t)\|_2^2. \tag{20}$$

For the inner product, we would use $F_t(\theta_{\star,t}) = 0$ and strong monotonicity of $F_t$ (an estimation similar to (Bhandari et al., 2018, Lemma 3)) to get

$$2\beta_t\langle F_t(\theta_t), \theta_t - \theta_{\star,t}\rangle = 2\beta_t\langle F_t(\theta_t) - F_t(\theta_{\star,t}), \theta_t - \theta_{\star,t}\rangle \geq 2\beta_t\lambda_{\min}^\theta\|\theta_t - \theta_{\star,t}\|_2^2. \tag{21}$$

Using this estimate, taking total expectation in (20) and using Young's inequality on the term involving $\|\theta_t - \theta_{\star,t}\|^2$ gives,

$$\mathbb{E}\|\theta_{t+1} - \theta_{\star,t}\|_2^2 \leq (1+\alpha)(1 - \beta_t \lambda_{\min}^\theta)\mathbb{E}\|\theta_t - \theta_{\star,t-1}\|_2^2$$

$$+(1+1/\alpha)(1-\beta_t\lambda_{\min}^\theta)\mathbb{E}\|\theta_{\star,t}-\theta_{\star,t-1}\|_2^2 + \beta_t^2 \mathbb{E}\|\tilde{F}_t(\theta_t,\xi_t)\|_2^2 + \frac{\beta_t\gamma^2|S||A|}{\lambda_{\min}^\theta}\mathbb{E}\|\hat{V}_{k-1} - V_{k-1}\|_\infty^2.$$

Picking $\alpha = \frac{\beta_t\lambda_{\min}^\theta}{2(1-\beta_t\lambda_{\min}^\theta)}$ with ensuring $\beta_t \leq \frac{1}{\lambda_{\min}^\theta}$ due to $\beta_t$ choice, and using $\|\theta_{\star,t} - \theta_{\star,t-1}\|_2 = \|\sum_b(y_t(b|s) - y_{t-1}(b|s))Q(s,a,b)\|_2 \leq \sqrt{|S||A|}\|\sum_b(y_t(b|s) - y_{t-1}(b|s))Q(s,a,b)\|_\infty \leq 2\sqrt{|S||A|}/(1-\gamma)\max_s\|y_t(\cdot|s) - y_{t-1}(\cdot|s)\|_1$ with Lemma B.1 give

$$\mathbb{E}\|\theta_{t+1} - \theta_{\star,t}\|_2^2 \leq \left(1 - \frac{\beta_t\lambda_{\min}^\theta}{2}\right)\mathbb{E}\|\theta_t - \theta_{\star,t-1}\|_2^2 + \beta_t^2\mathbb{E}\|\tilde{F}_t(\theta_t,\xi_t)\|_2^2$$

$$+ \frac{8|S||A|}{(1-\gamma)^2\beta_t\lambda_{\min}^\theta}\mathbb{E}\left(\max_s\|y_t(\cdot|s) - y_{t-1}(\cdot|s)\|_1\right)^2 + \frac{\beta_t\gamma^2|S||A|}{\lambda_{\min}^\theta}\mathbb{E}\|\hat{V}_{k-1} - V_{k-1}\|_\infty^2.$$

We only need to show that $\mathbb{E}\left(\max_s\|y_t(\cdot|s) - y_{t-1}(\cdot|s)\|_1\right)^2$ is small. This is easy since we use small step sizes $\eta_t$ for the policy update. In particular, the update rule of $x_t$ will give by Lemma B.5

$$\|x_{t+1}(\cdot|s) - x_t(\cdot|s)\|_1 \leq \eta_t\|\theta_{t+1}\|_\infty,$$

and by the symmetrical update for $y_t$, it holds that $\|y_{t+1}(\cdot|s) - y_t(\cdot|s)\|_1 \leq \eta_t\|\theta_{t+1}\|_\infty$. This is a deterministic inequality holding for all $s$, so we can take its maximum over $s$, square it, use Lemma B.1 and plug it into the main inequality.

$$\mathbb{E}\|\theta_{t+1} - \theta_{\star,t}\|_2^2 \leq \left(1 - \frac{\beta_t\lambda_{\min}^\theta}{2}\right)\mathbb{E}\|\theta_t - \theta_{\star,t-1}\|_2^2 + \beta_t^2\mathbb{E}\|\tilde{F}_t(\theta_t,\xi_t)\|_2^2$$

$$+ \frac{8|S||A|\eta_t^2\|\theta_{t+1}\|_\infty^2}{\beta_t(1-\gamma)^2\lambda_{\min}^\theta} + \frac{\beta_t\gamma^2|S||A|}{\lambda_{\min}^\theta}\mathbb{E}\|\hat{V}_{k-1} - V_{k-1}\|_\infty^2.$$

We plug in the bounds from Lemma B.1

$$\mathbb{E}\|\theta_{t+1} - \theta_{\star,t}\|^2 \leq \left(1 - \frac{\beta_t\lambda_{\min}^\theta}{2}\right)\mathbb{E}\|\theta_t - \theta_{\star,t-1}\|^2 + \frac{2\beta_t^2}{(1-\gamma)^2} + \frac{8|S||A|\eta_t^2}{\beta_t(1-\gamma)^4\lambda_{\min}^\theta}$$

$$+ \frac{\beta_t\gamma^2|S||A|}{\lambda_{\min}^\theta}\mathbb{E}\|\hat{V}_{k-1} - V_{k-1}\|_\infty^2.$$

Picking $\eta_t = \frac{1}{t^{3/4}}$ and $\beta_t = \frac{1}{\lambda_{\min}^\theta t^{1/2}}$, the recursion will be $u_{t+1} \leq (1 - c_0/\sqrt{t})u_t + \frac{c_1}{t}C_1 + \frac{c_2}{\sqrt{t}}C_2$. Therefore the final bound is

$$\frac{1}{T}\sum_{t=1}^T\mathbb{E}\|\theta_{t+1} - \theta_{\star,t}\|^2 \leq \frac{2}{\sqrt{T}}\|\theta_1 - \theta_{\star,0}\|^2 + \frac{4(1 + \log T)}{(\lambda_{\min}^\theta)^2(1-\gamma)^2\sqrt{T}}$$

$$+ \frac{16|S||A|(1 + \log T)}{\sqrt{T}(1-\gamma)^4} + \frac{4\gamma^2|S||A|}{(\lambda_{\min}^\theta)^2}\mathbb{E}\|\hat{V}_{k-1} - V_{k-1}\|_\infty^2.$$

$\square$

**Lemma E.7.** *Denote* $x_{out} = \frac{1}{T}\sum_{t=1}^T x_t$ *and* $y_{out} = \frac{1}{T}\sum_{t=1}^T y_t$. *Let Assumption 1, 2 hold and* $\eta_t = \frac{1}{t^{3/4}}$. *For the actor computed in stage 1 of Algorithm 5,*

$$\mathbb{E}\left[\max_{s,z=(x,y)} x_{out}^s Q^s y^s - x Q^s y_{out}^s\right] \leq \frac{\log|A||B|}{T^{1/4}} + \frac{4}{(1-\gamma)^2 T^{3/4}} + \frac{2}{T}\sum_{t=1}^T\mathbb{E}\|\theta_{t+1}^x - \theta_{\star,t}^x\|_\infty$$

$$+ \frac{2}{T}\sum_{t=1}^T\mathbb{E}\|\theta_{t+1}^y - \theta_{\star,t}^y\|_\infty. \quad (22)$$

**Corollary E.8.** *By plugging in the bound for $\frac{1}{T}\sum_{t=1}^{T}\mathbb{E}\|\theta_{t+1} - \theta_{\star,t}\|_\infty$ from Lemma E.2 after Jensen's inequality and by noting $\theta_t^x$ and $\theta_t^y$ admit the same bounds, and by extracting only the dependence on $\lambda_{\min}, |S|, |A|, \gamma, T$ gives*

$$\mathbb{E}\left[\max_{s,z=(x,y)} x_{out}^s Q^s y^s - x Q^s y_{out}^s\right] \leq \frac{1}{T^{1/4}}\tilde{\mathcal{O}}\left(\frac{\sqrt{|S|(|A| \vee |B|)}}{(1-\gamma)^2}\right) + \frac{1}{T^{1/4}}\tilde{\mathcal{O}}\left(\frac{1}{\lambda_{\min}(1-\gamma)}\right)$$

$$+ \sqrt{\mathbb{E}\|\hat{V}_{k-1} - V_{k-1}\|_\infty^2}\,\mathcal{O}\left(\frac{\gamma\sqrt{|S|(|A| \vee |B|)}}{\lambda_{\min}^\theta}\right). \quad (23)$$

**Remark E.9.** *We make sure $\mathbb{E}\|\hat{V}_{k-1} - V_{k-1}\|_\infty$ is small by the estimation of $\hat{V}_{k-1}$ in stage 2, in Lemma E.14, Lemma E.11.*

*Proof.* Recall the notation $x^s Q^s y^s = \sum_{a,b} x(a|s)y(b|s)Q(s,a,b)$. First by definition of $x_{\text{out}}$ and the standard arrangement for duality gap, it holds for all $s$

$$x_{out}^s Q^s y^s - x^s Q^s y_{out}^s = \frac{1}{T}\sum_{t=1}^{T}\langle \mathbb{E}_{a\sim x_t(\cdot|s)}Q(s,a,\cdot), y(\cdot|s)\rangle - \frac{1}{T}\sum_{t=1}^{T}\langle \mathbb{E}_{b\sim y_t(\cdot|s)}Q(s,\cdot,b), x(\cdot|s)\rangle$$

$$= \frac{1}{T}\sum_{t=1}^{T}\left[\langle \mathbb{E}_{a\sim x_t(\cdot|s)}Q(s,a,\cdot), y(\cdot|s) - y_t(\cdot|s)\rangle - \langle \mathbb{E}_{b\sim y_t(\cdot|s)}Q(s,\cdot,b), x(\cdot|s) - x_t(\cdot|s)\rangle\right]. \quad (24)$$

We are going to bound the inner products in RHS. From the update rule of $x_{t+1}$, for all $s, x(\cdot|s) \in \Delta$, it holds that (see (Tseng, 2008, Property 1))

$$\langle \nabla D(x_{t+1}(\cdot|s), x_t(\cdot|s)) + \eta_t \theta_{t+1}(s,\cdot), x(\cdot|s) - x_{t+1}(\cdot|s)\rangle \geq 0.$$

By three point identity and using the notation

$$D(x(\cdot|s), x_{t+1}(\cdot|s)) \leq D(x(\cdot|s), x_t(\cdot|s)) + \eta_t\langle\theta_{\star,t}(s,\cdot), x(\cdot|s) - x_{t+1}(\cdot|s)\rangle$$
$$+ \eta_t\langle\theta_{t+1}(s,\cdot) - \theta_{\star,t}(s,\cdot), x(\cdot|s) - x_{t+1}(\cdot|s)\rangle - D(x_{t+1}(\cdot|s), x_t(\cdot|s)). \quad (25)$$

We bound the inner products using Cauchy-Schwarz and Young's inequalities and Equation (8), since $x(\cdot|s) \in \Delta$),

$$\eta_t\langle\theta_{t+1}(s,\cdot) - \theta_{\star,t}(s,\cdot), x(\cdot|s) - x_{t+1}(\cdot|s)\rangle \leq 2\eta_t\|\theta_{t+1} - \theta_{\star,t}\|_\infty,$$
$$\eta_t\langle\theta_{\star,t}(s,\cdot), x(\cdot|s) - x_{t+1}(\cdot|s)\rangle = \eta_t\langle\theta_{\star,t}(s,\cdot), x(\cdot|s) - x_t(\cdot|s)\rangle + \eta_t\langle\theta_{\star,t}(s,\cdot), x_t(\cdot|s) - x_{t+1}(\cdot|s)\rangle$$
$$\leq \eta_t\langle\theta_{\star,t}(s,\cdot), x(\cdot|s) - x_t(\cdot|s)\rangle + \frac{\eta_t^2}{2}\|\theta_{\star,t}\|_\infty^2 + D(x_{t+1}(\cdot|s), x_t(\cdot|s))$$

Using these estimations in (25) gives with $\theta_{\star,t}(s,\cdot) = \mathbb{E}_{b\sim y_t(\cdot|s)}Q(s,\cdot,b)$ gives

$$\langle \mathbb{E}_{b\sim y_t(\cdot|s)}Q(s,\cdot,b), x_t(\cdot|s) - x(\cdot|s)\rangle + \frac{1}{\eta_t}D(x(\cdot|s), x_{t+1}(\cdot|s)) \leq \frac{1}{\eta_t}D(x(\cdot|s), x_t(\cdot|s))$$
$$+ 2\|\theta_{t+1} - \theta_{\star,t}\|_\infty + \frac{\eta_t}{2}\|\theta_{\star,t}\|_\infty^2.$$

We sum the inequality, use Lemma B.1 and $\max_{x,x_t} D(x(\cdot|s), x_t(\cdot|s)) \leq \log|A|$.

$$\frac{1}{T}\sum_{t=1}^{T}\langle \mathbb{E}_{b\sim y_t(\cdot|s)}Q(s,\cdot,b), x_t(\cdot|s) - x(\cdot|s)\rangle \leq \frac{1}{\eta_1 T}D(x(\cdot|s), x_1(\cdot|s)) + \frac{\log|A|}{T^{1/4}}$$
$$+ \frac{1}{T}\sum_{t=1}^{T}2\|\theta_{t+1} - \theta_{\star,t}\|_\infty + \frac{\sum_{t=1}^{T}\eta_t}{2(1-\gamma)^2 T}.$$

We use the same estimation for the other player, since it is symmetric, to bound the RHS of (24). Then, we take maximum of both sides w.r.t. $s, z$, take expectation and bring back superscripts of $x, y$ to $\theta_t$ since we will have error from both players. $\qquad\square$

### E.2 PROOFS FOR STAGE 2 OF SINGLE LOOP NAC WITH ETIQUETTE

**Remark E.10.** *Stage 2 is asymmetric for both players. As we are computing best response to $x_k$, $x$ player's policy remains fixed in this phase, it only computes $\hat{V}_{t-1}$ to be used in its next stage 1 step.*

First, we are going to show that while running this step, $y$-player can construct its stochastic oracle without access to policy or actions of $x$-player. Then, as the best response problem is essentially a single agent problem where the other player is part of the environment, our proofs are similar to the results for single agent setting (Khodadadian et al., 2021b; Hong et al., 2020). Let us denote the approximate best response as $\bar{y}_{\hat{t}}$. Main goal in this step is that we have to characterize explicitly the error $\|\hat{V}_k - V^{x_k,\bar{y}_{\hat{t}}}\|$ as it is used in the stage 2.

Note that this is generally not done in single agent setting as the goal is to compute a policy. However, here our main goal is to have access to an oracle approximation $V_k = V^{x_k,\bar{y}_{\hat{t}}}$, rather than the output policy $\bar{y}_{\hat{t}}$, therefore, we keep track of $\omega_t$ that tracks this value function with an explicit error estimate (see Lemma E.11).

#### E.2.1 FORMULATION

**Notation.** Here, the problem is to compute best response where the other player fixes its strategy. Let us fix $x_k$ and denote the best response as $y_k^*$. Here, since $x_k$ is fixed, it is a part of the environment for $y$-player and single agent MDP analyses will go through. We only need to be careful to make sure the "gradient" for $\bar{y}_t$ updates can be calculated by not knowing policy or actions of $x_k$.

For NPG updates, we will need to compute at iteration $t$, $\nu_{\star,t}(s,b) = \mathbb{E}_{a \sim x_k(\cdot|s)} Q^{x_k,\bar{y}_t}(s,a,b)$. Writing the Bellman equation and using the definition of value functions

$$Q^{x_k,\bar{y}_t}(s,a,b) = r(s,a,b) + \gamma \sum_{s'} P(s'|s,a,b) V^{x_k,\bar{y}_t}(s')$$

$$= r(s,a,b) + \gamma \sum_{s'} P(s'|s,a,b) \sum_{a',b'} x_k(a'|s') \bar{y}_t(b'|s') Q^{x_k,\bar{y}_t}(s',a',b')$$

We note $\nu_{\star,t}(s',b') = \mathbb{E}_{a' \sim x_k(\cdot|s')} Q^{x_k,\bar{y}_t}(s',a',b')$ and take expectation of previous equality with $a \sim x_k(\cdot|s)$,

$$\nu_{\star,t}(s,b) = \sum_a x_k(a|s) r(s,a,b) + \gamma \sum_{s',a,b'} P(s'|s,a,b) x_k(a|s) \bar{y}_t(b'|s') \nu_{\star,t}(s',b')$$

We use the sampling matrix (as (Lan, 2021, Sec. 5.2)) $\mathrm{diag}(\rho^{x_k,\bar{y}_t}) \otimes \mathrm{diag}(\bar{y}_t)$ and define the operator

$$F_t^\nu(\nu_t)(s,b) = \rho^{x_k,\bar{y}_t}(s)\bar{y}_t(b|s)\Big[\nu_t(s,b) - \sum_a x_k(a|s) r(s,a,b)$$
$$- \gamma \sum_{s',a,b'} x_k(a|s) P(s'|s,a,b) \bar{y}_t(b'|s') \nu_t(s',b')\Big],$$

such that $F_t^\nu(\nu_{\star,t}) = 0$. Strong monotonicity of $F_t^\nu$ with constant $\lambda_{\min}^\nu$ follows from Assumption 1, 2, and that the operator $T_{x_k}\nu(s,b) = \sum_a x_k(a|s) r(s,a,b) + \gamma \sum_{s',a,b'} x_k(a|s) P(s'|s,a,b) \bar{y}_t(b'|s') \nu(b',s')$ is $\gamma$ contraction in $\ell_\infty$ norm, (Zhao et al., 2021, Lemma 1) (Bauschke et al., 2011, Example 22.6 and 20.7). We define the stochastic operator after sampling $\xi_t = (s_t, a_t, b_t, s_{t+1}, b_{t+1})$ with $s_t \sim \rho^{x_k,\bar{y}_t}$, $a_t \sim x_k(\cdot|s_t)$, $b_t \sim \bar{y}_t(\cdot|s_t)$, $s_{t+1} \sim P(\cdot|s_t, a_t, b_t)$, $b_{t+1} \sim \bar{y}_t(\cdot|s_{t+1})$,

$$\tilde{F}_t^\nu(\nu_t, \xi_t) = e(s_t, b_t)\left(\nu_t(s_t, b_t) - r(s_t, a_t, b_t) - \gamma \nu_t(s_{t+1}, b_{t+1})\right),$$

and as we assume we can sample $s_t \sim \rho^{x_k,\bar{y}_t}$, $\mathbb{E}_{\xi_t}[\tilde{F}_t^\nu(\nu_t, \xi_t)] = F_t^\nu(\nu_t)$. In particular, we see that as long as $s_t, a_t, b_t, s_{t+1}, b_{t+1}$ are estimated in the prescribed way, there is no need for $\bar{y}_t$ update to see the actions or policy of $x_k$ for $\tilde{F}_t^\nu(\nu_t, \xi_t)$ to be unbiased estimate of $F_t^\nu(\nu_t)$. It only needs to see

its own actions $b_t, b_{t+1}, r(s_t, a_t, b_t)$ and $s_{t+1}$.

$$
\mathbb{E}_{\xi_t} F_t^\nu(\nu_t, \xi_t)
$$
$$
= \sum_{s,a,b,s',b'} \Pr(s_t = s, a_t = a, b_t = b, s_{t+1} = s', b_{t+1} = b') e(s,b) \left[ \nu_t(s,b) - r(s,a,b) - \gamma \nu_t(s',b') \right]
$$
$$
= \sum_{s,a,b,s',b'} \rho^{x_k,\bar{y}_t}(s) x_k(a|s) \bar{y}_t(b|s) P(s'|s,a,b) \bar{y}_t(b'|s') e(s,b) \left[ \nu_t(s,b) - r(s,a,b) - \gamma \nu_t(s',b') \right]
$$
$$
= \sum_{s,b} \rho^{x_k,\bar{y}_t}(s) \bar{y}_t(b|s) e(s,b) \Big[ \nu_t(s,b) - \sum_a x_k(a|s) r(s,a,b)
$$
$$
- \gamma \sum_{s',a,b'} x_k(a|s) P(s'|s,a,b) \bar{y}_t(b'|s') \nu_t(s',b') \Big]. \quad (26)
$$

The same estimations as Lemma E.2, without the bias from $\hat{V}_{k-1}$, as we have unbiased samples will give Lemma E.11.

Let us define the corresponding operator for learning state-value function

$$
V^{x_k,\bar{y}_t}(s) = \sum_{a,b} x_k(a|s) \bar{y}_t(b|s) r(s,a,b) + \sum_{s',a,b} x_k(a|s) \bar{y}_t(b|s) P(s'|s,a,b) V^{x_k,\bar{y}_t}(s').
$$

Similar to the $Q$ function, we can define $\omega_{\star,t} = V^{x_k,\bar{y}_t}$ and the operator

$$
F_t^\omega(\omega_t)(s) = \rho^{x_k,\bar{y}_t}(s) \Big( \omega_t(s) - \sum_{a,b} x_k(a|s) \bar{y}_t(b|s) r(s,a,b)
$$
$$
- \gamma \sum_{s',a,b} x_k(a|s) \bar{y}_t(b|s) P(s'|s,a,b) \omega_t(s') \Big).
$$

By Assumption 1, 2, this operator is strongly monotone with $\lambda_{\min}^\omega$, the justification of which is the same as the $F_t^\nu$ operator defined above. We also note that $F_t^\omega(\omega_{\star,t}) = 0$. The corresponding stochastic operator is defined as

$$
\tilde{F}_t^\omega(\omega_t, \xi_t) = e(s_t) \left( \omega_t(s_t) - r(s_t, a_t, b_t) - \gamma \omega_t(s_{t+1}) \right),
$$

where $s_t \sim \rho^{x_k,\bar{y}_t}$, $a_t \sim x_k(\cdot|s)$, $b_t \sim \bar{y}_t(\cdot|s)$, $s_{t+1} \sim P(\cdot|s,a,b)$ and as we assume we can sample $s_t \sim \rho^{x_k,\bar{y}_t}$,

$$
\mathbb{E}_{\xi_t}[\tilde{F}_t^\omega(\omega_t, \xi_t)] = F_t^\omega(\omega_t).
$$

### E.2.2 THEORETICAL RESULTS

First, we characterize the critic of stage 2 in Algorithm 5, denoted by $\nu_t, \omega_t$ for action value function and state value function, respectively.

**Lemma E.11.** *Let Assumption 1, 2 hold. Let $\beta_t^\nu = \frac{1}{\lambda_{\min}^\nu t^{1/2}}$, $\beta_t^\omega = \frac{1}{\lambda_{\min}^\omega t^{1/2}}$ and $\eta_t = \frac{1}{t^{3/4}}$. Then for the critic computed by stage 2 of Algorithm 5,*

$$
\frac{1}{T} \sum_{t=1}^T \mathbb{E}\|\nu_{t+1} - \nu_{\star,t}\|^2 = \frac{1}{\sqrt{T}} \left\{ \mathcal{O}\left( \frac{|S||B|}{(1-\gamma)^2} \right) + \tilde{\mathcal{O}}\left( \frac{1}{(\lambda_{\min}^\nu)^2 (1-\gamma)^2} \right) + \tilde{\mathcal{O}}\left( \frac{|S||B|}{(1-\gamma)^6} \right) \right\}.
$$

$$
\frac{1}{T} \sum_{t=1}^T \mathbb{E}\|\omega_{t+1} - \omega_{\star,t}\|^2 = \frac{1}{\sqrt{T}} \left\{ \mathcal{O}\left( \frac{|S|}{(1-\gamma)^2} \right) + \tilde{\mathcal{O}}\left( \frac{1}{(\lambda_{\min}^\omega)^2 (1-\gamma)^2} \right) + \tilde{\mathcal{O}}\left( \frac{|S|}{(1-\gamma)^6} \right) \right\}.
$$

**Remark E.12.** *Using the same samples for $\theta_t$ and $\hat{V}_t$ does not seem to cause a problem since the analysis in Lemma E.2 only takes conditional expectations conditioning on $\theta_t$ and uses that $\max_s \|x_t(\cdot|s) - x_{t+1}(\cdot|s)\|^2$ is small directly by small step sizes $\eta$.*

**Remark E.13.** *$\hat{V}_t$ is not used in the stage 2, but it is estimated to be used in the stage 1 and also to make the bound of Lemma E.7 small since the bound implies $\mathbb{E}\|\omega_{\hat{t}+1} - V^{x_k,y_{\hat{t}}}\|^2 = \mathbb{E}\|\hat{V}_{k-1} - V_{k-1}\|^2 \leq \tilde{\mathcal{O}}\left( 1/T^{1/2} \right)$.*

*Proof.* Let us recall $\nu_{\star,t}(s,b) = \mathbb{E}_{a \sim x_k(\cdot|s)} Q^{x_k,\bar{y}_t}(s,a,b)$ and $\omega_{\star,t} = V^{x_k,\bar{y}_t}$.

We expand the squared norm

$$\|\nu_{t+1} - \nu_{\star,t}\|_2^2 = \|\nu_t - \nu_{\star,t}\|_2^2 - 2\beta_t \langle \tilde{F}_t^\nu(\nu_t, \xi_t), \nu_t - \nu_{\star,t} \rangle + \beta_t^2 \|\tilde{F}_t^\nu(\nu_t, \xi_t)\|_2^2.$$

We take expectation w.r.t. the randomness of $\xi_t = (s_t, a_t, b_t, s_{t+1}, b_{t+1})$ and use from (26) that $\mathbb{E}_{\xi_t}[\tilde{F}_t^\nu(\nu_t, \xi_t)] = F_t^\nu(\nu_t)$.

$$\mathbb{E}_{\xi_t}\|\nu_{t+1} - \nu_{\star,t}\|_2^2 = \|\nu_t - \nu_{\star,t}\|_2^2 - 2\beta_t \langle F_t^\nu(\nu_t), \nu_t - \nu_{\star,t} \rangle + \beta_t^2 \mathbb{E}_{\xi_t}\|\tilde{F}_t^\nu(\nu_t, \xi_t)\|_2^2$$

By strong monotonicity and $F_t^\nu(\nu_{\star,t}) = 0$, it follows that

$$2\beta_t \langle F_t^\nu(\nu_t), \nu_t - \nu_{\star,t} \rangle = 2\beta_t \langle F_t^\nu(\nu_t) - F_t^\nu(\nu_{\star,t}), \nu_t - \nu_{\star,t} \rangle$$
$$\geq 2\beta_t \lambda_{\min}^\nu \|\nu_t - \nu_{\star,t}\|_2^2.$$

We use this estimation and then Young's inequality to obtain

$$\mathbb{E}_{\xi_t}\|\nu_{t+1} - \nu_{\star,t}\|_2^2 \leq (1 - 2\beta_t \lambda_{\min}^\nu)\|\nu_t - \nu_{\star,t}\|_2^2 + \beta_t^2 \mathbb{E}_{\xi_t}\|\tilde{F}_t^\nu(\nu_t, \xi_t)\|_2^2$$

$$\leq (1 - \beta_t \lambda_{\min}^\nu)\|\nu_t - \nu_{\star,t-1}\|_2^2 + \frac{2}{\beta_t \lambda_{\min}^\nu}\|\nu_{\star,t} - \nu_{\star,t-1}\|_2^2 + \beta_t^2 \mathbb{E}_{\xi_t}\|\tilde{F}_t^\nu(\nu_t, \xi_t)\|_2^2 \quad (27)$$

We now have to bound the second term on RHS. For this, we will use Lemma B.4, but first we have to transform the term into the form of Lemma B.4. We recall $\nu_{\star,t}(s,b) = \mathbb{E}_{a \sim x_k(\cdot|s)} Q^{x_k,\bar{y}_t}(s,a,b)$

$$\|\nu_{\star,t} - \nu_{\star,t-1}\|_2 \leq \sqrt{|S||B|}\|\nu_{\star,t} - \nu_{\star,t-1}\|_\infty$$
$$= \sqrt{|S||B|} \max_{s,b} \left| \mathbb{E}_{a \sim x_k(\cdot|s)} \left( Q^{x_k,\bar{y}_t}(s,a,b) - Q^{x_k,\bar{y}_{t-1}}(s,a,b) \right) \right|$$
$$\leq \sqrt{|S||B|} \max_{s,b} \mathbb{E}_{a \sim x_k(\cdot|s)} \left| Q^{x_k,\bar{y}_t}(s,a,b) - Q^{x_k,\bar{y}_{t-1}}(s,a,b) \right|$$
$$\leq \sqrt{|S||B|}\|Q^{x_k,\bar{y}_t} - Q^{x_k,\bar{y}_{t-1}}\|_\infty$$
$$\leq \frac{2\gamma\sqrt{|S||B|}}{(1-\gamma)^2} \max_s \|\bar{y}_t(\cdot|s) - \bar{y}_{t-1}(\cdot|s)\|_1$$

where the second inequality is by Jensen and the last inequality is by Lemma B.4.

As used in the proof of Lemma E.2, update rule of $\bar{y}_t(\cdot|s)$ gives $\|\bar{y}_t(\cdot|s) - \bar{y}_{t-1}(\cdot|s)\|_1 \leq \eta_t \|\nu_{t+1}\|_\infty$ by Lemma B.5. Using these estimates in (27) after taking total expectation gives

$$\mathbb{E}\|\nu_{t+1} - \nu_{\star,t}\|_2^2 \leq (1 - \beta_t \lambda_{\min}^\nu)\mathbb{E}\|\nu_t - \nu_{\star,t-1}\|_2^2 + \frac{8|S||B|\gamma^2 \eta_t^2 \mathbb{E}\|\nu_{t+1}\|_\infty^2}{\beta_t \lambda_{\min}^\nu (1-\gamma)^4}$$
$$+ \beta_t^2 \mathbb{E}\|\tilde{F}_t^\nu(\nu_t, \xi_t)\|_2^2.$$

We get the result by using the same argument as the end of the proof of Lemma E.2, by also using Lemma B.1 to bound $\|\tilde{F}_t^\nu(\nu_t, \xi_t)\|_2^2$ and $\|\nu_{t+1}\|_\infty^2$.

The proof of the second inequality is exactly the same except that in (27), instead of $\|\nu_{\star,t} - \nu_{\star,t-1}\|$ we will have $\|\omega_{\star,t} - \omega_{\star,t-1}\|$ and we will therefore use Lemma B.3. $\qquad\square$

Next, we will upper bound $V^{x_k,\bar{y}_t} - V^{x_k,y_k^*}$ which is a single agent problem as $x_k$ is fixed and we showed in this section how to do the policy evaluation without knowing the actions of $x_k$. The next lemma will be proven similar to single agent settings (Hong et al., 2020; Khodadadian et al., 2021b).

**Lemma E.14.** *Let Assumption 1, 2 hold. Let $\beta_t^\nu = \frac{1}{\lambda_{\min}^\nu t^{1/2}}$, $\beta_t^\omega = \frac{1}{\lambda_{\min}^\omega t^{1/2}}$ and $\eta_t = \frac{1}{t^{3/4}}$. Then for stage 2 of Algorithm 5,*

$$\frac{1}{T}\sum_{t=1}^T \mathbb{E}\mathbb{E}_{s_0 \sim \sigma}\left( V^{x_k,y_k^*}(s_0) - V^{x_k,y_t}(s_0) \right) \leq \frac{\log|B|}{(1-\gamma)T^{1/4}} + \frac{2}{(1-\gamma)^3 T^{3/4}}$$

$$+ \frac{2}{T(1-\gamma)}\sum_{t=1}^T \mathbb{E}\|\nu_{t+1} - \nu_{\star,t}\|_\infty$$

$$\leq \frac{1}{T^{1/4}}\left\{ \mathcal{O}\left( \frac{\sqrt{|S||B|}}{(1-\gamma)^2} \right) + \tilde{\mathcal{O}}\left( \frac{\sqrt{1}}{\lambda_{\min}^\nu (1-\gamma)^2} \right) + \tilde{\mathcal{O}}\left( \frac{\sqrt{|S||B|}}{(1-\gamma)^4} \right) \right\},$$

*where the second inequality follows by using the results in Lemma E.11.*

*Proof.* By the update rule of $\bar{y}_{t+1}$ for all $s$, $y(\cdot|s) \in \Delta$, (see (Tseng, 2008, Property 1))

$$
\begin{aligned}
D(y(\cdot|s), \bar{y}_{t+1}(\cdot|s)) &\leq D(y(\cdot|s), \bar{y}_t(\cdot|s)) - \eta_t \langle \nu_{t+1}(s, \cdot), y(\cdot|s) - \bar{y}_{t+1}(\cdot|s) \rangle - D(\bar{y}_{t+1}(\cdot|s), \bar{y}_t(\cdot|s)) \\
&= D(y(\cdot|s), \bar{y}_t(\cdot|s)) - \eta_t \langle \nu_{\star,t}(s, \cdot), y(\cdot|s) - \bar{y}_{t+1}(\cdot|s) \rangle \\
&\quad - \eta_t \langle \nu_{t+1}(s, \cdot) - \nu_{\star,t}(s, \cdot), y(\cdot|s) - \bar{y}_{t+1}(\cdot|s) \rangle - D(\bar{y}_{t+1}(\cdot|s), \bar{y}_t(\cdot|s))
\end{aligned}
$$

We estimate the inner products by Cauchy-Schwarz, Young's inequalities and Equation (8)

$$
\eta_t \langle \nu_{t+1}(s, \cdot) - \nu_{\star,t}(s, \cdot), y(\cdot|s) - y_{t+1}(\cdot|s) \rangle \leq 2\eta_t \|\nu_{t+1} - \nu_{\star,t}\|_\infty,
$$

and

$$
\begin{aligned}
-\eta_t \langle \nu_{\star,t}(s, \cdot), y(\cdot|s) - \bar{y}_{t+1}(\cdot|s) \rangle &= -\eta_t \langle \nu_{\star,t}(s, \cdot), y(\cdot|s) - \bar{y}_t(\cdot|s) \rangle - \eta_t \langle \nu_{\star,t}(s, \cdot), \bar{y}_t(\cdot|s) - \bar{y}_{t+1}(\cdot|s) \rangle \\
&\leq -\eta \langle \nu_{\star,t}(s, \cdot), y(\cdot|s) - \bar{y}_t(\cdot|s) \rangle + \frac{\eta_t^2 \|\nu_{\star,t}\|_\infty^2}{2} + D(\bar{y}_{t+1}(\cdot|s), \bar{y}_t(\cdot|s)).
\end{aligned}
$$

Consequently, by using the definition of $\nu_{\star,t}(s, b) = \mathbb{E}_{a \sim x_k(\cdot|s)} Q^{x_k, \bar{y}_t}(s, a, b)$

$$
\langle \mathbb{E}_{a \sim x_k(\cdot|s)} Q^{x_k, \bar{y}_t}(s, a, b), y(\cdot|s) - \bar{y}_t(\cdot|s) \rangle + \frac{1}{\eta_t} D(y(\cdot|s), \bar{y}_{t+1}(\cdot|s)) \leq \frac{1}{\eta_t} D(y(\cdot|s), \bar{y}_t(\cdot|s))
$$

$$
+ 2\|\nu_{t+1} - \nu_{\star,t}\|_\infty + \frac{\eta_t \|\nu_{\star,t}\|_\infty^2}{2}. \quad (28)
$$

We sum the inequality and use $\max_{y_1, y_2} D(y_1(\cdot|s), y_2(\cdot|s)) \leq \log|B|$ with KL divergence, to get

$$
\frac{1}{T} \sum_{t=1}^{T} \langle \mathbb{E}_{a \sim x_k(\cdot|s)} Q^{x_k, \bar{y}_t}(s, a, b), y(\cdot|s) - y_t(\cdot|s) \rangle \leq \frac{1}{\eta_1 T} D(y(\cdot|s), \bar{y}_1(\cdot|s)) + \frac{1}{T^{1/4}} \log|B|
$$

$$
+ \frac{2}{T} \sum_{t=1}^{T} \|\nu_{t+1} - \nu_{\star,t}\|_\infty + \frac{\sum_{t=1}^{T} \eta_t}{2T(1-\gamma)^2}. \quad (29)
$$

Let us recall that $y_k^*$ is a best response policy. By the performance difference lemma

$$
V^{x_k, y_k^*}(s_0) - V^{x_k, \bar{y}_t}(s_0) = \frac{1}{1-\gamma} \mathbb{E}_{s \sim d_{s_0}^{x_k, y_k^*}} \langle \mathbb{E}_{a \sim x_k(\cdot|s)} Q^{x_k, \bar{y}_t}(s, a, \cdot), y_k^*(\cdot|s) - \bar{y}_t(\cdot|s) \rangle.
$$

As $x_k$ and $y_k^*$ are independent of $t$ and fixed throughout the loop, in (29) we plug in $y = y_k^*$ and take $\mathbb{E}_{s \sim d_{s_0}^{x_k, y_k^*}}$. $\qquad \square$

---

**Algorithm 6** Reflected NAC with a game etiquette. (See Algorithm 1)

---

**Require:** Subroutine `Policy-Eval` (see Algorithm 2, Algorithm 3, Algorithm 4 ). Initial policies $x_0, y_0, \bar{y}_0$

   **for** $k = 0, 1, \dots$ **do**

      **Stage 1**

      **for** $t = 0, 1, \dots, T - 1$ **do**

         $[\hat{V}_{k-1}^x, \hat{V}_{k-1}^y] = [\texttt{Policy-Eval}(x_{k-1}, y_{k-1}, N, \beta_n^\omega), \texttt{Policy-Eval}(x_{k-1}, y_{k-1}, N, \beta_n^\omega)]$

         $[\theta_{t+1}^x, \theta_{t+1}^y] = [\texttt{Policy-Eval}(x_t, y_t, N, \hat{V}_{k-1}^x, \beta_n^\theta), \texttt{Policy-Eval}(x_t, y_t, N, \hat{V}_{k-1}^y, \beta_n^\theta)]$

         $x_{t+1}(\cdot|s) = \mathcal{P}(x_t(\cdot|s), \eta \left( 2\theta_{t+1}^x(s, \cdot) - \theta_t^x(s, \cdot) \right))$

         $y_{t+1}(\cdot|s) = \mathcal{P}(y_t(\cdot|s), -\eta \left( 2\theta_{t+1}^y(s, \cdot) - \theta_t^y(s, \cdot) \right))$

      Output $x_k = \frac{1}{T} \sum_{t=1}^T x_t$.

      **Stage 2**

      **for** $t = 0, 1, \dots, T - 1$ **do**

         $\nu_{t+1} = \texttt{Policy-Eval}(x_k, \bar{y}_t, N, \beta = \beta_n^\nu)$

         $\bar{y}_{t+1}(\cdot|s) = \mathcal{P}(\bar{y}_t(\cdot|s), -\eta \nu_{t+1}(s, \cdot))$

      Output $y_k = \bar{y}_{\hat{t}}$, where $\hat{t} \in [T]$ is selected uniformly at random.

---

**Algorithm 7** `Policy-Eval` (See Algorithm 2, Algorithm 3, Algorithm 4)

---

**Require:** Policy pair $x, y$, iteration counter $N$, oracle $\hat{V}_{k-1}$, step size $\beta$

   **for** $n = 0, 1, \dots, N - 1$ **do**

      Sample $s_n \sim \rho^{x,y}(\cdot)$, $a_n \sim x(\cdot|s_n)$, $b_n \sim y(\cdot|s_n)$, $s_{n+1} \sim P(\cdot|s_n, a_n, b_n)$.

      **if** $\beta = \beta_n^\omega$ **then**

         $\tilde{F}(\phi_n, \xi_n) = e(s_n)\left(\phi_n(s_n) - r(s_n, a_n, b_n) - \gamma \phi_n(s_{n+1})\right)$

      **else if** $\beta = \beta_n^\theta$ **then**

         $\tilde{F}(\phi_n, \xi_n) = e(s_n, b_n)\left(\phi_n(s_n, b_n) - r(s_n, a_n, b_n) - \gamma \hat{V}_{k-1}(s_{n+1})\right)$

      **else if** $\beta = \beta_n^\nu$ **then**

         Sample also $b_{n+1} \sim y(\cdot|s_{n+1})$.

         $\tilde{F}(\phi_n, \xi_n) = e(s_n, b_n)\left(\phi_n(s_n, b_n) - r(s_n, a_n, b_n) - \gamma \phi_n(s_{n+1}, b_{n+1})\right)$

      $\phi_{n+1} = \phi_n - \beta_n \tilde{F}(\phi_n, \xi_n)$

**Output:** $\phi_N$

---

# F   Proofs for Reflected NAC with a game etiquette

## F.1   Proofs for stage 1 of Reflected NAC with a game etiquette

### F.1.1   Formulation

For single loop actor-critic in the previous section, it was acceptable to do rough analysis since we used small step sizes. With inner-outer structure, as Lan (2021), we can do tighter analysis with constant step sizes for the outer loops (updates of $x, y$). Therefore, we can no longer use GDA that we used for single loop NAC, and the techniques from Lan (2021) are not sufficient as the algorithm therein would correspond to GDA in min-max setting. We will have to use a convergent algorithm for the matrix game solver, such as Mirror Prox (Nemirovski, 2004; Korpelevich, 1976) or FoRB ((Malitsky & Tam, 2020)) and extend the ideas from Lan (2021) to these more advanced algorithms to characterize the bias and variance separately. Also, we would have to do a tighter analysis for $\hat{V}_k$ estimation.

To obtain the desired oracle $\theta_{\star,t}(s,b) = \mathbb{E}_{a \sim x_t(\cdot|s)} Q(s,a,b)$, as in (17),

$$F_t(\theta)(s,b) = \rho^{x_t,y_t}(s) y_t(b|s) \Big[ \theta(s,b) - \sum_a x_t(a|s) r(s,a,b)$$
$$- \gamma \sum_{s',a} x_t(a|s) P(s'|s,a,b) V_{k-1}(s') \Big]. \quad (30)$$

We recall that $F_t$ is strongly monotone with $\lambda^\theta_{\min}$ under Assumption 1, 2. Moreover $F_t$ is Lipschitz with $\lambda_{\max}$. We refer to Appendix E.1.1 for how the oracles in the algorithm can be computed without accessing to other agent's policy or actions. Moreover, we do not put subscripts $\theta^x, \theta^y$ as the estimations will be symmetric again.

### F.1.2 THEORETICAL RESULTS

**Theorem 3.2.** *Let Assumption 1, 2 hold. For Algorithm 6, for the output of $x$-player*

$$\mathbb{E}\mathbb{E}_{s_0 \sim \mu} [\max_y V^{x_k,y}(s_0) - V^\star(s_0)] \leq \frac{C_{\mu,\sigma} k}{(1-\gamma)} \tilde{\mathcal{O}} \Big\{ \frac{1}{T(1-\gamma)^2} + \frac{|S|(|A| \vee |B|)}{\lambda^\theta_{\min}(1-\gamma)^2 N}$$
$$+ \frac{|S|^2(|A|^2 \vee |B|^2)}{(\lambda^\theta_{\min})^2 N^2 (1-\gamma)^2} + \frac{|S||A|}{(\lambda^\theta_{\min})^2 (\lambda^\omega_{\min})^2 (1-\gamma)^2 N}$$
$$\frac{|S||B|}{(1-\gamma)^4 N^2} + \frac{1}{N(1-\gamma)^4 (\lambda^\nu_{\min})^2} + \frac{\sqrt{|S||B|}}{(1-\gamma)^2 N} \Big\} + \mathcal{O} \Big( \frac{C_{\mu,\sigma} \gamma^k}{(1-\gamma)} \Big),$$

*which gives* $\tilde{\mathcal{O}} \big( \frac{C^2_{\mu,\sigma} |S|(|A| \vee |B|)}{\epsilon^2 (1-\gamma)^8 (\lambda^\theta_{\min} \lambda^\omega_{\min} \lambda^\nu_{\min})^2} \big)$ *sample complexity.*

We used Remark C.1 to bound $C_{\mu,\sigma}$ for the result in the main text.

Our theoretical results here bring together ideas from single agent NPG analysis of Lan (2021) and stochastic primal-dual optimization techniques from Malitsky & Tam (2020); Nemirovski et al. (2009). In particular, we will be using ideas from Malitsky & Tam (2020); Nemirovski et al. (2009) in the analysis we develop for extending ideas of Lan (2021) to the stage 1.

We first analyze the policy evaluation routine in Algorithm 6. In particular, we will bound the variance and bias of $\theta_{t+1}$ as an estimate of $\theta_{\star,t}(s,b) = \mathbb{E}_{a \sim x_t(\cdot|s)} Q_{k-1}(s,a,b)$. As this routine is in an inner loop (indexed by $n$), the policies we sample, consequently $F_t$ is fixed, therefore we drop the subscript. The proofs of these lemmas will be similar to Lan (2021), except the additional bias we have due to $\hat{V}_{k-1}$.

**Lemma 3.6.** *Let Assumption 1, 2 hold. Let $\beta^\theta_n = \frac{2}{\lambda^\theta_{\min}(n+n_0)}$ for $n_0 \geq 1$. Then, for Algorithms 6 and 7,*

$$\mathbb{E}\|\theta_N - \theta_{\star,t}\|_2^2 \leq \mathcal{O} \Big( \frac{|S||A|}{(1-\gamma)^2 N^2} + \frac{1}{N(\lambda^\theta_{\min})^2 (1-\gamma)^2} + \frac{|S||A|}{(\lambda^\theta_{\min})^2} \mathbb{E}\|\hat{V}_{k-1} - V_{k-1}\|_\infty^2 \Big).$$

*Proof.* Throughout this proof, $\mathbb{E}[\cdot]$ will stand for conditional expectation $\mathbb{E}[\cdot|x_t]$. By the definition of $\theta_n$,

$$\|\theta_{n+1} - \theta_{\star,t}\|_2^2 = \|\theta_n - \theta_{\star,t}\|_2^2 - 2\beta_n \langle \tilde{F}(\theta_n, \xi_n), \theta_n - \theta_{\star,t} \rangle + \beta_n^2 \|\tilde{F}^\theta(\theta_n, \xi_n)\|_2^2.$$

We will take expectation $\mathbb{E}_{\xi_n}$ where $\xi_n = (s_n, a_n, b_n, s_{n+1})$ is the sample at iteration $n$ of Algorithm 7

$$\mathbb{E}_{\xi_n} \tilde{F}(\theta_n, \xi_n) = F(\theta_n) + \gamma P_{x_t,y_t}(V_{k-1} - \hat{V}_{k-1}),$$

as in (18) where $P_{x_t,y_t}$ was also defined. As we stated, we omit the dependence of $F_t$ to $t$ as $t$ is fixed throughout this loop. Thus,

$$\mathbb{E}_{\xi_n} \|\theta_{n+1} - \theta_{\star,t}\|_2^2 = \|\theta_n - \theta_{\star,t}\|_2^2 - 2\beta_n \langle F(\theta_n), \theta_n - \theta_{\star,t} \rangle$$
$$- 2\beta_n \gamma \langle P_{x_t,y_t}(V_{k-1} - \hat{V}_{k-1}), \theta_n - \theta_{\star,t} \rangle + \beta_n^2 \|\tilde{F}(\theta_n, \xi_n)\|_2^2.$$

We use strong monotonicity (with $F(\theta_{\star,t}) = 0$) for the first inner product and Cauchy-Schwarz and Young's inequalities for the second inner product (exactly as in the proofs for policy evaluation with single loop NAC) to get

$$\mathbb{E}_{\xi_n}\|\theta_{n+1} - \theta_{\star,t}\|_2^2 \leq \left(1 - 2\beta_n\lambda_{\min}^\theta\right)\|\theta_n - \theta_{\star,t}\|_2^2 + \frac{\beta_n\gamma^2}{\lambda_{\min}^\theta}\|P_{x_t,y_t}(V_{k-1} - \hat{V}_{k-1})\|_2^2$$

$$+ \beta_n\lambda_{\min}^\theta\|\theta_n - \theta_{\star,t}\|_2^2 + \beta_n^2\mathbb{E}_{\xi_n}\|\tilde{F}(\theta_n, \xi_n)\|_2^2$$

$$= \left(1 - \beta_n\lambda_{\min}^\theta\right)\|\theta_n - \theta_{\star,t}\|_2^2 + \frac{\beta_n\gamma^2|S||A|}{\lambda_{\min}^\theta}\|V_{k-1} - \hat{V}_{k-1}\|_\infty^2 + \beta_n^2\mathbb{E}_{\xi_n}\|\tilde{F}(\theta_n, \xi_n)\|_2^2, \quad (31)$$

where we estimated $\|P_{x_t,y_t}(\hat{V}_{k-1} - V_{k-1})\|_2^2$ as in Lemma E.2. We will use Lemma B.1 to upper bound $\|\tilde{F}(\theta_n, \xi_n)\|_2^2 \leq \frac{2}{(1-\gamma)^2}$. We define $\Theta_n$ such that $\Theta_n(1 - \beta_n\lambda_{\min}) \leq \Theta_{n-1}$ with $\Theta_0 = \Theta_1 = 1$. We multiply both sides of the inequality with $\Theta_n$ after taking total expectation, to get

$$\Theta_n\mathbb{E}\|\theta_{n+1} - \theta_{\star,t}\|_2^2 \leq \Theta_{n-1}\mathbb{E}\|\theta_n - \theta_{\star,t}\|_2^2 + \frac{\Theta_n\beta_n\gamma^2|S||A|}{\lambda_{\min}^\theta}\mathbb{E}\|\hat{V}_{k-1} - V_{k-1}\|_\infty^2 + \frac{2\Theta_n\beta_n^2}{(1-\gamma)^2}.$$

Summing the inequality gives

$$\Theta_N\mathbb{E}\|\theta_{N+1} - \theta_{\star,t}\|_2^2 \leq \Theta_0\|\theta_1 - \theta_{\star,t}\|_2^2 + \sum_{n=1}^N \frac{\Theta_n\beta_n\gamma^2|S||A|}{\lambda_{\min}^\theta}\mathbb{E}\|\hat{V}_{k-1} - V_{k-1}\|_\infty^2 + \sum_{n=1}^N \frac{2\Theta_n\beta_n^2}{(1-\gamma)^2}.$$

Using the definition of $\beta_n$ and setting $\Theta_n(1 - \beta_n\lambda_{\min}) = \Theta_{n-1}$ gives $\Theta_n = \Theta_1 \frac{(n+n_0)(n+n_0-1)}{(n_0)(n_0+1)}$. Let us use $\Theta_0 = \Theta_1 = 1$ and bounds from Lemma B.1 for $\|\theta_1 - \theta_{\star,t}\|_2^2$,

$$\mathbb{E}\|\theta_N - \theta_{\star,t}\|_2^2 \leq \frac{2n_0(n_0+1)|S||A|}{(1-\gamma)^2(N+n_0)(N+n_0-1)} + \frac{8N}{(N+n_0)(N+n_0-1)(1-\gamma)^2(\lambda_{\min}^\theta)^2}$$

$$+ \frac{3\gamma^2|S||A|}{(\lambda_{\min}^\theta)^2}\mathbb{E}\|\hat{V}_{k-1} - V_{k-1}\|_\infty^2.$$

$\square$

We now analyze the bias for $\theta_{t+1}$ in Algorithm 6. Let us remark that the bias analysis for $\hat{V}_{k-1}$ in the next lemma is critical and it is the main reason that we get fresh estimates for $\hat{V}_{k-1}$ in this algorithm, in contrast to the stale estimation in the single loop NAC variant.

**Lemma 3.7.** *Let* $\beta_n = \frac{2}{\lambda_{\min}^\theta(n+n_0)}$ *where* $n_0 = \frac{6\lambda_{\max}^2}{\lambda_{\min}^2}$. *Then, for Algorithm 6 and its subroutine Algorithm 7*

$$\|\mathbb{E}[\theta_N|x_t] - \theta_{\star,t}\|^2 \leq \mathcal{O}\left(\frac{|S||A|}{(1-\gamma)^2N^2} + \frac{10|S||A|}{(\lambda_{\min}^\theta)^2}\|\mathbb{E}[\hat{V}_{k-1}|x_t] - V_{k-1}\|_\infty^2\right). \quad (32)$$

**Remark F.1.** *Since the bias term in the algorithm's analysis will be ( Lemma F.2)* $\|\mathbb{E}[\theta_N|x_t] - \theta_{\star,t}\|$, *we will have to take the square root of the result of this lemma. If* $\hat{V}_{k-1}$ *is estimated before* $x_t$, *then we will have in the main analysis* $\mathbb{E}\|\hat{V}_{k-1} - V_{k-1}\|$ *which will have the rate* $\frac{1}{\sqrt{N}}$. *On the other hand, if we estimate* $\hat{V}_{k-1}$ *freshly as in Algorithm 6, then we will be able to use the better bias bound* $\|\mathbb{E}[\hat{V}_{k-1}|x_t] - V_{k-1}\| = \mathcal{O}(1/N)$ *which seems to be enough for our bound.*

*Proof.* We are going to take expectation of the recursion

$$\theta_{n+1} = \theta_n - \beta_n\tilde{F}(\theta_n, \xi_n),$$

first w.r.t. sample $\xi_n$, where $\xi_n$ is as in the proof of Lemma 3.6

$$\mathbb{E}_{\xi_n}\theta_{n+1} = \theta_n - \beta_n\mathbb{E}_{\xi_n}\tilde{F}(\theta_n, \xi_n)$$

$$= \theta_n - \beta_nF(\theta_n) - \beta_n\gamma P_{x_t,y_t}(\hat{V}_{k-1} - V_{k-1}),$$

where we used (18) where $P_{x_t,y_t}$ was also defined.

We will now take expectation $\mathbb{E}[\cdot|x_t]$. We note $F$ and $P_{x_t,y_t}$ are linear

$$\mathbb{E}[\theta_{n+1}|x_t] = \mathbb{E}[\theta_n|x_t] - \beta_n F_t(\mathbb{E}[\theta_n|x_t]) - \beta_n \gamma P_{x_t,y_t}(\mathbb{E}[\hat{V}_{k-1}|x_t] - V_{k-1}).$$

We denote $\bar{\theta}_n = \mathbb{E}[\theta_n|x_t]$ and $\bar{\delta} = \gamma P_{x_t,y_t}(\mathbb{E}[\hat{V}_{k-1}|x_t] - V_{k-1})$ in the above equality which makes the recursion $\bar{\theta}_{n+1} = \bar{\theta}_n - \beta_n F(\bar{\theta}_n) - \beta_n \bar{\delta}$. We then have

$$\|\bar{\theta}_{n+1} - \theta_{\star,t}\|_2^2 = \|\bar{\theta}_n - \theta_{\star,t}\|_2^2 - 2\beta_n \langle F(\bar{\theta}_n), \bar{\theta}_n - \theta_{\star,t}\rangle - 2\beta_n \langle \bar{\delta}, \bar{\theta}_n - \theta_{\star,t}\rangle$$
$$+ \frac{3\beta_n^2}{2}\|F(\bar{\theta}_n)\|_2^2 + 3\beta_n^2\|\bar{\delta}\|_2^2, \quad (33)$$

where we also used Young's inequality to split the term $\beta_n^2\|F(\bar{\theta}_n) + \bar{\delta}\|^2$.

By strong monotonicity and Lipschitzness of $F$ along with $F(\theta_{\star,t}) = 0$,

$$2\beta_n \langle F(\bar{\theta}_n), \bar{\theta}_n - \theta_{\star,t}\rangle = 2\beta_n \langle F(\bar{\theta}_n) - F(\theta_{\star,t}), \bar{\theta}_n - \theta_{\star,t}\rangle \geq 2\beta_n \lambda_{\min}^\theta \|\bar{\theta}_n - \theta_{\star,t}\|_2^2,$$
$$\beta_n^2\|F(\bar{\theta}_n)\|_2^2 = \beta_n^2\|F(\bar{\theta}_n) - F(\theta_{\star,t})\|_2^2 \leq \beta_n^2 \lambda_{\max}^2 \|\bar{\theta}_n - \theta_{\star,t}\|_2^2.$$

By Cauchy-Schwarz and Young's inequalities, it follows that $2\beta_n \langle \bar{\delta}, \bar{\theta}_n - \theta_{\star,t}\rangle \leq \frac{\beta_n \lambda_{\min}^\theta}{2}\|\bar{\theta}_n - \theta_{\star,t}\|_2^2 + \frac{2\beta_n}{\lambda_{\min}^\theta}\|\bar{\delta}\|_2^2$. Using these three inequalities in (33) gives

$$\|\bar{\theta}_{n+1} - \theta_{\star,t}\|_2^2 \leq \left(1 - \frac{3}{2}\beta_n \lambda_{\min}^\theta + \frac{3}{2}\beta_n^2 \lambda_{\max}^2\right)\|\bar{\theta}_n - \theta_{\star,t}\|_2^2 + \frac{2\beta_n}{\lambda_{\min}^\theta}\|\bar{\delta}\|_2^2 + 3\beta_n^2\|\bar{\delta}\|_2^2.$$

We now use $n_0 = \frac{6\lambda_{\max}^2}{(\lambda_{\min}^\theta)^2}$ and $\beta_n = \frac{2}{\lambda_{\min}^\theta(n+n_0)}$ to estimate

$$\frac{3\beta_n}{2}\left(\lambda_{\min}^\theta - \beta_n \lambda_{\max}^2\right) = \frac{3\beta_n}{2}\left(\lambda_{\min}^\theta - \frac{2\lambda_{\max}^2}{\lambda_{\min}^\theta(n+n_0)}\right) \geq \frac{3\beta_n}{2}\left(\lambda_{\min}^\theta - \frac{2\lambda_{\max}^2}{\lambda_{\min}^\theta n_0}\right) = \beta_n \lambda_{\min}^\theta.$$

Therefore, the recursion is

$$\|\bar{\theta}_{n+1} - \theta_{\star,t}\|_2^2 \leq \left(1 - \beta_n \lambda_{\min}^\theta\right)\|\bar{\theta}_n - \theta_{\star,t}\|_2^2 + \frac{2\beta_n}{\lambda_{\min}^\theta}\|\bar{\delta}\|^2 + 3\beta_n^2\|\bar{\delta}\|_2^2.$$

This recursion is similar to (31), in particular, by noting $\beta_n \leq \frac{1}{\lambda_{\min}^\theta}$, and bounding $\|\bar{\delta}\|_2^2$ similar to Lemma E.2: $\|P_{x_t,y_t}(\mathbb{E}[\hat{V}_{k-1}|x_t] - V_{k-1})\|_2^2 \leq |S||A|\|P_{x_t,y_t}(\mathbb{E}[\hat{V}_{k-1}|x_t] - V_{k-1})\|_\infty^2 \leq |S||A|\|\mathbb{E}[\hat{V}_{k-1}|x_t] - V_{k-1}\|_\infty^2$

$$\|\bar{\theta}_{n+1} - \theta_{\star,t}\|^2 \leq \left(1 - \beta_n \lambda_{\min}^\theta\right)\|\bar{\theta}_n - \theta_{\star,t}\|^2 + \frac{5\beta_n|S||A|}{\lambda_{\min}^\theta}\|\mathbb{E}[\hat{V}_{k-1}|x_t] - V_{k-1}\|_\infty^2.$$

We finally define $\Theta_n$ as in the end of the proof of Lemma 3.6, in particular, $\Theta_n(1 - \beta_n \lambda_{\min}) = \Theta_{n-1}$ gives $\Theta_n = \Theta_1 \frac{(n+n_0)(n+n_0-1)}{n_0(n_0-1)}$, where $\Theta_0 = \Theta_1 = 1$. We multiply both sides of the inequality with $\Theta_n$ and sum to get the result. $\qquad\square$

We now have to estimate the bias and variance of the estimation of $\hat{V}_{k-1}$ in Algorithm 6, very similar to Lan (2021). Unlike Lan (2021) that derived $\mathcal{O}(1/N^3)$ bound for the bias, we are going to derive a $\mathcal{O}(1/N^2)$ bound which will be sufficient. Let us also note that the previous two lemmas had additional bias not present in Lan (2021), however the next result does not have this bias and therefore the arguments in Lan (2021) would be enough. We provide a brief proof to be self-contained.

Let us recall that $V_{k-1} = V^{x_{k-1},y_{k-1}}$ and by sampling $s_n \sim \rho^{x_{k-1},y_{k-1}}$, $a_n \sim x_{k-1}(\cdot|s_n)$, $b_n \sim y_{k-1}(\cdot|s_n)$, $s_{n+1} \sim P(\cdot|s_n,a_n,b_n)$, the oracle

$$\tilde{F}^\omega(\omega_n, \xi_n) = e(s_n)\left(\omega_n(s_n) - r(s_n, a_n, b_n) - \gamma\omega_n(s_{n+1})\right),$$

satisfies $\mathbb{E}_{\xi_n}\tilde{F}^\omega(\omega_n, \xi_n) = F^\omega(\omega_n)$, where $F^\omega$ is defined as

$$F_{k-1}^\omega(\omega)(s) = \rho^{x_{k-1},y_{k-1}}(s)\Big(\omega(s) - \sum_{a,b} x_{k-1}(a|s)y_{k-1}(b|s)r(s,a,b)$$
$$- \gamma \sum_{s',a,b} x_{k-1}(a|s)y_{k-1}(b|s)P(s'|s,a,b)\omega(s')\Big), \quad (34)$$

where $F_{k-1}^\omega(V_{k-1}) = 0$ and also as before $F_{k-1}^\omega$ is strongly monotone with $\lambda_{\min}^\omega$. We will drop the subscript of $F^\omega$ since $k$ is fixed in this loop.

**Lemma 3.8.** *Let Assumption 1, 2 hold and $\beta_n^\omega = \frac{2}{\lambda_{\min}^\omega(n+n_0)}$, with $n_0 = \frac{6\lambda_{\max}^2}{(\lambda_{\min}^\omega)^2}$. The variance and bias of $\hat{V}_{k-1}$, computed as in Algorithm 1 satisfies*

$$\mathbb{E}\|\omega_N - V_{k-1}\|_2^2 \le \mathcal{O}\left(\frac{|S||A|}{(1-\gamma)^2 N^2} + \frac{1}{N(1-\gamma)^2\lambda_{\min}^2}\right),$$

$$\|\mathbb{E}[\omega_N|x_t] - V_{k-1}\|_2^2 \le \mathcal{O}\left(\frac{|S||A|}{(1-\gamma)^2 N^2}\right).$$

*Proof.* For the variance, we have by taking expectation w.r.t. $\xi_n$

$$\mathbb{E}_{\xi_n}\|\omega_{n+1} - V_{k-1}\|_2^2 = \|\omega_n - V_{k-1}\|_2^2 - 2\beta_n\langle\mathbb{E}_{\xi_n}[\tilde{F}^\omega(\omega_n,\xi_n)], \omega_n - V_{k-1}\rangle + \beta_n^2\mathbb{E}_{\xi_n}\|\tilde{F}^\omega(\omega_n,\xi_n)\|_2^2.$$

By $\mathbb{E}_{\xi_n}\tilde{F}^\omega(\omega_n,\xi_n) = F^\omega(\omega_n)$, $F^\omega(V_{k-1}) = 0$, and strong monotonicity of $F^\omega$, similar to our previous proofs for policy evaluation,

$$\mathbb{E}\|\omega_{n+1} - V_{k-1}\|_2^2 = (1 - 2\beta_n\lambda_{\min}^\omega)\mathbb{E}\|\omega_n - V_{k-1}\|_2^2 + \beta_n^2\mathbb{E}\|\tilde{F}^\omega(\omega_n,\xi_n)\|_2^2.$$

The end of the proof is the same as Lemma 3.6, except that we do not have here the additional bias term in Lemma 3.6. Therefore, the result follows.

For the bias, we will argue as in Lemma 3.7. Taking expectation of the recursion w.r.t. $\xi_n$ gives

$$\mathbb{E}_{\xi_n}\omega_{n+1} = \omega_n - \beta_n F^\omega(\omega_n).$$

We now unroll the expectation until $x_t$ and use linearity of $F^\omega$

$$\mathbb{E}[\omega_{n+1}|x_t] = \mathbb{E}[\omega_n|x_t] - \beta_n F^\omega(\mathbb{E}[\omega_n|x_t]).$$

Denoting $\bar{\omega}_n = \mathbb{E}[\omega_n|x_t]$ gives the recursion $\bar{\omega}_{n+1} = \bar{\omega}_n - \beta_n F^\omega(\bar{\omega}_n)$, and therefore

$$\|\bar{\omega}_{n+1} - V_{k-1}\|_2^2 = \|\bar{\omega}_n - V_{k-1}\|_2^2 - 2\beta_n\langle F_{k-1}^\omega(\bar{\omega}_n), \bar{\omega}_n - V_{k-1}\rangle + \beta_n^2\|F^\omega(\bar{\omega}_n)\|_2^2.$$

We will now use Lipschitzness and strong monotonicity of $F^\omega$ and that $F^\omega(V_{k-1}) = 0$ and similar to Lemma 3.7, we obtain the recursion

$$\|\bar{\omega}_{n+1} - V_{k-1}\|_2^2 = \left(1 - 2\beta_n\lambda_{\min} + \beta_n^2\lambda_{\max}^2\right)\|\bar{\omega}_n - V_{k-1}\|_2^2.$$

By the choice of $n_0$ and $\beta_n$, similar to Lemma 3.7, it holds that $2\beta_n\lambda_{\min} - \beta_n^2\lambda_{\max}^2 \ge \beta_n\lambda_{\min}$. By defining $\Theta_n$ the same way as Lemma 3.7 and summing the inequality gives the result. $\square$

Now we analyze the outer algorithm for solving the matrix game in stage 1. The algorithm is based on FoRB from Malitsky & Tam (2020). The choice of this algorithm is due to its simple update with one projection and one oracle computation. We note that the existing analyses for stochastic versions of this algorithm are not suitable for us. In particular, in the stochastic variant in Malitsky & Tam (2020), deterministic oracle is also computed at each iteration. On the other hand, the analysis in Böhm et al. (2020) uses unbiased oracles with bounded variance and a decreasing step size. In our case, we will have biased samples and we will use inner loops to decrease bias and variance of this oracle. Therefore, we need to develop an analysis with constant step size and that characterizes the bias and variance explicitly.

Similar to Malitsky & Tam (2020), let us define the "Lyapunov-like" function

$$\Phi_{t+1}^s = D(x(\cdot|s), x_{t+1}(\cdot|s)) + \eta\langle\theta_{\star,t+1}(s,\cdot) - \theta_{t+1}(s,\cdot), x(\cdot|s) - x_{t+1}(\cdot|s)\rangle$$
$$+ \frac{1}{2}D(x_{t+1}(\cdot|s), x_t(\cdot|s)). \quad (35)$$

We call this "Lyapunov-like" since it is not non-decreasing. Moreover, unlike Malitsky & Tam (2020), $\Phi_t$ is not necessarily nonnegative. However, it is sufficient for our purposes as it is bounded. Note that we will also use the following error functions

$$e_{1,t} = \eta\langle\theta_{t+1}(\cdot|s) - \mathbb{E}[\theta_{t+1}(\cdot|s)|x_t], x(\cdot|s) - x_t(\cdot|s)\rangle$$
$$e_{2,t} = \eta\langle\theta_{t+1}^y(\cdot|s) - \mathbb{E}[\theta_{t+1}^y(\cdot|s)|y_t], y(\cdot|s) - y(\cdot|s)\rangle.$$

**Lemma F.2.** *[See Lemma 3.4] Let Assumption 1, 2 hold. Denote $x_{out} = \frac{1}{T}\sum_{t=1}^{T} x_t$ and $y_{out} = \frac{1}{T}\sum_{t=1}^{T} y_t$ and let $\eta = \frac{1-\gamma}{8}$*

$$\mathbb{E}\mathbb{E}_{s\sim\sigma}\left[\max_{x^s,y^s} x_{out}^s Q^s y^s - x^s Q^s y_{out}\right] = \mathcal{O}\left(\frac{\Phi_0^s - \Phi_T^s}{\eta T}\right) + \mathcal{O}\left(\frac{1}{T}\sum_{t=1}^{T}\mathbb{E}\|\mathbb{E}[\theta_{t+1}|x_t] - \theta_{\star,t}\|\right)$$

$$+ \mathcal{O}\left(\frac{1}{T}\sum_{t=1}^{T}\eta\mathbb{E}\|\theta_{t+1} - \theta_{\star,t}\|^2 + \mathbb{E}\|\theta_t - \theta_{\star,t-1}\|^2\right) + \frac{1}{T\eta}\mathbb{E}\mathbb{E}_{s\sim\sigma}\max_z\sum_{t=1}^{T}[e_{1,t} + e_{2,t}]).$$

**Remark F.3.** *When $D$ is KL divergence, we have $\max_{y_1,y_2} D(y_1,y_2) \leq \log|B|$ and equivalently for the other player. Therefore, we have the bound*

$$\Phi_0^s - \Phi_T^s + \Phi_{0,y}^s - \Phi_{T,y}^s \leq \mathcal{O}\left(\log|A| + \log|B| + \frac{\eta}{1-\gamma}\right).$$

**Corollary F.4.** *We use Lemmas 3.6, 3.7 and F.6,*

$$\tilde{\mathcal{O}}\left(\frac{1}{T(1-\gamma)}\right) + \mathcal{O}\left(\frac{\sqrt{|S||A|}}{(1-\gamma)N} + \frac{\sqrt{|S||A|}}{\lambda_{\min}^\theta}\|\mathbb{E}[\hat{V}_{k-1}|x_t] - V_{k-1}\|_\infty\right)$$

$$+ \mathcal{O}\left(\frac{|S||A|}{(1-\gamma)^2 N^2} + \frac{1}{N(\lambda_{\min}^\theta)^2(1-\gamma)^2} + \frac{|S||A|}{(\lambda_{\min}^\theta)^2}\mathbb{E}\|\hat{V}_{k-1} - V_{k-1}\|_\infty^2\right).$$

*We add the bound from Lemma 3.8*

$$\tilde{\mathcal{O}}\left(\frac{1}{T(1-\gamma)}\right) + \mathcal{O}\left(\frac{\sqrt{|S||A|}}{(1-\gamma)N} + \frac{\sqrt{|S||A|}}{\lambda_{\min}^\theta}\left[\frac{\sqrt{|S||A|}}{(1-\gamma)N}\right]\right)$$

$$+ \mathcal{O}\left(\frac{|S||A|}{(1-\gamma)^2 N^2} + \frac{1}{N(\lambda_{\min}^\theta)^2(1-\gamma)^2} + \frac{|S||A|}{(\lambda_{\min}^\theta)^2}\left[\frac{|S||A|}{(1-\gamma)^2 N^2} + \frac{1}{N(1-\gamma)^2(\lambda_{\min}^\omega)^2}\right]\right).$$

*We now refine the bound by only including the dominant terms*

$$\tilde{\mathcal{O}}\left(\frac{1}{T(1-\gamma)}\right) + \mathcal{O}\left(\frac{|S||A|}{\lambda_{\min}^\theta(1-\gamma)^2 N}\right)$$

$$+ \mathcal{O}\left(\frac{|S|^2|A|^2}{N^2(1-\gamma)^2(\lambda_{\min}^\theta)^2} + \frac{|S||A|}{(\lambda_{\min}^\theta)^2(\lambda_{\min}^\omega)^2(1-\gamma)^2 N}\right)$$

**Remark F.5.** *By Lemma 3.6 and Lemma 3.7, the second and third term will bring $\mathcal{O}\left(\frac{1}{N} + \mathbb{E}\|\mathbb{E}[\hat{V}_{k-1}|x_t] - V_{k-1}\|\right)$. We will see in the next lemma how to handle error terms $e_1, e_2$ and will use the bound derived earlier for $\mathbb{E}\|\mathbb{E}[\hat{V}_{k-1}|x_t] - V_{k-1}\|$.*

*Proof.* By the update rule, it follows for all $s$ and $x(\cdot|s) \in \Delta$,

$$\langle\nabla D(x_{t+1}(\cdot|s), x_t(\cdot|s)) + \eta(2\theta_{t+1}(s,\cdot) - \theta_t(s,\cdot)), x(\cdot|s) - x_{t+1}(\cdot|s)\rangle \geq 0.$$

By three point identity,

$$D(x(\cdot|s), x_{t+1}(\cdot|s)) \leq D(x(\cdot|s), x_t(\cdot|s)) - D(x_{t+1}(\cdot|s), x_t(\cdot|s))$$
$$+ \eta\langle(2\theta_{t+1}(s,\cdot) - \theta_t(s,\cdot)), x(\cdot|s) - x_{t+1}(\cdot|s)\rangle. \quad (36)$$

We now manipulate the inner product by adding and subtracting $\theta_{\star,t+1}$

$$\eta\langle(2\theta_{t+1}(s,\cdot) - \theta_t(s,\cdot)), x(\cdot|s) - x_{t+1}(\cdot|s)\rangle = \eta\langle\theta_{t+1} - \theta_{\star,t+1}(s,\cdot), x(\cdot|s) - x_{t+1}(\cdot|s)\rangle$$
$$+ \eta\langle\theta_{t+1}(s,\cdot) - \theta_t(s,\cdot) + \theta_{\star,t+1}(s,\cdot), x(\cdot|s) - x_{t+1}(\cdot|s)\rangle$$
$$= \eta\langle\theta_{t+1}(s,\cdot) - \theta_{\star,t+1}(s,\cdot), x(\cdot|s) - x_{t+1}(\cdot|s)\rangle + \eta\langle\theta_{t+1}(s,\cdot) - \theta_t(s,\cdot), x(s,\cdot) - x_t(\cdot|s)\rangle$$
$$+ \eta\langle\theta_{t+1}(s,\cdot) - \theta_t(s,\cdot), x_t(\cdot|s) - x_{t+1}(\cdot|s)\rangle + \eta\langle\theta_{\star,t+1}(s,\cdot), x(\cdot|s) - x_{t+1}(\cdot|s)\rangle. \quad (37)$$

The first two inner products in the final inequality will telescope if we can replace $\theta_{t+1}$ with $\theta_{\star,t}$ in the second one. For this we have to be careful with bias and variance. Let us take the second inner product

$$\eta\langle\theta_{t+1}(s,\cdot) - \theta_t(s,\cdot), x(\cdot|s) - x_t(\cdot|s)\rangle = \eta\langle\theta_{\star,t}(s,\cdot) - \theta_t(s,\cdot), x(\cdot|s) - x_t(\cdot|s)\rangle$$
$$+\eta\langle\theta_{t+1}(s,\cdot) - \theta_{\star,t}(s,\cdot), x(\cdot|s) - x_t(\cdot|s)\rangle.$$

Now in this estimation, we will add and subtract terms involving $\mathbb{E}[\theta_{t+1}(s,\cdot)|x_t]$ to obtain

$$\eta\langle\theta_{t+1}(s,\cdot) - \theta_t(s,\cdot), x(\cdot|s) - x_t(\cdot|s)\rangle = \eta\langle\theta_{\star,t}(s,\cdot) - \theta_t(s,\cdot), x(\cdot|s) - x_t(\cdot|s)\rangle$$
$$+ \eta\langle\mathbb{E}[\theta_{t+1}(s,\cdot)|x_t] - \theta_{\star,t}, x(\cdot|s) - x_t(\cdot|s)\rangle + \eta\langle\theta_{t+1}(s,\cdot) - \mathbb{E}[\theta_{t+1}(s,\cdot)|x_t], x(\cdot|s) - x_t(\cdot|s)\rangle$$
$$\leq \eta\langle\theta_{\star,t}(s,\cdot) - \theta_t(s,\cdot), x(\cdot|s) - x_t(\cdot|s)\rangle + 2\eta\|\mathbb{E}[\theta_{t+1}|x_t] - \theta_{\star,t}\|_\infty + e_{1,t}, \quad (38)$$

where the inequality is due to Cauchy-Schwarz and we use the definition of $e_{1,t}$ for the last term. Next, we use Cauchy-Schwarz and Young's inequalities for the third inner product in RHS of (37) to derive

$$\eta\langle\theta_{t+1}(s,\cdot) - \theta_t(\cdot|s), x_t(\cdot|s) - x_{t+1}(\cdot|s)\rangle \leq \eta^2\|\theta_{t+1}(s,\cdot) - \theta_t(\cdot|s)\|_\infty^2 + \frac{1}{4}\|x_t(\cdot|s) - x_{t+1}(\cdot|s)\|_1^2$$
$$\leq 4\eta^2\left[\|\theta_{t+1}(s,\cdot) - \theta_{\star,t}(s,\cdot)\|_\infty^2 + \|\theta_{\star,t}(s,\cdot) - \theta_{\star,t-1}(s,\cdot)\|_\infty^2 + \|\theta_{\star,t-1}(s,\cdot) - \theta_t(s,\cdot)\|_\infty^2\right]$$
$$+ \frac{1}{4}\|x_t(\cdot|s) - x_{t+1}(\cdot|s)\|_1^2. \quad (39)$$

As $\theta_{\star,t}(s,a) = \mathbb{E}_{b\sim y_t(\cdot|s)}Q(s,a,b)$, we have

$$\|\theta_{\star,t}(s,\cdot) - \theta_{\star,t-1}(s,\cdot)\|_\infty \leq \max_{b,a}|Q(s,a,b)|\|y_t(\cdot|s) - y_{t-1}(\cdot|s)\|_1$$
$$\leq \frac{2}{1-\gamma}\|y_t(\cdot|s) - y_{t-1}(\cdot|s)\|_1, \quad (40)$$

where the second inequality is by Lemma B.1 and the first by Jensen. We join (38), (39), and (40) in (37)

$$\eta\langle(2\theta_{t+1}(s,\cdot) - \theta_t(s,\cdot)), x(\cdot|s) - x_{t+1}(\cdot|s)\rangle \leq \eta\langle\theta_{t+1}(s,\cdot) - \theta_{\star,t+1}(s,\cdot), x(\cdot|s) - x_{t+1}(\cdot|s)\rangle$$
$$\eta\langle\theta_{\star,t}(s,\cdot) - \theta_t(s,\cdot), x(\cdot|s) - x_t(\cdot|s)\rangle + 2\eta\|\mathbb{E}[\theta_{t+1}|x_t] - \theta_{\star,t}\|_\infty + e_{1,t}$$
$$+ 4\eta^2\left[\|\theta_{t+1} - \theta_{\star,t}\|_\infty^2 + \|\theta_{\star,t-1} - \theta_t\|_\infty^2\right] + \frac{16\eta^2}{(1-\gamma)^2}\|y_t(\cdot|s) - y_{t-1}(\cdot|s)\|_1^2$$
$$+ \frac{1}{4}\|x_t(\cdot|s) - x_{t+1}(\cdot|s)\|_1^2 + \eta\langle\theta_{\star,t+1}(s,\cdot), x(\cdot|s) - x_{t+1}(\cdot|s)\rangle. \quad (41)$$

We note that by Equation (8), $\frac{1}{4}\|x_t(\cdot|s) - x_{t+1}(\cdot|s)\|_1^2 \leq \frac{1}{2}D(x_{t+1}(\cdot|s), x_t(\cdot|s))$ and similarly for the term involving difference of $y_t$ and $y_{t-1}$.

We insert (41) into (36) by using the definition of $\Phi_t$

$$\eta\langle\theta_{\star,t+1}(s,\cdot), x_{t+1}(\cdot|s) - x(\cdot|s)\rangle + \Phi_{t+1}^s \leq \Phi_t^s + e_{1,t} + 2\eta\|\mathbb{E}[\theta_{t+1}|x_t] - \theta_{\star,t}\|_\infty$$
$$+ 4\eta^2\left[\|\theta_{t+1} - \theta_{\star,t}\|_\infty^2 + \|\theta_{\star,t-1} - \theta_t\|_\infty^2\right]$$
$$+ \frac{32\eta^2}{(1-\gamma)^2}D(y_t(\cdot|s), y_{t-1}(\cdot|s)) - \frac{1}{2}D(x_t(\cdot|s), x_{t-1}(\cdot|s)).$$

We sum this inequality and use the definition of $\theta_{\star,t+1}$ to obtain

$$\frac{\eta}{T}\sum_{t=0}^{T-1}\langle\mathbb{E}_{b\sim y_{t+1}(\cdot|s)}Q(s,\cdot,b), x_{t+1}(\cdot|s) - x(\cdot|s)\rangle \leq \frac{\Phi_0^s - \Phi_T^s}{T} + \frac{1}{T}\sum_{t=0}^{T-1}e_{1,t}$$
$$+ \frac{1}{T}\sum_{t=0}^{T-1}2\eta\|\mathbb{E}[\theta_{t+1}|x_t] - \theta_{\star,t}\|_\infty + \frac{4\eta^2}{T}\sum_{t=1}^{T}\left[\|\theta_{t+1} - \theta_{\star,t}\|_\infty^2 + \|\theta_{\star,t-1} - \theta_t\|_\infty^2\right]$$
$$+ \frac{1}{T}\sum_{t=0}^{T}\frac{32\eta^2}{(1-\gamma)^2}D(y_t(\cdot|s), y_{t-1}(\cdot|s)) - \frac{1}{2}D(x_t(\cdot|s), x_{t-1}(\cdot|s)).$$

We have to estimate the error terms in the last line. The terms in the second line will be the bias and variance arising from using $\theta_{t+1}$ instead of the true oracle. First, by the symmetric estimation on the $y$ player, we can obtain the similar inequality. For making the comparison, we will denote the corresponding oracle as $\theta^y$ ($\theta$ in the previous estimations correspond to $\theta^x$). In particular $\theta^y_{\star,t+1}(s,b) = \mathbb{E}_{a \sim x_{t+1}(\cdot|s)} Q(s,a,b)$, and the corresponding Lyapunov-like function as $\Phi^s_{t,y}$

$$\frac{\eta}{T} \sum_{t=0}^{T-1} \langle \mathbb{E}_{a \sim x_{t+1}(\cdot|s)} Q(s,a,\cdot), y(\cdot|s) - y_{t+1}(\cdot|s)\rangle \leq \frac{\Phi^s_{0,y} - \Phi^s_{y,T}}{T} + \frac{1}{T} \sum_{t=0}^{T-1} e_{2,t}$$

$$+ \frac{1}{T} \sum_{t=0}^{T-1} 2\eta \|\mathbb{E}[\theta^y_{t+1}|x_t] - \theta^y_{\star,t}\|_\infty + \frac{4\eta^2}{T} \sum_{t=1}^{T} \left[\|\theta^y_{t+1} - \theta^y_{\star,t}\|^2_\infty + \|\theta^y_{\star,t-1} - \theta^y_t\|^2_\infty\right]$$

$$+ \frac{1}{T} \sum_{t=0}^{T} \frac{32\eta^2}{(1-\gamma)^2} D(x_t(\cdot|s), x_{t-1}(\cdot|s)) - \frac{1}{2} D(y_t(\cdot|s), y_{t-1}(\cdot|s)).$$

After summing up the two inequalities and recalling that we bound the RHS of (24), we pick $\eta \leq \frac{1-\gamma}{8}$ to cancel the last terms in the last lines of the estimations. Since we estimate $\theta_t$ and $\theta^y_t$ in the same way, their bounds as we derived in Lemma 3.7, Lemma 3.6 will be the same, therefore in the bound we do not include both and simply put them under big-Oh. Next, we take maximum over $x, y$, take expectation w.r.t. state distribution $\sigma$ and total expectation w.r.t. randomness in the algorithm and use the definitions of $x_{\text{out}}$ and $y_{\text{out}}$ to conclude the result. $\qquad\square$

For the error terms $e_{1,t}, e_{2,t}$, we will use the technique to change the order of maximum and expectation from the literature of stochastic primal-dual methods (Nemirovski et al., 2009, Lemma 3.1, Lemma 6.1). Let us recall their definitions:

$$e_{1,t} = \eta \langle \theta_{t+1}(\cdot|s) - \mathbb{E}[\theta_{t+1}(\cdot|s)|x_t], x(\cdot|s) - x_t(\cdot|s)\rangle$$
$$e_{2,t} = \eta \langle \theta^y_{t+1}(\cdot|s) - \mathbb{E}[\theta^y_{t+1}(\cdot|s)|y_t], y_t(\cdot|s) - y(\cdot|s)\rangle$$

We will derive the bound or $e_{1,t}$ and the bound for $e_{2,t}$ is symmetrical.

**Lemma F.6.** *We have*

$$\frac{1}{T} \mathbb{E}\mathbb{E}_{s \sim \sigma} \max_x \sum_{t=1}^{T} e_{1,t} \leq \frac{\log |A|}{T} + \frac{1}{T} \sum_{t=1}^{T} 4\eta^2 \mathbb{E}\|\theta_{t+1} - \theta_{\star,t}\|^2_\infty.$$

*Proof.* First note that $\langle \theta_{t+1}(\cdot|s) - \mathbb{E}[\theta_{t+1}(\cdot|s)|x_t], x_t(\cdot|s)\rangle$ does not depend on $x$ and by the tower property of conditional expectation,

$$\sum_{t=1}^{T} \mathbb{E}\mathbb{E}_{s \sim \sigma} \eta \langle \theta_{t+1}(\cdot|s) - \mathbb{E}[\theta_{t+1}(\cdot|s)|x_t], x_t(\cdot|s)\rangle$$

$$= \mathbb{E}\mathbb{E}_{s \sim \sigma} \eta \langle \mathbb{E}[\theta_{t+1}(\cdot|s)|x_t] - \mathbb{E}[\theta_{t+1}(\cdot|s)|x_t], x_t(\cdot|s)\rangle = 0.$$

Therefore, we have to estimate

$$\mathbb{E}\mathbb{E}_{s \sim \sigma} \max_x \sum_{t=1}^{T} \eta \langle \theta_{t+1}(\cdot|s) - \mathbb{E}[\theta_{t+1}(\cdot|s)|x_t], x(\cdot|s)\rangle.$$

Let $n_t(s,\cdot) = -\eta(\theta_{t+1}(\cdot|s) - \mathbb{E}[\theta_{t+1}(\cdot|s)|x_t])$. First, we note that $\mathbb{E}[n_t(s,\cdot)|x_t] = 0$. Next, we define the auxiliary "ghost" process

$$\tilde{x}_{t+1}(\cdot|s) = \arg\min_x \langle n_t(s,\cdot), x(\cdot|s)\rangle + D(x(\cdot|s), \tilde{x}_t(\cdot|s)).$$

Note that $\tilde{x}_t$ and $x_t$ depend on the same randomness by definition of $\tilde{x}_t$, therefore conditioned on $x_t$, $\tilde{x}_t$ is deterministic. Standard mirror descent analysis gives for any $x$

$$\langle n_t(s,\cdot), x(\cdot|s)\rangle \leq D(x(\cdot|s), \tilde{x}_t(\cdot|s)) - D(x(\cdot|s), \tilde{x}_{t+1}(\cdot|s)) + \langle n_t(s,\cdot), \tilde{x}_t(\cdot|s)\rangle + \|n_t(\cdot|s)\|^2_*.$$

We sum the inequality take maximum and then expectation

$$\mathbb{E}\mathbb{E}_{s\sim\sigma}\max_x \sum_{t=1}^{T}\langle -n_t(s,\cdot), x(\cdot|s)\rangle \leq \mathbb{E}_{s\sim\sigma}D(x(\cdot|s),\tilde{x}_1(\cdot|s)) + \sum_{t=1}^{T}\mathbb{E}\mathbb{E}_{s\sim\sigma}\langle -n_t(s,\cdot), \tilde{x}_t(\cdot|s)\rangle$$

$$+ \sum_{t=1}^{T}\mathbb{E}\mathbb{E}_{s\sim\rho}\|n_t(\cdot|s)\|_{\infty}^2.$$

By the tower property and that $\tilde{x}_t$ is deterministic conditioned on $x_t$, we have $\sum_{t=1}^{T}\mathbb{E}\langle n_t(s,\cdot),\tilde{x}_t(\cdot|s)\rangle = \sum_{t=1}^{T}\mathbb{E}\langle\mathbb{E}[n_t(s,\cdot)|x_t],\tilde{x}_t(\cdot|s)\rangle = 0$.

Recall the definition of $n_t$ and use Young's inequality with Jensen's inequality to get

$$\mathbb{E}\|n_t(s,\cdot)\|^2 = \mathbb{E}\eta^2\|\theta_{t+1}(\cdot|s) - \mathbb{E}[\theta_{t+1}(\cdot|s)|x_t]\|_{\infty}^2$$

$$\leq 2\mathbb{E}\eta^2\|\theta_{t+1}(\cdot|s) - \theta_{\star,t}\|_{\infty}^2 + 2\mathbb{E}\eta^2\|\theta_{\star,t} - \mathbb{E}[\theta_{t+1}(\cdot|s)|x_t]\|_{\infty}^2$$

$$\leq 4\mathbb{E}\eta^2\|\theta_{t+1}(\cdot|s) - \theta_{\star,t}\|_{\infty}^2.$$

$\square$

### F.2 PROOFS FOR STAGE 2 OF REFLECTED NAC WITH A GAME ETIQUETTE

Similar to single loop NAC variant, this part mirror closely the analyses for single agent setting, as the best response step is like a single agent problem where the other agent (fixed) can be seen as part of the environment. Therefore, the development in this part will be similar to Lan (2021). Let us restate that the main concern in this part was to make sure that $\bar{y}_t$ updates do not require seeing the policy $x_k$ or the actions of $x$-player. As we showed that it is the case (in Appendix E.2.1), we will only provide the proofs here, with mostly using the arguments of Lan (2021). Therefore, the proofs in this part are included for being self-contained and for easy navigation. Therefore, they will be brief.

First, we will prove the bias and variance of the estimate $\nu_t$, similar to Lemma 3.8. Let us recall the Bellman operator for the oracle $\nu_{\star,t} = \mathbb{E}_{a\sim x_k(\cdot|s)}Q^{x_k,\bar{y}_t}(\cdot,a,\cdot)$ that the update is using:

$$\nu_{\star,t}(s,b) = \sum_a x_k(a|s)r(s,a,b) + \gamma\sum_{s',a,b'}P(s'|s,a,b)x_k(a|s)\bar{y}_t(b'|s')\nu_{\star,t}(s',b')$$

We use the sampling matrix (as (Lan, 2021, Sec. 5.2)) $\mathrm{diag}(\rho^{x_k,\bar{y}_t})\otimes\mathrm{diag}(\bar{y}_t)$ and define the operator

$$F_t^{\nu}(\nu_t)(s,b) = \rho^{x_k,\bar{y}_t}\bar{y}_t(b|s)\Big[\nu_t(s,b) - \sum_a x_k(a|s)r(s,a,b)$$

$$- \gamma\sum_{s',a,b'}x_k(a|s)P(s'|s,a,b)\bar{y}_t(b'|s')\nu_t(s',b')\Big], \quad (42)$$

such that $F_t^{\nu}(\nu_{\star,t}) = 0$. Strong monotonicity of $F_t$ follows from Assumption 1, 2 and that the operator $T\nu(s,b) = \sum_a x_k(a|s)r(s,a,b) + \gamma\sum_{s',a,b'}x_k(a|s)P(s'|s,a,b)\bar{y}_t(b'|s')\nu(b',s')$ being $\gamma$ contraction in $\ell_{\infty}$ norm (Bauschke et al., 2011, Example 22.6 and 20.7). We define the stochastic operator after sampling $s_t\sim\rho^{x_k,\bar{y}_t}$, $a_t\sim x_k(\cdot|s_t)$, $b_t\sim\bar{y}_t(\cdot|s_t)$, $s_{t+1}\sim P(\cdot|s_t,a_t,b_t)$, $b_{t+1}\sim \bar{y}_t(\cdot|s_{t+1})$

$$\tilde{F}_t^{\nu}(\nu_t,\xi_t) = e(s_t,b_t)\left(\nu_t(s_t,b_t) - r(s_t,a_t,b_t) - \gamma\nu_t(s_{t+1},b_{t+1})\right),$$

and as we assume we can sample $s_t\sim\rho^{x_k,\bar{y}_t}$, $\mathbb{E}_{\xi_t}[\tilde{F}_t^{\nu}(\nu_t,\xi_t)] = F_t^{\nu}(\nu_t)$. In particular, as long as $s_t,a_t,b_t,s_{t+1}$ are estimated in the prescribed way, there is no need for $\bar{y}_t$ update to see the actions or policy of $x_k$ for $\tilde{F}_t^{\nu}(\nu_t,\xi_t)$ to be unbiased estimate of $F_t^{\nu}(\nu_t)$.

We note that unlike the NAC case, we are having an inner loop to estimate $\nu_{\star,t}$. At the point of view of this loop (runs from $n = 0,\cdots,N-1$), $\nu_{\star,t}$ is fixed.

**Lemma F.7.** *Let Assumption 1, 2 hold and $\beta_n = \frac{2}{\lambda_{\min}^{\nu}(n+n_0)}$ stage 2 in Algorithm 6 satisfies*

$$\mathbb{E}\|\nu_N - \nu_{\star,t}\|_2^2 \leq \mathcal{O}\left(\frac{|S||B|}{(1-\gamma)^2 N^2} + \frac{1}{N(1-\gamma)^2(\lambda_{\min}^{\nu})^2}\right),$$

$$\|\mathbb{E}[\nu_N|\bar{y}_t] - \nu_{\star,t}\|_2^2 \leq \mathcal{O}\left(\frac{2|S||B|}{(1-\gamma)^2 N^2}\right).$$

*Proof.* For the variance, we have by taking expectation w.r.t. $\xi_n = (s_n, a_n, b_n, s_{n+1}, b_{n+1})$

$$\mathbb{E}_{\xi_n}\|\nu_{n+1} - \nu_{\star,t}\|_2^2 = \|\nu_n - \nu_{\star,t}\|_2^2 - 2\beta_n\langle\mathbb{E}_{\xi_n}[\tilde{F}_t^\nu(\nu_n,\xi_n)],\nu_n - \nu_{\star,t}\rangle + \beta_n^2\mathbb{E}_{\xi_n}\|\tilde{F}_t^\nu(\nu_n,\xi_n)\|_2^2.$$

By $\mathbb{E}_{\xi_n}\tilde{F}_t^\nu(\nu_n,\xi_n) = F_t^\nu(\nu_n)$, $F_t^\nu(\nu_{\star,t}) = 0$, and strong monotonicity of $F_t^\nu$,

$$\mathbb{E}\|\nu_{n+1} - \nu_{\star,t}\|_2^2 = (1 - 2\beta_n\lambda_{\min}^\nu)\,\mathbb{E}\|\nu_n - \nu_{\star,t}\|_2^2 + \beta_n^2\mathbb{E}\|\tilde{F}_t^\nu(\nu_n,\xi_n)\|_2^2.$$

The end of the proof is the same as Lemma 3.6, except that we do not have here the additional bias term in Lemma 3.6. Therefore, the result follows.

For the bias, we will argue as in Lemma 3.7. Taking expectation of the recursion w.r.t. $\xi_n$ gives

$$\mathbb{E}_{\xi_n}\nu_{n+1} = \nu_n - \beta_n F_t^\nu(\nu_n).$$

We now unroll the expectation until $y_t$ and use linearity of $F_t^\nu$

$$\mathbb{E}[\nu_{n+1}|\bar{y}_t] = \mathbb{E}[\nu_n|\bar{y}_t] - \beta_n F_t^\nu(\mathbb{E}[\nu_n|\bar{y}_t]).$$

Denoting $\bar{\nu}_n = \mathbb{E}[\nu_n|\bar{y}_t]$ gives

$$\|\bar{\nu}_{n+1} - \nu_{\star,t}\|_2^2 = \|\bar{\nu}_n - \nu_{\star,t}\|_2^2 - 2\beta_n\langle F_t^\nu(\bar{\nu}_n), \bar{\nu}_n - \nu_{\star,t}\rangle + \beta_n^2\|F_t^\nu(\bar{\nu}_n)\|_2^2.$$

We will now use Lipschitzness and strong monotonicity of $F_t^\nu$ and that $F_t^\nu(\nu_{\star,t}) = 0$ and similar to Lemma 3.7, we obtain the recursion

$$\|\bar{\nu}_{n+1} - \nu_{\star,t}\|_2^2 = \left(1 - 2\beta_n\lambda_{\min}^\nu + \beta_n^2\lambda_{\max}^2\right)\|\bar{\nu}_n - \nu_{\star,t}\|_2^2.$$

By the choice of $n_0$ and $\beta_n$, similar to Lemma 3.7, it holds that $2\beta_n\lambda_{\min}^\nu - \beta_n^2\lambda_{\max}^2 \geq \beta_n\lambda_{\min}^\nu$. By defining $\Theta_n$ the same way as Lemma 3.7 and summing the inequality gives the result. $\square$

We will now give a proof similar to (Lan, 2021, Theorem 2), (Agarwal et al., 2020) regarding the NPG algorithm for finding the best response.

**Theorem F.8.** *Let Assumption 1, 2 hold and $\eta > 0$. For the stage 2 of Algorithm 6.*

$$\frac{1}{T}\sum_{t=1}^T V^{x_k,y_k^*}(s_0) - V^{x_k,\bar{y}_t}(s_0) \leq \frac{\eta}{(1-\gamma)T}\mathbb{E}\left[\mathbb{E}_{s\sim d_{s_0}^{x_k,y_k^*}}D(y_k^*(\cdot|s),y_1(\cdot|s)) - V^{x_k,\bar{y}_1}(s) + V^{x_k,\bar{y}_{t+1}}(s)\right]$$

$$+ \frac{\eta}{2(1-\gamma)^2}\frac{1}{T}\sum_{t=1}^T\mathbb{E}\|\nu_{\star,t} - \nu_{t+1}\|_\infty^2 + \frac{2}{1-\gamma}\frac{1}{T}\sum_{t=1}^T\mathbb{E}\|\mathbb{E}[\nu_{t+1}|\bar{y}_t] - \nu_{\star,t}\|_\infty$$

**Corollary F.9.** *We use the bound from Lemma F.7 to obtain*

$$\tilde{\mathcal{O}}\left(\frac{1}{T(1-\gamma)^2}\right) + \mathcal{O}\left(\frac{|S||B|}{(1-\gamma)^4 N^2} + \frac{1}{N(1-\gamma)^4(\lambda_{\min}^\nu)^2}\right) + \mathcal{O}\left(\frac{\sqrt{|S||B|}}{(1-\gamma)^2 N}\right). \tag{43}$$

*Proof.* By the update rule of $\bar{y}_{t+1}$, it follows for any $s, \bar{y}$ (Tseng, 2008, Property 1)

$$D(\bar{y}(\cdot|s),\bar{y}_{t+1}(\cdot|s)) \leq D(\bar{y}(\cdot|s),\bar{y}_t(\cdot|s)) - D(\bar{y}_{t+1}(\cdot|s),\bar{y}_t(\cdot|s)) - \langle\eta\nu_{t+1}(s,\cdot),\bar{y}(\cdot|s) - \bar{y}_{t+1}(\cdot|s)\rangle. \tag{44}$$

We manipulate the inner product

$$-\eta\langle\nu_{t+1}(s,\cdot),\bar{y}(\cdot|s) - \bar{y}_{t+1}(\cdot|s)\rangle = -\eta\langle\nu_{t+1}(s,\cdot),\bar{y}(\cdot|s) - \bar{y}_t(\cdot|s)\rangle - \eta\langle\nu_{t+1}(s,\cdot),\bar{y}_t(\cdot|s) - \bar{y}_{t+1}(\cdot|s)\rangle$$

$$= -\eta\langle\nu_{\star,t}(s,\cdot),\bar{y}(\cdot|s) - \bar{y}_t(\cdot|s)\rangle - \eta\langle\nu_{t+1}(s,\cdot),\bar{y}_t(\cdot|s) - \bar{y}_{t+1}(\cdot|s)\rangle$$

$$- \eta\langle\nu_{t+1}(s,\cdot) - \nu_{\star,t}(s,\cdot),\bar{y}(\cdot|s) - \bar{y}_t(\cdot|s)\rangle. \tag{45}$$

By the performance difference lemma and using the definition of $\nu_{\star,t} = \mathbb{E}_{a\sim x_k} Q^{x_k,\bar{y}_t}(\cdot, a, \cdot)$.

$$V^{x_k,\bar{y}_{t+1}}(s_0) - V^{x_k,\bar{y}_t}(s_0) = \frac{1}{1-\gamma}\mathbb{E}_{s\sim d_{s_0}^{x_k,\bar{y}_{t+1}}} \langle \mathbb{E}_{a\sim x_k(\cdot|s)} Q^{x_k,\bar{y}_t}(s,a,\cdot), \bar{y}_{t+1}(\cdot|s) - \bar{y}_t(\cdot|s) \rangle$$

$$= \frac{1}{1-\gamma}\mathbb{E}_{s\sim d_{s_0}^{x_k,\bar{y}_{t+1}}} \langle \nu_{\star,t}(s,\cdot), \bar{y}_{t+1}(\cdot|s) - \bar{y}_t(\cdot|s) \rangle$$

$$= \frac{1}{1-\gamma}\mathbb{E}_{s\sim d_{s_0}^{x_k,\bar{y}_{t+1}}} \Big[ \langle \nu_{t+1}(s,\cdot), \bar{y}_{t+1}(\cdot|s) - \bar{y}_t(\cdot|s) \rangle$$

$$+ \langle \nu_{\star,t}(s,\cdot) - \nu_{t+1}(s,\cdot), \bar{y}_{t+1}(\cdot|s) - \bar{y}_t(\cdot|s) \rangle \Big]$$

$$\geq \frac{1}{1-\gamma}\mathbb{E}_{s\sim d_{s_0}^{x_k,\bar{y}_{t+1}}} \Big[ \langle \nu_{t+1}(s,\cdot), \bar{y}_{t+1}(\cdot|s) - \bar{y}_t(\cdot|s) \rangle - \frac{\eta}{2}\|\nu_{\star,t}(s,\cdot) - \nu_{t+1}(s,\cdot)\|_\infty^2$$

$$- \frac{1}{2\eta}\|\bar{y}_{t+1}(\cdot|s) - \bar{y}_t(\cdot|s)\|_1^2 \Big], \tag{46}$$

where the last step uses Cauchy-Schwarz and Young's inequalities.

Plugging in $\bar{y} = \bar{y}_t$ in (44) and using Equation (8) gives

$$-\eta\langle \nu_{t+1}(s,\cdot), \bar{y}_t(\cdot|s) - \bar{y}_{t+1}(\cdot|s) \rangle \geq D(\bar{y}_t(\cdot|s), \bar{y}_{t+1}(\cdot|s)) + D(\bar{y}_{t+1}(\cdot|s), \bar{y}_t(\cdot|s))$$

$$\geq \|\bar{y}_{t+1}(\cdot|s) - \bar{y}_t(\cdot|s)\|_1^2,$$

which implies that $\langle \nu_{t+1}(s,\cdot), \bar{y}_{t+1}(\cdot|s) - \bar{y}_t(\cdot|s) \rangle - \frac{1}{2\eta}\|\bar{y}_{t+1}(\cdot|s) - \bar{y}_t(\cdot|s) \geq 0$.

Recall that $d_{s_0}^{x_k,\bar{y}{t+1}}(s) = (1-\gamma)\sum_{t=0}^\infty \gamma^t \Pr^{x_k,\bar{y}_{t+1}}(s_t = s|s_0)$, therefore $1 - \gamma \leq d_{s_0}^{x_k,\bar{y}{t+1}}(s_0) \leq 1$. Using the two previous inequalities in (46) gives

$$\langle \nu_{t+1}(s,\cdot), \bar{y}_{t+1}(\cdot|s) - \bar{y}_t(\cdot|s) \rangle \leq V^{x_k,\bar{y}_{t+1}}(s) - V^{x_k,\bar{y}_t}(s) + \frac{1}{2\eta}\|\bar{y}_{t+1}(\cdot|s) - \bar{y}_t(\cdot|s)\|_1^2$$

$$+ \frac{\eta}{2(1-\gamma)}\|\nu_{\star,t} - \nu_{t+1}\|_\infty^2.$$

We use the final inequality, (45), and (8) in (44) to get

$$\eta\langle \nu_{\star,t}(s,\cdot), \bar{y}(\cdot|s) - \bar{y}_t(\cdot|s) \rangle + D(\bar{y}(\cdot|s), y_{t+1}(\cdot|s)) - \eta V^{x_k,\bar{y}_{t+1}}(s) \leq D(\bar{y}(\cdot|s), y_t(\cdot|s))$$

$$- \eta V^{x_k,\bar{y}_t}(s) + \frac{\eta^2}{2(1-\gamma)}\|\nu_{\star,t} - \nu_{t+1}\|_\infty^2 + \eta\langle \nu_{\star,t}(s,\cdot) - \nu_{t+1}(s,\cdot), \bar{y}(\cdot|s) - \bar{y}_t(\cdot|s) \rangle. \tag{47}$$

In view of the definition $\nu_{\star,t} = \mathbb{E}_{a\sim x_k(\cdot|s)} Q^{x_k,\bar{y}_t}(\cdot,a,\cdot)$, performance difference lemma gives $(1 - \gamma)(V^{x_k,y_k^*}(s_0) - V^{x_k,\bar{y}_t}(s_0)) = \mathbb{E}_{s\sim d_{s_0}^{x_k,y_k^*}}\langle \nu_{\star,t}(s,\cdot), y_k^*(\cdot|s) - y_t(\cdot|s) \rangle$. Plugging in $y = y_k^*$ in (47) and taking $\mathbb{E}_{s\sim d_{s_0}^{x_k,y_k^*}}$ of both sides give

$$\eta(1-\gamma)(V^{x_k,y_k^*}(s_0) - V^{x_k,\bar{y}_t}(s_0)) + \mathbb{E}_{s\sim d_{s_0}^{x_k,y_k^*}} \Big[ D(y_k^*(\cdot|s), \bar{y}_{t+1}(\cdot|s)) - \eta V^{x_k,\bar{y}_{t+1}}(s) \Big]$$

$$\leq \mathbb{E}_{s\sim d_{s_0}^{x_k,y_k^*}} \Big[ D(y_k^*(\cdot|s), \bar{y}_t(\cdot|s)) - \eta V^{x_k,\bar{y}_t}(s) + \frac{\eta^2}{2(1-\gamma)}\|\nu_{\star,t} - \nu_{t+1}\|_\infty^2$$

$$+ \eta\langle \nu_{\star,t}(s,\cdot) - \nu_{t+1}(s,\cdot), y_k^*(\cdot|s) - \bar{y}_t(\cdot|s) \rangle \Big]. \tag{48}$$

Now we take expectation w.r.t. the randomness in the algorithm, use tower property, the fact that conditioned on $\bar{y}_t$, $y_k^*(\cdot|s) - \bar{y}_t(\cdot|s)$ is deterministic, Cauchy-Schwarz inequality, $\bar{y}(\cdot|s) \in \Delta$ for any $\bar{y}, s$. After those steps, we note that $d_{s_0}^{x_k,y_k^*}$ does not depend on $t$ and sum the inequality over $t$ to get the result. $\qquad\square$

---

**Algorithm 8** Single loop NAC with a game etiquette and $\zeta$-greedy exploration

---

**Require:** $\hat{V}_0$ such that $\|\hat{V}_0 - V^{x_0,y_0}\|_\infty^2 \le \epsilon$
  **for** $k = 1, 2, \dots$ **do**
    **Stage 1**
    **for** $t = 0, 1, \dots, T-1$ **do**
      Sample $(s_t, a_t, b_t, s_{t+1})$, with policy pair $\hat{x}_t, \hat{y}_t$ observe $s_t, b_t, r(s_t, a_t, b_t), s_{t+1}$
      $\hat{\theta}_{t+1}^x = \hat{\theta}_t^x - \beta_t e(s_t, a_t)\big(\hat{\theta}_t^x(s_t, a_t) - r(s_t, a_t, b_t) - \gamma\hat{V}_{k-1}(s_{t+1})\big)$
      $\hat{\theta}_{t+1}^y = \hat{\theta}_t^y - \beta_t e(s_t, b_t)\big(\hat{\theta}_t^x(s_t, b_t) - r(s_t, a_t, b_t) - \gamma\hat{V}_{k-1}(s_{t+1})\big)$
      $[x_{t+1}(\cdot|s),\ y_{t+1}(\cdot|s)] = [\mathcal{P}(x_t(\cdot|s), \eta\hat{\theta}_{t+1}^x(s, \cdot)),\ \mathcal{P}(y_t(\cdot|s), -\eta\hat{\theta}_{t+1}^y(s, \cdot))]$
    Output $x_k = \frac{1}{T}\sum_{t=1}^T x_t$.
    **Stage 2**
    **for** $t = 0, 1, \dots, T-1$ **do**
      Sample $(s_t, a_t, b_t, s_{t+1}, b_{t+1})$ with policy pair $\hat{x}_k, \hat{\bar{y}}_t$, observe $s_t, b_t, r(s_t, a_t, b_t), s_{t+1}, b_{t+1}$
      $\hat{\nu}_{t+1} = \hat{\nu}_t - \beta_t^\nu e(s_t, b_t)\,(\hat{\nu}_t(s_t, b_t) - r(s_t, a_t, b_t) - \gamma\hat{\nu}_t(s_{t+1}, b_{t+1}))$
      $\hat{\omega}_{t+1} = \hat{\omega}_t - \beta_t^\omega e(s_t)\,(\hat{\omega}_t(s_t) - r(s_t, a_t, b_t) - \gamma\hat{\omega}_t(s_{t+1}))$
      $\bar{y}_{t+1}(\cdot|s) = \mathcal{P}(\bar{y}_t(\cdot|s), -\eta\hat{\nu}_{t+1}(s, \cdot))$
    Output $y_k = \bar{y}_{\hat{t}}$, $\hat{V}_k = \hat{\omega}_{\hat{t}+1}$, where $\hat{t} \in [T]$ is selected uniformly at random and $V_k = V^{x_k, y_k}$.

---

# G    GREEDY EXPLORATION TO REMOVE ASSUMPTION 2

## G.1    SINGLE LOOP NAC WITH A GAME ETIQUETTE AND $\zeta$-GREEDY EXPLORATION

**Remark G.1.** *In this section, we see how to avoid Assumption 2 as mentioned in the main text. Essentially the idea is similar to Khodadadian et al. (2021b) and (Lan, 2021, Remark 1) in the single agent case. We are going to use $\zeta$ greedy to avoid Assumption 2.*

Let us define the modified policies with greedy exploration

$$\hat{x}_t(a|s) = (1 - \zeta)x_t(a|s) + \frac{\zeta}{|A|}, \quad \hat{y}_t(b|s) = (1 - \zeta)y_t(b|s) + \frac{\zeta}{|B|}.$$

Now we are going to sample with the $\hat{x}_t, \hat{y}_t$ and the algorithm will read as Algorithm 8.

### G.1.1    STAGE 1 OF SINGLE LOOP NAC WITH A GAME ETIQUETTE AND $\zeta$-GREEDY EXPLORATION (SEE ALGORITHM 8)

Let us recall the notation and introduce more notations. At iteration $k$, we solve

$$Q^s(a, b) = Q(s, a, b) = r(s, a, b) + \gamma\sum_{s'} P(s'|s, a, b)V^{x_{k-1}, y_{k-1}}(s').$$

The problem is to find $x_{out}, y_{out}$ such that

$$\mathbb{E}\max_{x^s, y^s} x_{out}^s Q^s y^s - x^s Q^s y_{out}^s.$$

Now we define

$$\hat{Q}^s(a, b) = \hat{Q}(s, a, b) = r(s, a, b) + \gamma\sum_{s'} P(s'|s, a, b)V^{\hat{x}_{k-1}, \hat{y}_{k-1}}(s'),$$

as we sample with greedy policies, at the stage 2 we will learn $V^{\hat{x}_{k-1}, \hat{y}_{k-1}}$. So we will have the oracle $\hat{V}_{k-1}$ such that $\|\hat{V}_{k-1} - V^{\hat{x}_{k-1}, \hat{y}_{k-1}}\|$ is small.

In this step, with the abovementioned oracle, we expect to learn

$$\hat{\theta}_{\star, t}^x(s, a) = \mathbb{E}_{b \sim \hat{y}_t^s}\hat{Q}^s(a, b), \qquad \hat{\theta}_{\star, t}^y(s, b) = \mathbb{E}_{a \sim \hat{x}_t^s}\hat{Q}^s(a, b).$$

Let us continue with $x$-player and drop the superscript,

$$\hat{\theta}_{\star,t}(s,a) = \mathbb{E}_{b\sim\hat{y}_t(\cdot|s)}\hat{Q}^s(a,b)$$
$$= \sum_b \hat{y}_t^s \hat{Q}^s(a,b)$$
$$= \sum_b \hat{y}_t(b|s)r(s,a,b) + \gamma \sum_{b,s'} \hat{y}_t(b|s)P(s'|s,a,b)V^{\hat{x}_{k-1},\hat{y}_{k-1}}(s').$$

Therefore the operator is

$$\hat{F}_t(\hat{\theta}_t)(s,a) = \rho^{\hat{x}_t,\hat{y}_t}(s)\hat{x}_t(a|s)\left(\hat{\theta}_t(s,a) - \sum_b \hat{y}_t(b|s)r(s,a,b)\right.$$
$$\left. - \gamma \sum_{s',b} \hat{y}_t(b|s)P(s'|s,a,b)V^{\hat{x}_{k-1},\hat{y}_{k-1}}(s)\right). \quad (49)$$

As $\hat{x}_t(\cdot|s) \geq \frac{\zeta}{|A|}$ and $\hat{y}_t(\cdot|s) \geq \frac{\zeta}{|B|}$ by definition, we have that $\hat{F}_t$ is strongly monotone with $\underline{\rho}\frac{\zeta}{|A|}$ where $\underline{\rho}$ is the lower bound given in Assumption 1, which holds when the induced Markov chain is aperiodic and irreducible. Let us call this $\hat{\lambda}_{\min}^\theta = \underline{\rho}\frac{\zeta}{|A|}$.

We state the main result which follows by the results we will prove afterwards.

**Theorem G.2.** *Let Assumption 1 hold. By combining Lemma G.11, Lemma G.4 and their corresponding corollaries, we can show $\tilde{\mathcal{O}}(\epsilon^{-7})$ sample complexity for Algorithm 8.*

*Proof.* Insert Corollary G.6 and lemma G.11 into (14) to get the result. □

**Remark G.3.** *In single agent setting, Khodadadian et al. (2021b) obtains $\tilde{\mathcal{O}}(\epsilon^{-6})$ with greedy exploration, to avoid Assumption 2. Our estimate is an $\epsilon$ factor away from this rate.*

Below, we analyze the actor in the stage 1 of Algorithm 8.

**Lemma G.4.** *Let Assumption 1 hold and $\eta_t = \frac{1}{(t+7)^{6/7}}$, $\zeta = \frac{1}{T^{1/7}}$. Then, for the stage 1 of Algorithm 8*

$$\mathbb{E}\max\frac{1}{T}\sum_{t=1}^T \langle\mathbb{E}_{b\sim y_t^s}Q(s,\cdot,b), x_t^s - x^s\rangle \leq \frac{\log|A|}{T\eta_1} + \frac{\log|A|}{T\eta_T} + 2\frac{1}{T}\sum_{t=1}^T \mathbb{E}\|\hat{\theta}_{t+1}^s - \hat{\theta}_{\star,t}^s\|_\infty$$
$$+ \frac{4\xi|B|}{(1-\gamma)} + \frac{8\gamma\xi(|A|\vee|B|)}{(1-\gamma)^2} + \frac{\sum_{t=1}^T \eta_t}{2T(1-\gamma)^2}.$$

**Corollary G.5.** *By using $\eta_t, \zeta$, we get the bound*

$$\mathcal{O}\left(\frac{|A|\vee|B|}{(1-\gamma)^2 T^{1/7}} + \frac{1}{T^{6/7}} + \frac{1}{T}\sum_{t=1}^T \|\hat{\theta}_{t+1} - \hat{\theta}_{\star,t}\|_\infty\right).$$

**Corollary G.6.** *We can plug in the bound of $\mathbb{E}\|\hat{\theta}_{t+1} - \hat{\theta}_{\star,t}\|^2$ (see Corollary G.8 ) and use the same bound for the other player to get*

$$\mathbb{E}\max_{x^s,y^s} x_{out}Q^s y^s - x^s Q^s y_{out} \leq \tilde{\mathcal{O}}\left(\frac{|A|\vee|B|}{(1-\gamma)^2 T^{1/7}} + \frac{\sqrt{|S||A|}}{T^{1.5/7}} + \frac{|S||A|}{T^{2/7}(1-\gamma)^2} + \frac{1}{T^{1/7}(1-\gamma)}\right) +$$
$$\mathcal{O}(\sqrt{|S||A|}T^{1/7})\mathbb{E}\|\hat{V}_{k-1} - V^{\hat{x}_{k-1},\hat{y}_{k-1}}\|_\infty. \quad (50)$$

*Plugging in the bound for the final term from Lemma G.10*

$$\mathbb{E}\max_{x^s,y^s} x_{out}Q^s y^s - x^s Q^s y_{out} \leq \tilde{\mathcal{O}}\left(\frac{|A|\vee|B|}{(1-\gamma)^2 T^{1/7}} + \frac{\sqrt{|S||A|}}{T^{1.5/7}} + \frac{|S||A|}{T^{2/7}(1-\gamma)^2} + \frac{1}{T^{1/7}(1-\gamma)}\right) +$$
$$\mathcal{O}\left(\frac{|S|\sqrt{|A|}}{(1-\gamma)^{1.5}\underline{\rho}T^{1/7}}\right), \quad (51)$$

*which gives the $\mathcal{O}(\epsilon^{-7})$ complexity for stage 1.*

*Proof of Lemma G.4.* Let us denote $x_t^s = x_t(\cdot|s)$ and similarly for variables $y, \hat{x}_t, \hat{y}_t, \theta$. Let us recall (24) with our new notations

$$x_{out}^s Q^s y^s - x^s Q^s y_{out}^s = \frac{1}{T}\sum_{t=1}^{T}\langle \mathbb{E}_{a\sim x_t^s}Q(s,a,\cdot), y^s\rangle - \frac{1}{T}\sum_{t=1}^{T}\langle \mathbb{E}_{b\sim y_t^s}Q(s,\cdot,b), x^s\rangle$$

$$= \frac{1}{T}\sum_{t=1}^{T}\left[\langle \mathbb{E}_{a\sim x_t^s}Q(s,a,\cdot), y^s - y_t^s\rangle - \langle \mathbb{E}_{b\sim y_t^s}Q(s,\cdot,b), x^s - x_t^s\rangle\right]. \quad (52)$$

We have to convert this to the game when the matrix is $\hat{Q}$

$$x_{out}^s Q^s y^s - x^s Q^s y_{out}^s = \frac{1}{T}\sum_{t=1}^{T}\left[\langle \mathbb{E}_{a\sim x_t^s}\hat{Q}^s(a,\cdot), y^s - y_t^s\rangle - \langle \mathbb{E}_{b\sim y_t^s}\hat{Q}^s(\cdot,b), x^s - x_t^s\rangle\right]$$

$$+ \frac{1}{T}\sum_{t=1}^{T}\langle \mathbb{E}_{a\sim x_t^s}[Q^s(a,\cdot) - \hat{Q}^s(a,\cdot)], y^s - y_t^s\rangle - \langle \mathbb{E}_{b\sim y_t^s}[Q^s(\cdot,b) - \hat{Q}^s(a,\cdot)], x^s - x_t^s\rangle.$$

For the error terms note

$$\langle \mathbb{E}_{a\sim x_t^s}[Q^s(a,\cdot) - \hat{Q}^s(a,\cdot)], y^s - y_t^s\rangle \leq 2\|\mathbb{E}_{a\sim x_t^s}[Q^s(a,\cdot) - \hat{Q}^s(a,\cdot)]\|_\infty$$

$$\leq 2\|Q^s - \hat{Q}^s\|_\infty$$

$$\leq 2\gamma \max_{a,b}|\sum_{s'}P(s'|s,a,b)(V^{x_{k-1},y_{k-1}}(s') - V^{\hat{x}_{k-1},\hat{y}_{k-1}}(s'))|$$

$$\leq 2\gamma\|V^{x_{k-1},y_{k-1}} - V^{\hat{x}_{k-1},\hat{y}_{k-1}}\|_\infty$$

$$\leq \frac{4\gamma\zeta(|A|\vee|B|)}{(1-\gamma)^2}, \quad (53)$$

where the last step is due to the Lipschitzness of the value function due to performance difference lemma and that the policies $\hat{x}_{k-1}, \hat{y}_{k-1}$ and $x_{k-1}, y_{k-1}$ differ at most by $\zeta$.

With the similar estimation for the other error term, we have

$$x_{out}^s Q^s y^s - x^s Q^s y_{out}^s = \frac{1}{T}\sum_{t=1}^{T}\left[\langle \mathbb{E}_{a\sim x_t^s}\hat{Q}^s(a,\cdot), y^s - y_t^s\rangle - \langle \mathbb{E}_{b\sim y_t^s}\hat{Q}^s(\cdot,b), x^s - x_t^s\rangle\right]$$

$$+ \frac{8\zeta(|A|\vee|B|)}{(1-\gamma)^2}. \quad (54)$$

Let us denote

$$\theta_{\star,t}^s = \mathbb{E}_{b\sim y_t^s}\hat{Q}^s(\cdot,b). \quad (55)$$

Note that this notation is not consistent with previous sections. Here by $\theta_{\star,t}$ we mean the quantity we need to upper bound (54).

By the update rule of $x_t$, for any $x^s$

$$D(x^s, x_t^s) \leq D(x^s, x_{t+1}^s) - D(x_{t+1}^s, x_t^s) + \eta_t\langle\hat{\theta}_{t+1}^s, x^s - x_{t+1}^s\rangle.$$

Let us estimate the inner product

$$\eta_t\langle\hat{\theta}_{t+1}^s, x^s - x_{t+1}^s\rangle = \eta_t\langle\hat{\theta}_{t+1}^s, x^s - x_t^s\rangle + \eta_t\langle\hat{\theta}_{t+1}^s, x_t^s - x_{t+1}^s\rangle$$

$$= \eta_t\langle\theta_{\star,t}^s, x^s - x_t^s\rangle + \eta_t\langle\hat{\theta}_{t+1}^s - \theta_{\star,t}^s, x^s - x_t^s\rangle + \eta_t\langle\hat{\theta}_{t+1}^s, x_t^s - x_{t+1}^s\rangle$$

As we will be actually learning $\hat{\theta}_{\star,t}^s$, we will further manipulate the inner product

$$\eta_t\langle\hat{\theta}_{t+1}^s, x^s - x_{t+1}^s\rangle = \eta_t\langle\theta_{\star,t}^s, x^s - x_t^s\rangle + \eta_t\langle\hat{\theta}_{t+1}^s - \hat{\theta}_{\star,t}^s, x^s - x_t^s\rangle + \eta_t\langle\hat{\theta}_{\star,t}^s - \theta_{\star,t}^s, x^s - x_t^s\rangle$$

$$+ \eta_t\langle\hat{\theta}_{t+1}^s, x_t^s - x_{t+1}^s\rangle.$$

Let us estimate the third inner product on RHS

$$
\begin{aligned}
\eta_t \langle \hat{\theta}^s_{\star,t} - \theta^s_{\star,t}, x^s - x^s_t \rangle &\leq 2\eta_t \|\hat{\theta}^s_{\star,t} - \theta^s_{\star,t}\|_\infty \\
&= 2\eta_t \max_a | \sum_b (y_t(b|s) - \hat{y}_t(b|s))\hat{Q}^s(a,b)| \\
&\leq \frac{4\eta_t}{1-\gamma} \sum_b |y_t(b|s) - \hat{y}_t(b|s)| \\
&= \frac{4\eta_t}{1-\gamma} \sum_b |\zeta(1/|B| - y_t(b|s))| \\
&\leq \frac{4\eta_t |B|\zeta}{1-\gamma}
\end{aligned}
\tag{56}
$$

We use this estimation and Cauchy-Schwarz and Young's inequalities to derive

$$
\begin{aligned}
\eta_t \langle \hat{\theta}^s_{t+1}, x^s - x^s_{t+1} \rangle \leq{}& \eta_t \langle \theta^s_{\star,t}, x^s - x^s_t \rangle + 2\eta_t \|\hat{\theta}^s_{t+1} - \hat{\theta}^s_{\star,t}\|_\infty + \frac{4\eta_t |B|\zeta}{1-\gamma} \\
&+ \frac{\eta_t^2}{2}\|\hat{\theta}_{t+1}\|^2_\infty + \frac{1}{2}\|x^s_t - x^s_{t+1}\|^2_1.
\end{aligned}
$$

We use this estimate in the main inequality with strong convexity of Bregman distance to get

$$
\eta_t \langle \theta^s_{\star,t}, x^s_t - x^s \rangle + D(x^s, x^s_t) \leq D(x^s, x^s_{t+1}) + 2\eta_t \|\hat{\theta}^s_{t+1} - \hat{\theta}^s_{\star,t}\|_\infty + \frac{4\eta_t \zeta|B|}{1-\gamma} + \frac{\eta_t^2}{2(1-\gamma)^2}.
$$

We divide both sides by $\eta_t$ to get

$$
\langle \theta^s_{\star,t}, x^s_t - x^s \rangle + \frac{1}{\eta_t} D(x^s, x^s_t) \leq \frac{1}{\eta_t} D(x^s, x^s_{t+1}) + 2\|\hat{\theta}^s_{t+1} - \hat{\theta}^s_{\star,t}\|_\infty + \frac{4\zeta|B|}{1-\gamma} + \frac{\eta_t}{2(1-\gamma)^2}.
$$

We sum this inequality and use (55) to get

$$
\begin{aligned}
\frac{1}{T}\sum_{t=1}^T \langle \mathbb{E}_{b\sim y^s_t}\hat{Q}^s(\cdot, b), x^s_t - x^s \rangle \leq{}& \frac{\log|A|}{T\eta_1} + \frac{\log|A|}{T\eta_T} + 2\frac{1}{T}\sum_{t=1}^t \|\hat{\theta}^s_{t+1} - \hat{\theta}^s_{\star,t}\|_\infty \\
&+ \frac{4\zeta|B|}{(1-\gamma)} + \frac{\sum_{t=1}^T \eta_t}{2T(1-\gamma)^2}.
\end{aligned}
$$

$\square$

We next analyze the critic in stage 1 of Algorithm 8.

**Lemma G.7.** *Let Assumption 1 hold and* $\eta_t = \frac{1}{(t+7)^{6/7}}$, $\zeta = \frac{1}{T^{1/7}}$, $\beta_t = \frac{2}{(t+7)^{4/7}\hat{\lambda}^\theta_{\min}}$. *Then, for the critic of stage 1 of Algorithm 8*

$$
\begin{aligned}
\frac{1}{T}\sum_{t=1}^T \mathbb{E}\|\hat{\theta}_t - \hat{\theta}_{\star,t-1}\|_2^2 \leq{}& \mathcal{O}\left(\frac{|S||A|}{T^{3/7}}\right) + \mathcal{O}\left(\frac{|S|^2|A|^2}{T^{4/7}(1-\gamma)^4}\right) + \mathcal{O}\left(\frac{1}{T^{4/7}(\hat{\lambda}^\theta_{\min})^2(1-\gamma)^2}\right) \\
&+ \frac{8|S||A|}{(\hat{\lambda}^\theta_{\min})^2}\|\hat{V}_{k-1} - V^{\hat{x}_{k-1},\hat{y}_{k-1}}\|_\infty^2.
\end{aligned}
\tag{57}
$$

**Corollary G.8.** *Recall that* $\zeta = \frac{1}{T^{1/7}}$ *and* $\hat{\lambda}^\theta_{\min} = \varrho\zeta/|A|$,

$$
\begin{aligned}
\frac{1}{T}\sum_{t=1}^T \mathbb{E}\|\hat{\theta}_t - \hat{\theta}_{\star,t-1}\|_2^2 \leq{}& \mathcal{O}\left(\frac{|S||A|}{T^{3/7}}\right) + \mathcal{O}\left(\frac{|S|^2|A|^2}{T^{4/7}(1-\gamma)^4}\right) + \mathcal{O}\left(\frac{1}{T^{2/7}(1-\gamma)^2}\right) \\
&+ 8|S||A|T^{2/7}\|\hat{V}_{k-1} - V^{\hat{x}_{k-1},\hat{y}_{k-1}}\|_\infty^2.
\end{aligned}
\tag{58}
$$

*Proof of Lemma G.7.* As we are sampling with $\hat{x}_t, \hat{y}_t$ and learning $\hat{\theta}_{\star,t}(s,a) = \mathbb{E}_{b \sim \hat{y}_t^s} \hat{Q}^s(a,b)$, we have that (see (49))

$$\hat{F}_t(\hat{\theta}_{\star,t}) = 0.$$

We also denote

$$\tilde{F}(\hat{\theta}_t, \xi_t) = e(s_t, a_t)(\hat{\theta}_t(s_t, a_t) - r(s_t, a_t, b_t) - \gamma \hat{\tilde{V}}_{k-1}),$$

where $s_t, a_t, b_t, s_{t+1}$ are sampled according to $\hat{x}_t, \hat{y}_t$.

Let us derive

$$\|\hat{\theta}_{t+1} - \hat{\theta}_{\star,t}\|_2^2 = \|\hat{\theta}_t - \hat{\theta}_{\star,t}\|_2^2 - 2\beta_t \langle \tilde{F}(\hat{\theta}_t, \xi_t), \hat{\theta}_t - \hat{\theta}_{\star,t}\rangle + \beta_t^2 \|\tilde{F}(\hat{\theta}_t, \xi_t)\|_2^2.$$

Now, by i.i.d. sampling $s_t \sim \rho^{\hat{x}_t, \hat{y}_t}$, we have $\mathbb{E}_{\xi_t} \tilde{F}_t(\hat{\theta}_t, \xi_t) = \hat{F}_t(\hat{\theta}_t) + \gamma P_{\hat{x}_t, \hat{y}_t}(\hat{\tilde{V}}_{k-1} - V^{\hat{x}_{k-1}, \hat{y}_{k-1}})$ as in (18). We have by strong monotonicity and $\hat{F}_t(\hat{\theta}_{\star,t}) = 0$,

$$2\beta_t \mathbb{E}_{\xi_t} \langle \tilde{F}(\hat{\theta}_t, \xi_t), \hat{\theta}_t - \hat{\theta}_{\star,t}\rangle = 2\beta_t \langle \hat{F}(\hat{\theta}_t), \hat{\theta}_t - \hat{\theta}_{\star,t}\rangle = 2\beta_t \langle \hat{F}(\hat{\theta}_t) - \hat{F}_t(\hat{\theta}_{\star,t}), \hat{\theta}_t - \hat{\theta}_{\star,t}\rangle$$
$$\geq 2\beta_t \hat{\lambda}_{\min}^\theta \|\hat{\theta}_t - \theta_{\star,t}\|_2^2.$$

The recursion becomes

$$\mathbb{E}_{\xi_t} \|\hat{\theta}_{t+1} - \hat{\theta}_{\star,t}\|_2^2 \leq \left(1 - 2\beta_t \hat{\lambda}_{\min}^\theta\right) \|\hat{\theta}_t - \hat{\theta}_{\star,t}\|_2^2$$
$$- 2\gamma \beta_t \langle P_{\hat{x}_t, \hat{y}_t}(\hat{\tilde{V}}_{k-1} - V^{\hat{x}_{k-1}, \hat{y}_{k-1}}), \theta_t - \theta_{\star,t}\rangle + \beta_t^2 \mathbb{E}_{\xi_t} \|\tilde{F}(\hat{\theta}_t, \xi_t)\|_2^2.$$

By Cauchy-Schwarz and Young's inequalities for the inner product

$$-2\gamma \beta_t \langle P_{\hat{x}_t, \hat{y}_t}(\hat{\tilde{V}}_{k-1} - V^{\hat{x}_{k-1}, \hat{y}_{k-1}}), \hat{\theta}_t - \hat{\theta}_{\star,t}\rangle \leq 2\gamma \beta_t \|P_{\hat{x}_t, \hat{y}_t}(\hat{\tilde{V}}_{k-1} - V^{\hat{x}_{k-1}, \hat{y}_{k-1}})\|_2 \|\hat{\theta}_t - \hat{\theta}_{\star,t}\|_2$$
$$\leq \frac{\beta_t \hat{\lambda}_{\min}^\theta}{2} \|\hat{\theta}_t - \hat{\theta}_{\star,t}\|_2^2 + \frac{\beta_t 8\gamma^2 |S||A|}{\hat{\lambda}_{\min}^\theta} \|\hat{\tilde{V}}_{k-1} - V^{\hat{x}_{k-1}, \hat{y}_{k-1}}\|_\infty^2.$$

The recursion then becomes

$$\mathbb{E}_{\xi_t} \|\hat{\theta}_{t+1} - \hat{\theta}_{\star,t}\|_2^2 \leq \left(1 - 3\beta_t \hat{\lambda}_{\min}^\theta/2\right) \|\hat{\theta}_t - \hat{\theta}_{\star,t}\|_2^2$$
$$+ \frac{\beta_t 8\gamma^2 |S||A|}{\hat{\lambda}_{\min}^\theta} \|\hat{\tilde{V}}_{k-1} - V^{\hat{x}_{k-1}, \hat{y}_{k-1}}\|_\infty^2 + \beta_t^2 \mathbb{E}_{\xi_t} \|\tilde{F}(\hat{\theta}_t, \xi_t)\|_2^2.$$

By Young's inequality,

$$\mathbb{E}_{\xi_t} \|\hat{\theta}_{t+1} - \hat{\theta}_{\star,t}\|_2^2 \leq \left(1 - \beta_t \hat{\lambda}_{\min}^\theta\right) \|\hat{\theta}_t - \hat{\theta}_{\star,t-1}\|_2^2 + \frac{6}{\beta_t \hat{\lambda}_{\min}^\theta} \|\hat{\theta}_{\star,t} - \hat{\theta}_{\star,t-1}\|_2^2$$
$$+ \frac{\beta_t 8\gamma^2 |S||A|}{\hat{\lambda}_{\min}^\theta} \|\hat{\tilde{V}}_{k-1} - V^{\hat{x}_{k-1}, \hat{y}_{k-1}}\|_\infty^2 + \beta_t^2 \mathbb{E}_{\xi_t} \|\tilde{F}(\hat{\theta}_t, \xi_t)\|_2^2.$$

Now we have to bound $\|\hat{\theta}_{\star,t} - \hat{\theta}_{\star,t-1}\|_2^2$:

$$\|\hat{\theta}_{\star,t} - \hat{\theta}_{\star,t-1}\|_2 \leq \sqrt{|S||A|} \|\hat{\theta}_{\star,t} - \hat{\theta}_{\star,t-1}\|_\infty$$
$$= \sqrt{|S||A|} \max_{a,s} |\mathbb{E}_{b \sim \hat{y}_t^s} \hat{Q}^s(a,b) - \mathbb{E}_{b \sim \hat{y}_{t-1}^s} \hat{Q}^s(a,b)|$$
$$\leq 2 \frac{\sqrt{|S||A|}}{1-\gamma} \max_s \|\hat{y}_t^s - \hat{y}_{t-1}^s\|_1$$
$$\leq 2 \frac{\sqrt{|S||A|}}{1-\gamma} \max_s \|y_t^s - y_{t-1}^s\|_1,$$

where we used $\hat{y}_t = (1-\zeta)y_t + \zeta/|B|$. To show that $\max_s \|y_t^s - y_{t-1}^s\|_1$ is small, we use Lemma B.5 and Lemma B.1 and have the final bound for this quantity

$$\|\hat{\theta}_{\star,t} - \hat{\theta}_{\star,t-1}\|_2^2 \leq 4 \frac{|S|^2 |A|^2}{(1-\gamma)^4} \eta_t^2.$$

By also bounding $\|\tilde{F}(\hat{\theta}_t, \xi_t)\|_2^2$ by Lemma B.1 and taking total expectation, the main recursion becomes

$$\mathbb{E}\|\hat{\theta}_{t+1} - \hat{\theta}_{\star,t}\|_2^2 \leq (1 - \beta_t \hat{\lambda}_{\min}^\theta)\mathbb{E}\|\hat{\theta}_{\star,t} - \hat{\theta}_{\star,t-1}\|_2^2 + \frac{24|S|^2|A|^2\eta_t^2}{\beta_t \hat{\lambda}_{\min}^\theta(1-\gamma)^4} + \frac{2\beta_t^2}{(1-\gamma)^2}$$
$$+ \frac{8\beta_t|S||A|}{\hat{\lambda}_{\min}^\theta}\mathbb{E}\|\hat{V}_{k-1} - V^{\hat{x}_{k-1},\hat{y}_{k-1}}\|_\infty^2.$$

This inequality gives

$$\frac{1}{T}\sum_{t=1}^T \mathbb{E}\|\hat{\theta}_t - \hat{\theta}_{\star,t-1}\|_2^2 \leq \frac{1}{T}\sum_{t=1}^T \frac{1}{\beta_t \hat{\lambda}_{\min}^\theta}\left(\|\hat{\theta}_t - \theta_{\star,t-1}\|_2^2 - \|\hat{\theta}_{t+1} - \theta_{\star,t}\|_2^2\right)$$

$$+ \frac{1}{T}\sum_{t=1}^T \frac{24|S|^2|A|^2\eta_t^2}{\beta_t^2(\hat{\lambda}_{\min}^\theta)^2(1-\gamma)^4} + \frac{1}{T}\sum_{t=1}^T \frac{2\beta_t}{\hat{\lambda}_{\min}^\theta(1-\gamma)^2} + \frac{8|S||A|}{(\hat{\lambda}_{\min}^\theta)^2}\|\hat{V}_{k-1} - V^{\hat{x}_{k-1},\hat{y}_{k-1}}\|_\infty^2. \quad (59)$$

Recall that $\eta_t = \frac{1}{(t+7)^{6/7}}$ and $\beta_t = \frac{2}{(t+7)^{4/7}\hat{\lambda}_{\min}^\theta}$. For the summations, we have $\sum_{t=1}^T \eta_t^2/\beta_t^2 = \mathcal{O}(T^{3/7}(\hat{\lambda}_{\min}^\theta)^2)$, $\sum_{t=1}^T \beta_t = \mathcal{O}(T^{3/7}/\hat{\lambda}_{\min}^\theta)$. By using $\beta_{t+1} \leq \beta_t$, and denoting for simplicity $a_t = \|\theta_t - \theta_{\star,t-1}\|_2^2$ so $0 \leq a_t \leq \frac{|S||A|}{(1-\gamma)^2}$ simple estimation gives

$$\sum_{t=1}^T \frac{1}{\beta_t}\left(\|\hat{\theta}_t - \theta_{\star,t-1}\|_2^2 - \|\hat{\theta}_{t+1} - \theta_{\star,t}\|_2^2\right) = \sum_{t=1}^T \frac{1}{\beta_t}(a_t - a_{t+1})$$

$$= \sum_{t=1}^T \left[\frac{1}{\beta_t}a_t - \frac{1}{\beta_{t+1}}a_{t+1} + \left(\frac{1}{\beta_{t+1}} - \frac{1}{\beta_t}\right)a_{t+1}\right]$$

$$\leq \sum_{t=1}^T \left(\frac{1}{\beta_t}a_t - \frac{1}{\beta_{t+1}}a_{t+1} + \frac{|S||A|}{(1-\gamma)^2}\left|\frac{1}{\beta_{t+1}} - \frac{1}{\beta_t}\right|\right)$$

$$= \frac{1}{\beta_1}a_1 - \frac{1}{b_{T+1}}a_{T+1} + \frac{|S||A|}{(1-\gamma)^2}\sum_{t=1}^T \left(\frac{1}{\beta_{t+1}} - \frac{1}{\beta_t}\right)$$

$$\leq \frac{1}{\beta_1}a_1 + \frac{|S||A|}{(1-\gamma)^2\beta_{T+1}} = \mathcal{O}\left(\frac{|S||A|T^{4/7}\hat{\lambda}_{\min}^\theta}{(1-\gamma)^2}\right)$$

Plugging these estimations into (59)

$$\frac{1}{T}\sum_{t=1}^T \mathbb{E}\|\hat{\theta}_t - \hat{\theta}_{\star,t-1}\|_2^2 \leq \mathcal{O}\left(\frac{|S||A|}{T^{3/7}}\right) + \mathcal{O}\left(\frac{|S|^2|A|^2}{T^{4/7}(1-\gamma)^4}\right) + \mathcal{O}\left(\frac{1}{T^{4/7}(\hat{\lambda}_{\min}^\theta)^2(1-\gamma)^2}\right)$$

$$+ \frac{8|S||A|}{(\hat{\lambda}_{\min}^\theta)^2}\|\hat{V}_{k-1} - V^{\hat{x}_{k-1},\hat{y}_{k-1}}\|_\infty^2. \quad (60)$$

$\square$

### G.1.2 STAGE 2 OF SINGLE LOOP NAC WITH A GAME ETIQUETTE AND $\zeta$-GREEDY EXPLORATION (SEE ALGORITHM 8)

We are now going to estimate $\|\hat{V}_{k-1} - V^{\hat{x}_{k-1},\hat{y}_{k-1}}\|_\infty^2$. First, we need the following lemma for numerical sequences from Mokhtari et al. (2020).

**Lemma G.9.** *(Lemma 19 in Mokhtari et al. (2020)) Let $b \geq 0$, $c > 1$. Let $\phi_t$ be a sequence of real numbers satisfying*

$$\phi_t \leq \left(1 - \frac{c}{(t+t_0)^\alpha}\right)\phi_{t-1} + \frac{b}{(t+t_0)^{2\alpha}}, \quad (61)$$

*for some $\alpha \in [0, 1]$ and $t_0 \geq 0$. Then $\phi_t$ converges to zero with the rate*

$$\phi_t \leq \frac{\max(\phi_0(t_0+1)^\alpha, b/(c-1))}{(t+t_0+1)^\alpha}.$$

For the stage 2, let $\eta_t = \frac{1}{(t+7)^{6/7}}$ and $\beta_t = \frac{3}{2(t+7)^{4/7}\underline{\rho}}$ and let us analyze the critic for $\hat{V}_{k-1}$.

**Lemma G.10.** *Let Assumption 1 hold. Let $\eta_t = \frac{1}{(t+7)^{6/7}}$ and $\beta_t = \frac{3}{2\underline{\rho}(t+7)^{4/7}}$ and recall $\hat{\omega}_{\hat{t}+1} = \hat{V}_k$ and $\hat{\omega}_{\star,t} = V^{\hat{x}_k, \hat{y}_t}$ and that we take as output $y_k = \bar{y}_{\hat{t}}$ in Algorithm 8*

$$\mathbb{E}\|\hat{V}_k - V^{\hat{x}_k, \hat{y}_k}\|^2 = \frac{1}{T}\sum_{t=1}^{T}\mathbb{E}\|\hat{\omega}_{t+1} - \hat{\omega}_{\star,t}\|^2 \leq \mathcal{O}\left(\frac{|S|}{(1-\gamma)^3\underline{\rho}^2 T^{4/7}}\right). \tag{62}$$

*Proof.* Let us develop similar recursion to previous lemma

$$\|\hat{\omega}_{t+1} - \hat{\omega}_{\star,t}\|_2^2 = \|\hat{\omega}_t - \hat{\omega}_{\star,t}\|_2^2 - 2\beta_t\langle\tilde{F}_t^\omega(\hat{\omega}_t, \xi_t), \hat{\omega}_t - \hat{\omega}_{\star,t}\rangle + \beta_t^2\|\tilde{F}_t^\omega(\hat{\omega}_t, \xi_t)\|_2^2.$$

By taking expectation, using that $s_t \sim \rho^{\hat{x}_k, \hat{y}_t}$, we have $\mathbb{E}_{\xi_t}\tilde{F}_t^\omega(\hat{\omega}_t, \xi_t) = \hat{F}_t^\omega(\hat{\omega}_t)$ and $\hat{F}_t(\omega_t)$ is strongly monotone with $\underline{\rho}$ as it only requires Assumption 1. We also have $\hat{F}_t^\omega(\hat{\omega}_{\star,t}) = 0$. The main recursion becomes

$$\mathbb{E}_{\xi_t}\|\hat{\omega}_{t+1} - \hat{\omega}_{\star,t}\|_2^2 \leq \left(1 - 2\beta_t\underline{\rho}\right)\|\hat{\omega}_t - \hat{\omega}_{\star,t}\|_2^2 + \beta_t^2\mathbb{E}_{\xi_t}\|\tilde{F}_t^\omega(\hat{\omega}_t, \xi_t)\|_2^2.$$

By Young's inequality for $\alpha > 0$

$$(1-2\beta_t\underline{\rho})\|\hat{\omega}_t - \hat{\omega}_{\star,t}\|_2^2 \leq (1-2\beta_t\underline{\rho})(1+\alpha)\|\hat{\omega}_t - \hat{\omega}_{\star,t-1}\|_2^2 + (1-2\beta_t\underline{\rho})(1+1/\alpha)\|\hat{\omega}_{\star,t-1} - \hat{\omega}_{\star,t}\|_2^2.$$

Let us set $\alpha$ such that $(1-2\beta_t\underline{\rho})(1+\alpha) = (1 - \beta_t\underline{\rho})$, which gives the choice $\alpha = \frac{\beta_t\underline{\rho}}{1-2\beta_t\underline{\rho}}$, with the requirement $1 - 2\beta_t\underline{\rho} > 0$. Recall that since $\beta_t = \frac{3}{2\underline{\rho}(t+7)^{4/7}}$, we have $2\beta_t\underline{\rho} < 1$ and $t \geq 0$. Note also that $(1 - 2\beta_t\underline{\rho})(1 + 1/\alpha) \leq \frac{2}{\beta_t\underline{\rho}}$ with out choice of $\alpha$.

The recursion now becomes

$$\mathbb{E}\|\hat{\omega}_{t+1} - \hat{\omega}_{\star,t}\|_2^2 \leq \left(1 - \beta_t\underline{\rho}\right)\mathbb{E}\|\hat{\omega}_t - \hat{\omega}_{\star,t-1}\|_2^2 + \frac{2}{\beta_t\underline{\rho}}\mathbb{E}\|\hat{\omega}_{\star,t-1} - \hat{\omega}_{\star,t}\|_2^2 + \beta_t^2\mathbb{E}\|\tilde{F}_t^\omega(\hat{\omega}_t, \xi_t)\|_2^2.$$

Let us estimate the second term on RHS

$$\begin{aligned}
\|\hat{\omega}_{\star,t-1} - \hat{\omega}_{\star,t}\|_2 &\leq |S|\|\hat{\omega}_{\star,t-1} - \hat{\omega}_{\star,t}\|_\infty \\
&= |S|\|V^{\hat{x}_k, \hat{y}_t} - V^{\hat{x}_k, \hat{y}_{t-1}}\|_\infty \\
&\leq \frac{4|S|}{(1-\gamma)^2}\max_s\|\hat{y}_t^s - \hat{y}_{t-1}^s\|_\infty,
\end{aligned}$$

where the last inequality is by Lemma B.3. To further upper bound this quantity, use the update rule of $\hat{y}_t$, Lemma B.5 and Lemma B.1 to get for all s $\|\bar{y}_t^s - \bar{y}_{t-1}^s\|_1 \leq \frac{\eta_t}{1-\gamma}$. We also use that $\hat{y}_t^s(b) = (1-\zeta)\bar{y}_t^s(b) + \zeta/|B|$

$$\|\hat{\omega}_{\star,t-1} - \hat{\omega}_{\star,t}\|_2 \leq \frac{2|S|\eta_t}{(1-\gamma)^3}$$

Therefore, in the main inequality we have (after also bounding the last term by Lemma B.1)

$$\mathbb{E}\|\hat{\omega}_{t+1} - \hat{\omega}_{\star,t}\|_2^2 \leq \left(1 - \beta_t\underline{\rho}\right)\mathbb{E}\|\hat{\omega}_t - \hat{\omega}_{\star,t-1}\|_2^2 + \frac{4|S|\eta_t^2}{\beta_t\underline{\rho}(1-\gamma)^3} + \frac{2\beta_t^2}{(1-\gamma)^2}. \tag{63}$$

Note that by the definitions of $\beta_t, \eta_t$, we have

$$\beta_t\underline{\rho} = \frac{3}{2(t+7)^{4/7}}, \quad \frac{\eta_t^2}{\beta_t} = \frac{2\underline{\rho}}{3(t+7)^{8/7}}, \quad \beta_t^2 = \frac{9}{4\underline{\rho}^2(t+7)^{8/7}}.$$

Therefore, we have

$$\mathbb{E}\|\hat{\omega}_{t+1} - \hat{\omega}_{\star,t}\|_2^2 \leq \left(1 - \frac{3}{2(t+7)^{4/7}}\right)\mathbb{E}\|\hat{\omega}_t - \hat{\omega}_{\star,t-1}\|_2^2 + \frac{8|S|}{3(1-\gamma)^3(t+7)^{4/7}} + \frac{18}{4\underline{\rho}^2(1-\gamma)^2(t+7)^{8/7}}.$$

Now, we use Lemma G.9 with

$$\phi_t = \mathbb{E}\|\hat{\omega}_{t+1} - \hat{\omega}_{\star,t}\|_2^2$$

$$b = \max\left(\frac{8|S|}{3(1-\gamma)^3}, \frac{18}{4\underline{\rho}^2(1-\gamma)^2}\right) \geq 0$$

$$c = \frac{3}{2} > 1$$

$$t_0 = 7$$

$$\alpha = 4/7,$$

gives

$$\mathbb{E}\|\hat{\omega}_{t+1} - \hat{\omega}_{\star,t}\|_2^2 \leq \mathcal{O}\left(\frac{|S|}{(1-\gamma)^3\underline{\rho}^2(t+7)^{4/7}}\right)$$

Sum the inequality to get

$$\frac{1}{T}\sum_{t=1}^T \mathbb{E}\|\hat{\omega}_{t+1} - \hat{\omega}_{\star,t}\|_2^2 \leq \frac{1}{T}\sum_{t=1}^T \mathcal{O}\left(\frac{|S|}{(1-\gamma)^3\underline{\rho}^2(t+7)^{4/7}}\right)$$

$$\leq \frac{1}{T}\sum_{t=1}^T \mathcal{O}\left(\frac{|S|T^{3/7}}{(1-\gamma)^3\underline{\rho}^2}\right)$$

$$\leq \mathcal{O}\left(\frac{|S|}{(1-\gamma)^3\underline{\rho}^2 T^{4/7}}\right)$$

$$\square$$

The accuracy for the stage 2 will follow a similar proof to single agent settings. We would like to bound

$$V^{x_k, y_k^*} - V^{x_k, \bar{y}_t},$$

by sampling $\hat{x}_k^s(a) = (1 - \zeta)x_k^s(a) + \zeta/|A|$ and $\hat{\bar{y}}_t^s(b) = (1 - \zeta)\bar{y}_t^s(b) + \zeta/|B|$.

Let us define the oracles and the operators

$$\hat{\nu}_{\star,t}(s,b) = \mathbb{E}_{a \sim \hat{x}_k^s} Q^{\hat{x}_k, \hat{\bar{y}}_t}(s,a,b).$$

$$\tilde{F}_t(\hat{\nu}_t, \xi_t)(s,b) = \rho^{\hat{x}_k, \hat{\bar{y}}_t}(s)\hat{\bar{y}}_t(b|s)e(s,b)\left(\hat{\nu}_t(s,b) - r(s,a,b) - \gamma\hat{\nu}_t(s',b')\right),$$

where $s \sim \rho^{\hat{x}_k, \hat{\bar{y}}_t}$, $a \sim \hat{x}_k^s$, $b \sim \hat{\bar{y}}_t^s$, $s' \sim P(\cdot|s,a,b)$, $b' \sim \hat{\bar{y}}_t^{s'}$, and

$$\hat{F}_t(\hat{\nu}_t)(s,b) = \rho^{\hat{x}_k, \hat{\bar{y}}_t}(s)\hat{\bar{y}}_t(b|s)\Big(\hat{\nu}_t(s,b) - \sum_a \hat{x}_k(a|s)r(s,a,b)$$

$$- \gamma\sum_{s',a,b'} P(s'|s,a,b)\hat{x}_k(a|s)\hat{\bar{y}}_t(b'|s')\hat{\nu}_t(s',b')\Big),$$

and therefore $\hat{F}_t(\hat{\nu}_{\star,t}) = 0$ and $\hat{F}_t$ is strongly monotone with $\underline{\rho}\zeta/|B| = \hat{\lambda}_{\min}^\nu$.

**Lemma G.11.** *Let Assumption 1 hold. Let us set $\eta_t = \frac{1}{(t+7)^{6/7}}$, and $\zeta = \frac{1}{T^{1/7}}$. Then for the actor of the stage 2 in Algorithm 8,*

$$\mathbb{E}\frac{1}{T}\sum_{t=1}^T V^{x_k, y_k^*}(s_0) - V^{x_k, \bar{y}_t}(s_0) \leq \tilde{\mathcal{O}}\left(\frac{1}{(1-\gamma)T^{1/7}}\right) + \mathcal{O}\left(\frac{1}{1-\gamma}\mathbb{E}\|\hat{\nu}_{t+1} - \hat{\nu}_{\star,t}\|_\infty\right)$$

$$+ \mathcal{O}\left(\frac{|B|\zeta}{(1-\gamma)^2}\right) + \mathcal{O}\left(\frac{1}{T^{6/7}(1-\gamma)^3}\right)$$

**Corollary G.12.** *By using Lemma G.13, we get the overall rate $\mathcal{O}(1/T^{1/7})$.*

*Proof.* By the update rule of the algorithm,

$$D(y^s, \bar{y}_{t+1}^s) \leq D(y^s, \bar{y}_t^s) - D(\bar{y}_{t+1}^s, \bar{y}_t^s) - \eta_t \langle \hat{\nu}_{t+1}^s, y^s - \bar{y}_{t+1}^s \rangle$$
$$= D(y^s, \bar{y}_t^s) - D(\bar{y}_{t+1}^s, \bar{y}_t^s) - \eta_t \langle \hat{\nu}_{\star,t}^s, y^s - \bar{y}_{t+1}^s \rangle - \eta_t \langle \hat{\nu}_{t+1}^s - \hat{\nu}_{\star,t}^s, y^s - \bar{y}_{t+1}^s \rangle.$$

We now estimate the inner products by Cauchy-Schwarz, Young's inequalities and Lemma B.1

$$-\eta_t \langle \hat{\nu}_{\star,t}^s, y^s - \bar{y}_{t+1}^s \rangle = -\eta_t \langle \hat{\nu}_{\star,t}^s, y^s - \bar{y}_t^s \rangle - \eta_t \langle \hat{\nu}_{\star,t}^s, \bar{y}_t^s - \bar{y}_{t+1}^s \rangle$$

$$\leq -\eta_t \langle \hat{\nu}_{\star,t}^s, y^s - \bar{y}_t^s \rangle + \frac{\eta_t^2}{2} \|\hat{\nu}_{\star,t}^s\|_\infty^2 + \frac{1}{2} \|\bar{y}_t^s - \bar{y}_{t+1}^s\|_1^2$$

$$= -\eta_t \langle \hat{\nu}_{\star,t}^s, y^s - \hat{\bar{y}}_t^s \rangle - \eta_t \langle \hat{\nu}_{\star,t}^s, \hat{\bar{y}}_t^s - \bar{y}_t^s \rangle + \frac{\eta_t^2}{2} \|\hat{\nu}_{\star,t}^s\|_\infty^2 + \frac{1}{2} \|\bar{y}_t^s - \bar{y}_{t+1}^s\|_1^2$$

$$\leq -\eta_t \langle \hat{\nu}_{\star,t}^s, y^s - \hat{\bar{y}}_t^s \rangle + \frac{\eta_t}{1-\gamma} \|\hat{\bar{y}}_t^s - \bar{y}_t^s\|_1 + \frac{\eta_t^2}{2(1-\gamma)^2} + \frac{1}{2} \|\bar{y}_t^s - \bar{y}_{t+1}^s\|_1^2$$

Since $\hat{\bar{y}}_t^s(b) = (1-\zeta)\bar{y}_t^s(b) + \zeta/|B|$, $\|\hat{\bar{y}}_t^s - \bar{y}_t^s\|_1 \leq \mathcal{O}(|B|\zeta)$. We join everyting in the main inequality to get

$$\eta_t \langle \nu_{\star,t}, y^s - \hat{\bar{y}}_t^s \rangle + D(y^s, \bar{y}_{t+1}^s) \leq D(y^s, \bar{y}_t^s) + 2\eta_t \|\hat{\nu}_{t+1}^s - \hat{\nu}_{\star,t}^s\|_\infty + \frac{\eta_t}{1-\gamma} \mathcal{O}(|B|\zeta) + \frac{\eta_t^2}{2(1-\gamma)^2}.$$

We recall the definition of $\hat{\nu}_{\star,t}(s,b) = \mathbb{E}_{a \sim \hat{x}_k^s} Q^{\hat{x}_k, \hat{\bar{y}}_t}(s,a,b)$. By the performance difference lemma

$$V^{\hat{x}_k, y_k^*}(s_0) - V^{\hat{x}_k, \hat{\bar{y}}_t}(s_0) = \frac{1}{1-\gamma} \mathbb{E}_{s \sim d_{s_0}^{\hat{x}_k, y_k^*}} \langle \hat{\nu}_{\star,t}^s, y_k^{*,s} - \hat{\bar{y}}_t^s \rangle,$$

where $y_k^*$ is the best response to $x_k$. Note that by Lemma B.3, $V^{x_k, y_k^*}(s_0) - V^{\hat{x}_k, y_k^*}(s_0) \leq \frac{2|B|\zeta}{(1-\gamma)^2}$. Therefore, we have

$$\mathbb{E} \frac{1}{T} \sum_{t=1}^T V^{x_k, y_k^*}(s_0) - V^{\hat{x}_k, \hat{\bar{y}}_t}(s_0) \leq \tilde{\mathcal{O}} \left( \frac{1}{(1-\gamma)T\eta_T} \right) + \mathcal{O} \left( \frac{1}{1-\gamma} \mathbb{E} \|\hat{\nu}_{t+1}^s - \hat{\nu}_{\star,t}^s\|_\infty \right)$$

$$+ \mathcal{O} \left( \frac{|B|\zeta}{(1-\gamma)^2} \right) + \mathcal{O} \left( \frac{1}{T^{6/7}(1-\gamma)^3} \right)$$

Finally due to Lemma B.3, $V^{\hat{x}_k, \hat{\bar{y}}_t}(s_0) - V^{x_k, \bar{y}_t}(s_0) \leq \mathcal{O} \left( \frac{\zeta(|A| \vee |B|)}{(1-\gamma)^2} \right)$, therefore the final inequality is

$$\mathbb{E} \frac{1}{T} \sum_{t=1}^T V^{x_k, y_k^*}(s_0) - V^{x_k, \bar{y}_t}(s_0) \leq \tilde{\mathcal{O}} \left( \frac{1}{(1-\gamma)T^{1/7}} \right) + \mathcal{O} \left( \frac{1}{1-\gamma} \mathbb{E} \|\hat{\nu}_{t+1}^s - \hat{\nu}_{\star,t}^s\|_\infty \right)$$

$$+ \mathcal{O} \left( \frac{|B|\zeta}{(1-\gamma)^2} \right) + \mathcal{O} \left( \frac{1}{T^{6/7}(1-\gamma)^3} \right)$$

$$\square$$

We finally analyze the critic for stage 2,

**Lemma G.13.** *Let Assumption 1 hold. For the critic in stage 2 in Algorithm 8, let* $\beta_t = \frac{2}{\hat{\lambda}_{\min}^\nu (t+7)^{4/7}}$ *and recall* $\hat{\lambda}_{\min}^\nu = \rho\zeta/|B|$

$$\frac{1}{T} \sum_{t=1}^T \mathbb{E} \|\hat{\nu}_t - \hat{\nu}_{\star,t-1}\|_2^2 \leq \tilde{\mathcal{O}} \left( \frac{1}{T^{3/7}} \right) + \mathcal{O} \left( \frac{|S||B|}{(1-\gamma)^3 T^{4/7}} \right) + \mathcal{O} \left( \frac{1}{(1-\gamma)^2 T^{2/7}} \right) \qquad (64)$$

*Proof.* By the same steps in Lemma G.10, we can get a similar inequality to (63)

$$\mathbb{E} \|\hat{\nu}_{t+1} - \hat{\nu}_{\star,t}\|_2^2 \leq \left( 1 - \beta_t \hat{\lambda}_{\min}^\nu \right) \mathbb{E} \|\hat{\nu}_t - \hat{\nu}_{\star,t-1}\|_2^2 + \frac{4|B||S|\eta_t^2}{\beta_t \hat{\lambda}_{\min}^\nu (1-\gamma)^3} + \frac{2\beta_t^2}{(1-\gamma)^2}. \qquad (65)$$

Equivalently,

$$\frac{1}{T}\sum_{t=1}^{T}\mathbb{E}\|\hat{\nu}_t - \hat{\nu}_{\star,t-1}\|_2^2 \leq \tilde{\mathcal{O}}\left(\frac{1}{T^{3/7}}\right) + \frac{1}{T}\sum_{t=1}^{T}\frac{4|B||S|\eta_t^2}{\beta_t^2(\hat{\lambda}_{\min}^\nu)^2(1-\gamma)^3} + \frac{2\beta_t}{\hat{\lambda}_{\min}^\nu(1-\gamma)^2}. \tag{66}$$

We can estimate other terms

$$\frac{1}{T}\sum_{t=1}^{T}\mathbb{E}\|\hat{\nu}_t - \hat{\nu}_{\star,t-1}\|_2^2 \leq \tilde{\mathcal{O}}\left(\frac{1}{T^{3/7}}\right) + \mathcal{O}\left(\frac{|S||B|}{(1-\gamma)^3 T^{4/7}}\right) + \mathcal{O}\left(\frac{1}{(1-\gamma)^2 T^{2/7}}\right) \tag{67}$$

$\square$

## G.2 REFLECTED NAC WITH A GAME ETIQUETTE AND $\zeta$-GREEDY EXPLORATION

In this section, we give a version of reflected NAC without Assumption 2, in a similar way described in (Lan, 2021, Remark 1). This algorithm will be Algorithm 1 with nonzero $\zeta$. Since this result is not derived in Lan (2021) for single agent MDP, we first derive it with single agent MDP.

### G.2.1 SINGLE AGENT RESULT WITH $\zeta$-GREEDY EXPLORATION

The algorithm will be Algorithm 9.

---

**Algorithm 9** Single player case

---

**Require:** $\hat{V}_0$ such that $\|\hat{V}_0 - V^{x_0,y_0}\|_\infty^2 \le \epsilon$
    **for** $t = 0, 1, \ldots, T-1$ **do**
        Set $\hat{y}_t(b|s) = (1-\zeta)y_t(b|s) + \zeta/|B|$
        $\hat{\nu}_{t+1} = \texttt{Policy-Eval}(\hat{y}_t, N)$
        $y_{t+1}(\cdot|s) = \mathcal{P}(y_t(\cdot|s), -\eta\hat{\nu}_{t+1}(s, \cdot))$

---

**Theorem G.14.** *Let Assumption 1 hold. For $\eta > 0$, the iterates of Algorithm 9 satisfies*

$$\frac{1}{T}\sum_{t=1}^T [V^\star(s_0) - \mathbb{E}[V^{y_t}(s_0)]] \le \frac{1}{\eta(1-\gamma)T}\mathbb{E}_{s \sim d_{s_0}^{y^\star}}\left[D(y^{\star,s}, y_1^s)\right]$$

$$+ \frac{1}{T(1-\gamma)}\mathbb{E}\mathbb{E}_{s \sim d_{s_0}^{y^\star}}\left[V^{y_{T+1}}(s) - V^{y_1}(s)\right] + \frac{\eta}{(1-\gamma)^2}\frac{1}{T}\sum_{t=1}^T \mathbb{E}\|\hat{\nu}_{\star,t} - \hat{\nu}_{t+1}\|_\infty^2$$

$$+ \frac{2}{1-\gamma}\frac{1}{T}\sum_{t=1}^T \mathbb{E}\|\mathbb{E}[\hat{\nu}_{t+1}|y_t] - \hat{\nu}_{\star,t}\|_\infty + \frac{4\eta|B|^2\zeta^2}{(1-\gamma)^6} + \frac{4|B|\zeta}{(1-\gamma)^3} \quad (68)$$

*Proof.* Recall the definitions

$$\nu_{\star,t} = Q^{y_t}, \quad \hat{\nu}_{\star,t} = Q^{\hat{y}_t}.$$

Recall that the update rule of $y_{t+1}$ implies $D(y^s, y_{t+1}^s) \le D(y^s, y_t^s) - D(y_{t+1}^s, y_t^s) - \eta\langle\hat{\nu}_{t+1}^s, y^s - y_{t+1}^s\rangle$ and by plugging in $y = y_t$, we get

$$D(y_t^s, y_{t+1}^s) + D(y_{t+1}^s, y_t^s) \le -\eta\langle\hat{\nu}_{t+1}^s, y_t^s - y_{t+1}^s\rangle.$$

By (8),

$$0 \le -\langle\hat{\nu}_{t+1}^s, y_t^s - y_{t+1}^s\rangle - \frac{1}{\eta}\|y_{t+1}^s - y_t^s\|_1^2 \le -\langle\hat{\nu}_{t+1}^s, y_t^s - y_{t+1}^s\rangle - \frac{1}{2\eta}\|y_{t+1}^s - y_t^s\|_1^2. \quad (69)$$

By performance difference lemma and Young's inequality

$$V^{y_{t+1}}(s_0) - V^{y_t}(s_0) = \frac{1}{1-\gamma}\mathbb{E}_{s \sim d_{s_0}^{y_{t+1}}}\langle Q^{y_t,s}, y_{t+1}^s - y_t^s\rangle$$

$$= \frac{1}{1-\gamma}\mathbb{E}_{s \sim d_{s_0}^{y_{t+1}}}\langle\nu_{\star,t}^s, y_{t+1}^s - y_t^s\rangle$$

$$= \frac{1}{1-\gamma}\mathbb{E}_{s \sim d_{s_0}^{y_{t+1}}}\left[\langle\hat{\nu}_{t+1}^s, y_{t+1}^s - y_t^s\rangle + \langle\nu_{\star,t}^s - \hat{\nu}_{t+1}^s, y_{t+1}^s - y_t^s\rangle\right]$$

$$\ge \frac{1}{1-\gamma}\mathbb{E}_{s \sim d_{s_0}^{y_{t+1}}}\left[\langle\hat{\nu}_{t+1}^s, y_{t+1}^s - y_t^s\rangle - \frac{\eta}{2}\|\nu_{\star,t}^s - \hat{\nu}_{t+1}^s\|_\infty^2 - \frac{1}{2\eta}\|y_{t+1}^s - y_t^s\|_1^2\right].$$

Therefore, we get by using (69) and $1-\gamma \le d_{s_0}^{y_{t+1}}(s_0) \le 1$

$$\langle\hat{\nu}_{t+1}^s, y_{t+1}^s - y_t^s\rangle \le V^{y_{t+1}}(s) - V^{y_t}(s) + \frac{1}{2\eta}\|y_{t+1}^s - y_t^s\|_1^2 + \frac{\eta}{2(1-\gamma)}\|\nu_{\star,t} - \hat{\nu}_{t+1}\|_\infty^2. \quad (70)$$

We use again the definition of $y_{t+1}$

$$
\begin{aligned}
D(y^s, y_{t+1}^s) &\leq D(y^s, y_t^s) - D(y_{t+1}^s, y_t^s) - \eta\langle \hat{\nu}_{t+1}^s, y^s - y_{t+1}^s\rangle \\
&= D(y^s, y_t^s) - D(y_{t+1}^s, y_t^s) - \eta\langle \hat{\nu}_{t+1}^s, y^s - y_t^s\rangle - \eta\langle \hat{\nu}_{t+1}^s, y_t^s - y_{t+1}^s\rangle.
\end{aligned} \tag{71}
$$

Let us use (70) and

$$
-\eta\langle \hat{\nu}_{t+1}^s, y^s - y_t^s\rangle = -\eta\langle \nu_{\star,t}^s, y^s - y_t^s\rangle - \eta\langle \hat{\nu}_{t+1}^s - \nu_{\star,t}^s, y^s - y_t^s\rangle
$$

on (71) to get (along with (8))

$$
\begin{aligned}
\eta\langle \nu_{\star,t}^s, y^s - y_t^s\rangle + D(y^s, y_{t+1}^s) &\leq D(y^s, y_t^s) + \eta\left(V^{y_{t+1}}(s) - V^{y_t}(s)\right) \\
&\quad + \frac{\eta^2}{2(1-\gamma)}\|\nu_{\star,t} - \hat{\nu}_{t+1}\|_\infty^2 - \eta\langle \hat{\nu}_{t+1}^s - \nu_{\star,t}^s, y^s - y_t^s\rangle
\end{aligned} \tag{72}
$$

Since by performance difference lemma, $V^\star(s_0) - V^{y_t}(s_0) = \frac{1}{1-\gamma}\mathbb{E}_{s\sim d_{s_0}^{y^\star}}\langle \nu_{\star,t+1}^s, y^{\star,s} - y_t^s\rangle$, we plug in $y = y^\star$ and take expectation in (72) w.r.t. $d_{s_0}^{y^\star}$,

$$
\begin{aligned}
\eta(1-\gamma)(V^\star(s_0) - V^{y_t}(s_0)) &\leq \mathbb{E}_{s\sim d_{s_0}^{y^\star}}\left[D(y^s, y_t^s) - D(y^s, y_{t+1}^s)\right] + \eta\mathbb{E}_{s\sim d_{s_0}^{y^\star}}\left[V^{y_{t+1}}(s) - V^{y_t}(s)\right] \\
&\quad + \frac{\eta^2}{2(1-\gamma)}\|\nu_{\star,t} - \hat{\nu}_{t+1}\|_\infty^2 - \mathbb{E}_{s\sim d_{s_0}^{y^\star}}[\eta\langle \hat{\nu}_{t+1}^s - \nu_{\star,t}^s, y^s - y_t^s\rangle]
\end{aligned} \tag{73}
$$

Next, we take expectation w.r.t. randomness in the algorithm and use tower property to get

$$
\begin{aligned}
\eta(1-\gamma)(V^\star(s_0) - \mathbb{E}[V^{y_t}(s_0)]) &\leq \mathbb{E}\mathbb{E}_{s\sim d_{s_0}^{y^\star}}\left[D(y^{\star,s}, y_t^s) - D(y^{\star,s}, y_{t+1}^s)\right] \\
&\quad + \eta\mathbb{E}\mathbb{E}_{s\sim d_{s_0}^{y^\star}}\left[V^{y_{t+1}}(s) - V^{y_t}(s)\right] + \frac{\eta^2}{2(1-\gamma)}\mathbb{E}\|\nu_{\star,t} - \hat{\nu}_{t+1}\|_\infty^2 \\
&\quad - \mathbb{E}\mathbb{E}_{s\sim d_{s_0}^{y^\star}}[\eta\langle \mathbb{E}[\hat{\nu}_{t+1}^s|y_t] - \nu_{\star,t}^s, y^s - y_t^s\rangle].
\end{aligned} \tag{74}
$$

We also note that by Cauchy-Schwarz inequality $-\eta\langle \mathbb{E}[\hat{\nu}_{t+1}^s|y_t] - \nu_{\star,t}^s, y^s - y_t^s\rangle \leq 2\|\mathbb{E}[\hat{v}_{t+1}|y_t] - \nu_{\star,t}\|_\infty$.

By triangle inequality and Young's inequality

$$
\begin{aligned}
\frac{\eta^2}{2(1-\gamma)}\|\nu_{\star,t} - \hat{\nu}_{t+1}\|_\infty^2 + 2\eta\|\mathbb{E}[\hat{v}_{t+1}|y_t] - \nu_{\star,t}\|_\infty &\leq \frac{\eta^2}{1-\gamma}\|\hat{\nu}_{\star,t} - \hat{\nu}_{t+1}\|_\infty^2 \\
&\quad + \frac{\eta^2}{1-\gamma}\|\hat{\nu}_{\star,t} - \nu_{\star,t}\|_\infty^2 + 2\eta\|\mathbb{E}[\hat{v}_{t+1}|y_t] - \hat{\nu}_{\star,t}\|_\infty + 2\eta\|\nu_{\star,t} - \hat{\nu}_{\star,t}\|_\infty \\
&\leq \frac{\eta^2}{1-\gamma}\|\hat{\nu}_{\star,t} - \hat{\nu}_{t+1}\|_\infty^2 + \frac{4\eta^2}{(1-\gamma)^5}(\max_s \|\hat{y}_t^s - y_t^s\|_1)^2 \\
&\quad + 2\eta\|\mathbb{E}[\hat{v}_{t+1}|y_t] - \hat{\nu}_{\star,t}\|_\infty + \frac{4\eta}{(1-\gamma)^2}\max_s \|\hat{y}_t^s - y_t^s\|_1,
\end{aligned}
$$

where the last step used Lemma B.4 and definitions $\nu_{\star,t} = Q^{y_t}$, $\hat{\nu}_{\star,t} = Q^{\hat{y}_t}$. We use the estimation as in (56), to get $\|\hat{y}_t - y_t\|_1 \leq |B|\zeta$. With these estimations, (74) becomes

$$
\begin{aligned}
V^\star(s_0) - \mathbb{E}[V^{y_t}(s_0)] &\leq \frac{1}{\eta(1-\gamma)}\mathbb{E}\mathbb{E}_{s\sim d_{s_0}^{y^\star}}\left[D(y^{\star,s}, y_t^s) - D(y^{\star,s}, y_{t+1}^s)\right]
\end{aligned}
$$

$$
\begin{aligned}
+\frac{1}{1-\gamma}\mathbb{E}\mathbb{E}_{s\sim d_{s_0}^{y^\star}}\left[V^{y_{t+1}}(s) - V^{y_t}(s)\right] + \frac{\eta}{(1-\gamma)^2}\mathbb{E}\|\hat{\nu}_{\star,t} - \hat{\nu}_{t+1}\|_\infty^2 + \frac{2}{1-\gamma}\mathbb{E}\|\mathbb{E}[\hat{v}_{t+1}|y_t] - \hat{\nu}_{\star,t}\|_\infty \\
+ \frac{4\eta|B|^2\zeta^2}{(1-\gamma)^6} + \frac{4|B|\zeta}{(1-\gamma)^3}
\end{aligned} \tag{75}
$$

We sum over $t$ to conclude. $\qquad\square$

Note that we will learn $\hat{\nu}_{\star,t}$ by sampling with $\hat{y}_t$. As in Appendix G.1.1, the strong monotonicity constant of the operator

$$
F^{y_t}(\nu)(s,b) = \rho^{\hat{y}_t}(s)y_t(b|s)\left(\nu(s,b) - r(s,b) - \gamma\sum_{s'}P(s'|s,b)y_t(b'|s')\nu(s',b')\right)
$$

is $\hat{\lambda}^{\nu}_{\min} = (1-\gamma)\rho\zeta/(|B|)$ (see also Lan (2021)). Moreover, $F^{y_t}(\nu_{\star,t}) = 0$.

**Lemma G.15.** *Let Assumption 1 hold and* $\beta_n = \frac{2}{\hat{\lambda}^{\nu}_{\min}(n+n_0)}$ *stage 2 in Algorithm 9 satisfies*

$$
\mathbb{E}\|\hat{\nu}_N - \hat{\nu}_{\star,t}\|_2^2 \leq \mathcal{O}\left(\frac{|S||B|}{(1-\gamma)^2N^2} + \frac{1}{N(1-\gamma)^2(\hat{\lambda}^{\nu}_{\min})^2}\right),
$$

$$
\|\mathbb{E}[\hat{\nu}_N|y_t] - \hat{\nu}_{\star,t}\|_2^2 \leq \mathcal{O}\left(\frac{2|S||B|}{(1-\gamma)^2N^2}\right).
$$

*Proof.* We can use Lemma F.7 with $\nu = \hat{\nu}$, $\lambda^{\nu}_{\min} = \hat{\lambda}^{\nu}_{\min}$ and $\tilde{F}(\nu,\xi) = \nu(s,b) - r(s,b) - \gamma\nu(s',b')$ where $s \sim \rho^{\hat{y}_t}$, $b \sim \hat{y}_t(\cdot|s)$, $s' \sim P(\cdot|s,b)$, $b' \sim \hat{y}_t(\cdot|s')$ to get the result. $\square$

**Corollary G.16.** *Let Assumption 1 hold. Let* $\zeta = (1-\gamma)^3\epsilon/|B|$. *The sample complexity of Algorithm 9 for obtaining globally optimal policy in single agent MDP is* $O(|B|^4|S|(1-\gamma)^{-13}\rho^{-2}\epsilon^{-4})$.

*Proof.*

$$
\frac{1}{T}\sum_{t=1}^{T}[V^{\star}(s_0) - \mathbb{E}[V^{y_t}(s_0)]] \leq \frac{1}{\eta(1-\gamma)T}\mathbb{E}_{s\sim d^{y^{\star}}_{s_0}}\left[D(y^{\star,s}, y_1^s)\right]
$$

$$
+ \frac{1}{T(1-\gamma)}\mathbb{E}\mathbb{E}_{s\sim d^{y^{\star}}_{s_0}}\left[V^{y_{T+1}}(s) - V^{y_1}(s)\right] + \frac{\eta}{(1-\gamma)^2}\frac{1}{T}\sum_{t=1}^{T}\mathbb{E}\|\hat{\nu}_{\star,t} - \hat{\nu}_{t+1}\|_{\infty}^2
$$

$$
+ \frac{2}{1-\gamma}\frac{1}{T}\sum_{t=1}^{T}\mathbb{E}\|\mathbb{E}[\hat{\nu}_{t+1}|y_t] - \hat{\nu}_{\star,t}\|_{\infty} + \frac{4\eta|B|^2\zeta^2}{(1-\gamma)^6} + \frac{4|B|\zeta}{(1-\gamma)^3} \quad (76)
$$

By using Lemma G.15 in the previous lemma with $\hat{\lambda}^{\nu}_{\min} = (1-\gamma)\rho\zeta/|B|$ gives

$$
\frac{1}{T}\sum_{t=1}^{T}[V^{\star}(s_0) - \mathbb{E}[V^{y_t}(s_0)]] \leq \tilde{\mathcal{O}}\left(\frac{1}{T(1-\gamma)^2}\right) + \frac{4\eta|B|^2\zeta^2}{(1-\gamma)^6} + \frac{4|B|\zeta}{(1-\gamma)^3}
$$

$$
+ \mathcal{O}\left(\frac{\eta}{(1-\gamma)^2}\left(\frac{|S||B|}{(1-\gamma)^2N^2} + \frac{|B|^2}{N(1-\gamma)^4\rho^2\zeta^2}\right) + \frac{\sqrt{|S||B|}}{(1-\gamma)N}\right). \quad (77)
$$

By picking $\eta = (1-\gamma)$ and $\zeta = (1-\gamma)^3\epsilon/|B|$, we get the complexity $O(|B|^4|S|(1-\gamma)^{-13}\rho^{-2}\epsilon^{-4})$. $\square$

### G.2.2 Reflected NAC with a game etiquette and $\zeta$-greedy exploration in Algorithm 10

We first state the main result in the next corollary. As most of the results depend on previous sections, we will make heavy use of those estimations and provide a brief proof by highlighting the differences and extracting the error due to greedy exploration. Algorithm 1 is repeated here for convenience.

**Theorem 3.3.** *Let Assumption 1 hold. Use the parameter choices from Corollaries G.20 and G.21, use Remark C.1 and use the estimates in these corollaries with (14). Then, the complexity for Algorithm 10 is* $\tilde{\mathcal{O}}(|S|^4(|A| \vee |B|)^6(1-\gamma)^{-15}\epsilon^{-4}\rho^2)$.

---

**Algorithm 10** Reflected NAC with a game etiquette and $\zeta$-greedy exploration

---

**Require:** $\mathcal{P}^{\mathrm{KL}}$ defined in (1) in Section 1. Exploration parameter $\zeta \geq 0$ (equal to 0 if Assumption 2 holds). Subroutine `Policy-Eval` (see Algorithm 2, Algorithm 3, Algorithm 4). Initial policies $x_0, y_0, \bar{y}_0$

  **for** $k = 0, 1, \ldots$ **do**

    **Stage 1**

    **for** $t = 0, 1, \ldots, T-1$ **do**

      Define $\hat{x}_{k-1}(a|s) = (1 - \zeta)x_{k-1}(a|s) + \frac{\zeta}{|A|}$, $\hat{y}_{k-1}(b|s) = (1 - \zeta)y_{k-1}(b|s) + \frac{\zeta}{|B|}$ and correspondingly for $\hat{x}_{k,t}, \hat{y}_{k,t}$.

      $[\hat{\hat{V}}^x_{k-1}, \hat{\hat{V}}^y_{k-1}] = [\texttt{Policy-Eval}(\hat{x}_{k-1}, \hat{y}_{k-1}, N, \beta^\omega_n), \texttt{Policy-Eval}(\hat{x}_{k-1}, \hat{y}_{k-1}, N, \beta^\omega_n)]$

      $[\hat{\theta}^x_{t+1}, \hat{\theta}^y_{t+1}] = [\texttt{Policy-Eval}(\hat{x}_{k,t}, \hat{y}_{k,t}, N, \hat{V}^x_{k-1}, \beta^\theta_n), \texttt{Policy-Eval}(\hat{x}_{k,t}, \hat{y}_{k,t}, N, \hat{V}^y_{k-1}, \beta^\theta_n)]$

      $x^s_{k,t+1}(\cdot) = \mathcal{P}^{\mathrm{KL}}\left(x^s_{k,t}(\cdot), \eta\left(2\hat{\theta}^x_{t+1}(s, \cdot) - \hat{\theta}^x_t(s, \cdot)\right)\right)$

      $y^s_{k,t+1}(\cdot) = \mathcal{P}^{\mathrm{KL}}\left(y^s_{k,t}(\cdot), -\eta\left(2\hat{\theta}^y_{t+1}(s, \cdot) - \hat{\theta}^y_t(s, \cdot)\right)\right)$

    Output $x_k = \frac{1}{T}\sum_{t=1}^T x_{k,t}$.

    **Stage 2**

    **for** $t = 0, 1, \ldots, T-1$ **do**

      $\hat{x}^s_k(a) = (1 - \zeta)x^s_k(a) + \frac{\zeta}{|A|}$, $\hat{\bar{y}}^s_{k,t}(b) = (1 - \zeta)\bar{y}^s_{k,t}(b) + \frac{\zeta}{|B|}$

      $\hat{\nu}_{t+1} = \texttt{Policy-Eval}(\hat{x}_k, \hat{\bar{y}}_t, N, \beta = \beta^\nu_n)$

      $\bar{y}^s_{k,t+1}(\cdot) = \mathcal{P}^{\mathrm{KL}}(\bar{y}^s_{k,t}(\cdot), -\eta\hat{\nu}_{t+1}(s, \cdot))$

    Output $y_k = \bar{y}_{k,\hat{t}}$, where $\hat{t} \in [T]$ is selected uniformly at random.

---

*Theorem 3.3.* ***Extended form*** *Let Assumption 1 hold. Use the parameter choices from Corollaries G.20 and G.21, use Remark C.1 and use the estimates in these corollaries with (14). Then the suboptimality gap is bounded as follows:*

$$\mathbb{E}_{s_0 \sim \mu}[\max_y V^{x_k,y}(s_0) - V^\star(s_0)] \leq \frac{C_{\mu,\sigma}k}{(1-\gamma)}\tilde{\mathcal{O}}\left\{\frac{1}{T(1-\gamma)^2}\right.$$

$$\left. + \frac{|S|^2(|A| \vee |B|)^6}{(1-\gamma)^5\rho^2\epsilon^2 N} + \frac{|S||B|}{(1-\gamma)^3 N^2} + \frac{|B|^2}{N(1-\gamma)^{11}\rho^2\epsilon^2} + \frac{\sqrt{|S||B|}}{(1-\gamma)N}\right\} + \mathcal{O}\left(\frac{C_{\mu,\sigma}\gamma^k}{(1-\gamma)}\right)$$

*Then, the complexity for Algorithm 10 is $\tilde{\mathcal{O}}(|S|^4(|A| \vee |B|)^6(1-\gamma)^{-15}\epsilon^{-4}\rho^2)$.*

**Remark G.17.** *This result does not require Assumption 2, which makes our set of assumptions similar to Wei et al. (2021) which also used greedy exploration. Comparing with (Wei et al., 2021, Corollary 5), our result has the same $\epsilon^{-4}$ dependence, a better dependence on $(1 - \gamma)$. We note that even though our dependence with $|A| \vee |B|$ seems worse compared to (Wei et al., 2021, Corollary 5), the result in (Wei et al., 2021, Corollary 5) has dependence $C^{-10}$ where $C$ is the metric subregularity constant. This constant implicitly depends on dimensions of variables $x, y$, therefore it depends on $|A|, |B|, |S|$. More importantly, this result in Wei et al. (2021) does not cover NPG since it requires Euclidean projections on the simplex. As a result, the comparison with (Wei et al., 2021, Corollary 5) in terms of $|A|, |B|$ is not possible. On the other hand, the result of (Wei et al., 2021, Corollary 4) without metric subregularity applies to NPG, and it can be compared with our result, in which case our sample complexity scales as $\epsilon^{-4}$ whereas (Wei et al., 2021, Corollary 4) as $\epsilon^{-8}$.*

**Stage 1 of Reflected NAC with a game etiquette and $\zeta$-greedy exploration in Algorithm 10** We first recall the discussion in Appendix G.1.1 and include it here for easy reference. Let us recall the notation and introduce more notations. At iteration $k$, we solve

$$Q^s(a, b) = Q(s, a, b) = r(s, a, b) + \gamma\sum_{s'} P(s'|s, a, b)V^{x_{k-1},y_{k-1}}(s').$$

The problem is to find $x_{out}, y_{out}$ such that

$$\mathbb{E}\max_{x^s,y^s} x^s_{out}Q^s y^s - x^s Q^s y^s_{out}.$$

Now we define

$$\hat{Q}^s(a,b) = \hat{Q}(s,a,b) = r(s,a,b) + \gamma \sum_{s'} P(s'|s,a,b) V^{\hat{x}_{k-1}, \hat{y}_{k-1}}(s'),$$

as we sample with policies with greedy exploration, at stage 2 we will learn $V^{\hat{x}_{k-1}, \hat{y}_{k-1}}$. So we will have the oracle $\hat{V}_{k-1}$ such that $\|\hat{V}_{k-1} - V^{\hat{x}_{k-1}, \hat{y}_{k-1}}\|$ is small.

In this step, with the abovementioned oracle, we expect to learn

$$\hat{\theta}^x_{\star,t}(s,a) = \mathbb{E}_{b \sim \hat{y}^s_t} \hat{Q}^s(a,b), \qquad \hat{\theta}^y_{\star,t}(s,b) = \mathbb{E}_{a \sim \hat{x}^s_t} \hat{Q}^s(a,b).$$

Let us continue with $x$-player and drop the superscript,

$$\begin{aligned}
\hat{\theta}_{\star,t}(s,a) &= \mathbb{E}_{b \sim \hat{y}_t(\cdot|s)} \hat{Q}^s(a,b) \\
&= \sum_b \hat{y}^s_t \hat{Q}^s(a,b) \\
&= \sum_b \hat{y}_t(b|s) r(s,a,b) + \gamma \sum_{b,s'} \hat{y}_t(b|s) P(s'|s,a,b) V^{\hat{x}_{k-1}, \hat{y}_{k-1}}(s').
\end{aligned}$$

Therefore the operator is

$$\begin{aligned}
\hat{F}_t(\hat{\theta}_t) = \rho^{\hat{x}_t, \hat{y}_t}(s) \hat{x}_t(a|s) \Bigg( &\hat{\theta}_t(s,a) - \sum_b \hat{y}_t(b|s) r(s,a,b) \\
&- \gamma \sum_{s',b} \hat{y}_t(b|s) P(s'|s,a,b) V^{\hat{x}_{k-1}, \hat{y}_{k-1}}(s) \Bigg). \quad (78)
\end{aligned}$$

As $\hat{x}_t(\cdot|s) \geq \frac{\zeta}{|A|}$ and $\hat{y}_t(\cdot|s) \geq \frac{\zeta}{|B|}$, we have that $\hat{F}_t$ is strongly monotone with $\underline{\rho}\frac{\zeta}{|A|}$ where $\underline{\rho}$ is the lower bound of the stationary state distribution. Let us call this $\hat{\lambda}^\theta_{\min} = \underline{\rho}\frac{\zeta}{|A|}$.

As in Lemma F.2, we introduce notation similar to Malitsky & Tam (2020), let us define $\Phi$, which is slightly different this time.

$$\Phi^s_{t+1} = D(x^s, x^s_{t+1}) + \eta \langle \theta^s_{\star,t+1} - \hat{\theta}^s_{t+1}, x^s - x^s_{t+1} \rangle + \frac{1}{2} D(x^s_{t+1}, x^s_t). \quad (79)$$

We also use the following error functions and recall the definitions

$$\begin{aligned}
e_{1,t} &= \eta \langle \hat{\theta}_{t+1}(\cdot|s) - \mathbb{E}[\hat{\theta}_{t+1}(\cdot|s)|x_t], x(\cdot|s) - x_t(\cdot|s) \rangle \\
e_{2,t} &= \eta \langle \hat{\theta}^y_{t+1}(\cdot|s) - \mathbb{E}[\hat{\theta}^y_{t+1}(\cdot|s)|y_t], y_t(\cdot|s) - y(\cdot|s) \rangle, \\
\hat{\theta}_{\star,t}(s,a) &= \mathbb{E}_{b \sim \hat{y}_t} \hat{Q}^s(a,\cdot) \\
\theta_{\star,t}(s,a) &= \mathbb{E}_{b \sim y_t} Q^s(a,\cdot).
\end{aligned}$$

**Lemma G.18.** *Let Assumption 1 hold. Let $\eta = \frac{1-\gamma}{8}$. Denote $x_{out} = \frac{1}{T}\sum_{t=1}^T x_t$ and $y_{out} = \frac{1}{T}\sum_{t=1}^T y_t$. For stage 1 in Algorithm 10*

$$\begin{aligned}
\mathbb{E}\mathbb{E}_{s \sim \sigma} \left[ \max_{x^s, y^s} x^s_{out} Q^s y^s - x^s Q^s y_{out} \right] = \mathcal{O}\left( \frac{\Phi^s_0 - \Phi^s_T}{\eta T} \right) + \mathcal{O}\left( \frac{1}{T}\sum_{t=1}^T \mathbb{E}\|\mathbb{E}[\hat{\theta}_{t+1}|x_t] - \hat{\theta}_{\star,t}\| \right) \\
+ \mathcal{O}\left( \frac{\eta}{T}\sum_{t=1}^T \mathbb{E}\|\hat{\theta}_{t+1} - \hat{\theta}_{\star,t}\|^2 + \mathbb{E}\|\hat{\theta}_t - \hat{\theta}_{\star,t-1}\|^2 \right) + \frac{1}{T\eta}\mathbb{E}\mathbb{E}_{s \sim \sigma} \max_z \sum_{t=1}^T [e_{1,t} + e_{2,t}]) \\
+ \frac{4\zeta(|A| \vee |B|)}{(1-\gamma)^2} + \frac{4\zeta|B|}{1-\gamma}.
\end{aligned}$$

**Remark G.19.** *The idea of this lemma is to extract the error due to greedy exploration and make the errors due to policy evaluation depending on the policies with greedy exploration. In particular, $\hat{\theta}_{t+1}$ learns $\hat{\theta}_{\star,t}$, by sampling $\hat{x}_t, \hat{y}_t$. Therefore, we can utilize the results we derived earlier for policy evaluation steps.*

*Proof.* We start from the quantity we would like to bound. Recall from (24)

$$x_{\text{out}}^s Q^s y^s - x^s Q^s y_{\text{out}}^s = \frac{1}{T}\sum_{t=1}^{T}\langle \mathbb{E}_{a\sim x_t^s}Q^s(a,\cdot), y^s\rangle - \frac{1}{T}\sum_{t=1}^{T}\langle \mathbb{E}_{b\sim y_t^s}Q^s(\cdot,b), x^s\rangle$$

$$= \frac{1}{T}\sum_{t=1}^{T}\left[\langle \mathbb{E}_{a\sim x_t^s}Q^s(a,\cdot), y^s - y_t^s\rangle - \langle \mathbb{E}_{b\sim y_t^s}Q^s(\cdot,b), x^s - x_t^s\rangle\right].$$

$$= \frac{1}{T}\sum_{t=1}^{T}\left[\langle \theta_{\star,t}^y, y^s - y_t^s\rangle - \langle \theta_{\star,t}^x, x^s - x_t^s\rangle\right]. \quad (80)$$

We will drop the subscript of $\theta$ to denote $\theta^x$ for lighter notation and derive the part of $x$-player. The $y$-player case will be symmetrical.

By the definition of $x_{t+1}$,

$$D(x^s, x_{t+1}^s) \le D(x^s, x_t^s) - D(x_{t+1}^s, x_t^s) + \eta\langle 2\hat{\theta}_{t+1}^s - \hat{\theta}_t^s, x^s - x_{t+1}^s\rangle. \quad (81)$$

We manipulate the inner products as Lemma F.2

$$\langle 2\hat{\theta}_{t+1}^s - \hat{\theta}_t^s, x^s - x_{t+1}^s\rangle = \langle \hat{\theta}_{t+1}^s - \theta_{\star,t+1}^s, x^s - x_{t+1}^s\rangle + \langle \hat{\theta}_{t+1}^s - \hat{\theta}_t^s, x^s - x_{t+1}^s\rangle + \langle \theta_{\star,t+1}^s, x^s - x_{t+1}^s\rangle$$

$$= \langle \hat{\theta}_{t+1}^s - \theta_{\star,t+1}^s, x^s - x_{t+1}^s\rangle + \langle \hat{\theta}_{t+1}^s - \hat{\theta}_t^s, x^s - x_t^s\rangle + \langle \hat{\theta}_{t+1}^s - \hat{\theta}_t^s, x_t^s - x_{t+1}^s\rangle$$

$$+ \langle \theta_{\star,t+1}^s, x^s - x_{t+1}^s\rangle$$

$$= \langle \hat{\theta}_{t+1}^s - \theta_{\star,t+1}^s, x^s - x_{t+1}^s\rangle + \langle \theta_{\star,t}^s - \hat{\theta}_t^s, x^s - x_t^s\rangle + \langle \hat{\theta}_{t+1}^s - \theta_{\star,t}^s, x^s - x_t^s\rangle$$

$$+ \langle \hat{\theta}_{t+1}^s - \hat{\theta}_t^s, x_t^s - x_{t+1}^s\rangle + \langle \theta_{\star,t+1}^s, x^s - x_{t+1}^s\rangle.$$

Note that the first two terms will be used while forming $\Phi_t^s$ and $\Phi_{t+1}^s$, final term will be what we bound for (80). The third and fourth terms are the error terms. We first analyze the third term

$$\langle \hat{\theta}_{t+1}^s - \theta_{\star,t}^s, x^s - x_t^s\rangle = \langle \mathbb{E}[\hat{\theta}_{t+1}^s|x_t] - \theta_{\star,t}^s, x^s - x_t^s\rangle + \langle \hat{\theta}_{t+1}^s - \mathbb{E}[\hat{\theta}_{t+1}^s|x_t], x^s - x_t^s\rangle$$

$$= \langle \mathbb{E}[\hat{\theta}_{t+1}^s|x_t] - \theta_{\star,t}^s, x^s - x_t^s\rangle + e_{1,t} \le 2\|\mathbb{E}[\hat{\theta}_{t+1}|x_t] - \theta_{\star,t}\|_\infty + e_{1,t}.$$

Next, by Cauchy-Schwarz and Young's inequalities

$$\langle \hat{\theta}_{t+1}^s - \hat{\theta}_t^s, x_t^s - x_{t+1}^s\rangle \le \eta\|\hat{\theta}_{t+1} - \hat{\theta}_t\|_\infty^2 + \frac{1}{4\eta}\|x_t^s - x_{t+1}^s\|_1^2$$

$$\le 2\eta\|\hat{\theta}_{t+1} - \hat{\theta}_{\star,t}\|_\infty^2 + 4\eta\|\hat{\theta}_{\star,t} - \hat{\theta}_{\star,t-1}\|_\infty^2 + 4\eta\|\hat{\theta}_{\star,t-1} - \hat{\theta}_t\|_\infty^2 + \frac{1}{4\eta}\|x_t^s - x_{t+1}^s\|_1^2$$

$$\le 2\eta\|\hat{\theta}_{t+1} - \hat{\theta}_{\star,t}\|_\infty^2 + \frac{16\eta}{(1-\gamma)^2}\|y_t - y_{t-1}\|_\infty^2 + 4\eta\|\hat{\theta}_{\star,t-1} - \hat{\theta}_t\|_\infty^2 + \frac{1}{2\eta}D(x_{t+1}^s, x_t^s).$$

where the last inequality is similar to (40) when $\theta, y_t$ are replaced by $\hat{\theta}, \hat{y}_t$ and we also used $\|\hat{y}_t^s - \hat{y}_{t-1}^s\|_1 \le \|y_t^s - y_{t-1}^s\|_1$ along with (8). We collect all the estimations in (81)

$$\eta\langle \theta_{\star,t+1}^s, x^s - x_{t+1}^s\rangle + D(x^s, x_{t+1}^s) + \eta\langle \theta_{\star,t+1}^s - \hat{\theta}_{t+1}^s, x^s - x_{t+1}^s\rangle \le D(x^s, x_t^s) - \frac{1}{2}D(x_{t+1}^s, x_t^s)$$

$$+ \eta\langle \theta_{\star,t}^s - \hat{\theta}_t^s, x^s - x_t^s\rangle + e_{1,t} + 2\|\mathbb{E}[\hat{\theta}_{t+1}|x_t] - \theta_{\star,t}\|_\infty + \frac{16\eta^2}{(1-\gamma)^2}\|y_t^s - y_{t-1}^s\|_1^2$$

$$+ 2\eta^2\|\hat{\theta}_{t+1} - \hat{\theta}_{\star,t}\|_\infty^2 + 4\eta^2\|\hat{\theta}_t - \hat{\theta}_{\star,t-1}\|_\infty^2.$$

We use (8) and definition of $\Phi_t^s$ to get

$$\eta\langle \theta_{\star,t+1}^s, x^s - x_{t+1}^s\rangle + \Phi_{t+1}^s \le \Phi_t^s + 2\|\mathbb{E}[\hat{\theta}_{t+1}|x_t] - \theta_{\star,t}\|_\infty$$

$$+ 2\eta^2\|\hat{\theta}_{t+1} - \hat{\theta}_{\star,t}\|_\infty^2 + 4\eta^2\|\hat{\theta}_t - \hat{\theta}_{\star,t-1}\|_\infty^2 - \frac{1}{2}D(x_t^s, x_{t-1}^s) + \frac{32\eta^2}{(1-\gamma)^2}D(y_t^s, y_{t-1}^s).$$

We finally estimate the bias term

$$
\begin{aligned}
\|\mathbb{E}[\hat{\theta}_{t+1}|x_t] - \theta_{\star,t}\|_\infty &\leq \|\mathbb{E}[\hat{\theta}_{t+1}|x_t] - \hat{\theta}_{\star,t}\|_\infty + \|\hat{\theta}_{\star,t} - \theta_{\star,t}\|_\infty \\
&= \|\mathbb{E}[\hat{\theta}_{t+1}|x_t] - \hat{\theta}_{\star,t}\|_\infty + \|\mathbb{E}_{b\sim y_t^s}Q_t^s - \mathbb{E}_{b\sim\hat{y}_t^s}\hat{Q}_t^s\|_\infty \\
&= \|\mathbb{E}[\hat{\theta}_{t+1}|x_t] - \hat{\theta}_{\star,t}\|_\infty + \|\mathbb{E}_{b\sim y_t^s}Q_t^s - \mathbb{E}_{b\sim y_t^s}\hat{Q}_t^s\|_\infty + \|\mathbb{E}_{b\sim\hat{y}_t^s}\hat{Q}_t^s - \mathbb{E}_{b\sim y_t^s}\hat{Q}_t^s\|_\infty \\
&\leq \|\mathbb{E}[\hat{\theta}_{t+1}|x_t] - \hat{\theta}_{\star,t}\|_\infty + \frac{2\zeta(|A|\vee|B|)}{(1-\gamma)^2} + \frac{2\zeta|B|}{1-\gamma},
\end{aligned}
$$

where the last step is by (53) and (56).

Inserting this estimate leads to

$$
\langle \theta_{\star,t+1}^s, x^s - x_{t+1}^s \rangle \leq \frac{1}{\eta}\left(\Phi_t^s - \Phi_{t+1}^s\right) + 2\|\mathbb{E}[\hat{\theta}_{t+1}|x_t] - \hat{\theta}_{\star,t}\|_\infty + 2\eta\|\hat{\theta}_{t+1} - \hat{\theta}_{\star,t}\|_\infty^2
$$
$$
+ 4\eta\|\hat{\theta}_t - \hat{\theta}_{\star,t-1}\|_\infty^2 - \frac{1}{2\eta}D(x_t^s, x_{t-1}^s) + \frac{32\eta}{(1-\gamma)^2}D(y_t^s, y_{t-1}^s) + \frac{4\zeta(|A|\vee|B|)}{(1-\gamma)^2} + \frac{4\zeta|B|}{1-\gamma}.
$$

As in Lemma F.2, we can derive the symmetrical inequality for the $y$-player and picking $\eta = \frac{1-\gamma}{8}$ will make the terms $\frac{32\eta}{(1-\gamma)^2}(D(y_t^s, y_{t-1}^s) + D(x_t^s, x_{t-1}^s))$ disappear.

We sum the inequality along with its $y$-player counterpart and insert into RHS of (80) to conclude. □

**Corollary G.20.** *Set $\zeta = \frac{(1-\gamma)^2\epsilon}{|A|\vee|B|}$ and step sizes as from Lemmas 3.6 to 3.8 with $\hat{\lambda}_{\min}^\theta = \underline{\rho}\zeta/(|A|\vee |B|)$, $\hat{\lambda}_{\min}^\omega = \underline{\rho}$, $\zeta = \frac{\epsilon(1-\gamma)^2}{|A|\vee|B|}$. Then, the complexity for stage 1 of Algorithm 10 is $\tilde{\mathcal{O}}(|S|^2(|A|\vee |B|)^6(1-\gamma)^{-5}\epsilon^{-4}\underline{\rho}^2)$.*

*Proof.* First, we notice that both variance and bias terms in Lemma G.18, now we learn $\hat{\theta}_{\star,t}, \hat{\theta}_{\star,t}^y$ by sampling $\hat{x}_t, \hat{y}_t$. Therefore, from the point of view of policy evaluation, we can directly apply the results we derived earlier with strong monotonicity constants $\hat{\lambda}_{\min}^\theta = \underline{\rho}\zeta/(|A|\vee|B|)$, $\hat{\lambda}_{\min}^\omega = \underline{\rho}$. The bound in Lemma G.18 becomes

$$
\mathbb{E}\mathbb{E}_{s\sim\sigma}\left[\max_{x^s, y^s} x_{\text{out}}^s Q^s y^s - x^s Q^s y_{\text{out}}\right] = \mathcal{O}\left(\frac{\Phi_0^s - \Phi_T^s}{\eta T}\right) + \mathcal{O}\left(\frac{1}{T}\sum_{t=1}^T \mathbb{E}\|\mathbb{E}[\hat{\theta}_{t+1}|x_t] - \hat{\theta}_{\star,t}\|\right)
$$
$$
+ \mathcal{O}\left(\frac{\eta}{T}\sum_{t=1}^T \mathbb{E}\|\hat{\theta}_{t+1} - \hat{\theta}_{\star,t}\|^2 + \mathbb{E}\|\hat{\theta}_t - \hat{\theta}_{\star,t-1}\|^2\right) + \frac{1}{T\eta}\mathbb{E}\mathbb{E}_{s\sim\sigma}\max_z\sum_{t=1}^T [e_{1,t} + e_{2,t}])
$$
$$
+ \frac{4\xi(|A|\vee|B|)}{(1-\gamma)^2} + \frac{4\xi|B|}{1-\gamma}.
$$

By using Lemma 3.6, we can derive the variance of $\hat{\theta}$ by sampling with $\hat{x}_t, \hat{y}_t$

$$
\mathbb{E}\|\hat{\theta}_N - \hat{\theta}_{\star,t}\|_2^2 \leq \mathcal{O}\left(\frac{|S||A|}{(1-\gamma)^2 N^2} + \frac{1}{N(\hat{\lambda}_{\min}^\theta)^2(1-\gamma)^2} + \frac{|S||A|}{(\hat{\lambda}_{\min}^\theta)^2}\mathbb{E}\|\hat{V}_{k-1} - V^{\hat{x}_{k-1},\hat{y}_{k-1}}\|_\infty^2\right).
$$

For the final term in this bound, we use Lemma 3.8 to estimate $V^{\hat{x}_{k-1},\hat{y}_{k-1}}$ by sampling with $\hat{x}_{k-1}$ and $\hat{y}_{k-1}$.

$$
\mathbb{E}\|\hat{V}_{k-1} - V^{\hat{x}_{k-1},\hat{y}_{k-1}}\|_2^2 \leq \mathcal{O}\left(\frac{|S||A|}{(1-\gamma)^2 N^2} + \frac{1}{N(1-\gamma)^2(\hat{\lambda}_{\min}^\omega)^2}\right),
$$

For the bias of $\hat{\theta}_{t+1}$, we use Lemma 3.7

$$
\|\mathbb{E}[\hat{\theta}_t + 1|x_t] - \hat{\theta}_{\star,t}\|^2 \leq \mathcal{O}\left(\frac{|S||A|}{(1-\gamma)^2 N^2} + \frac{10|S||A|}{(\hat{\lambda}_{\min}^\theta)^2}\|\mathbb{E}[\hat{V}_{k-1}|x_t] - V^{x_{k-1},y_{k-1}}\|_\infty^2\right). \quad (82)
$$

On this bound, we use Lemma 3.8

$$\|\mathbb{E}[\hat{V}_{k-1}|x_t] - V^{x_{k-1},y_{k-1}}\|_2^2 \leq \mathcal{O}\left(\frac{|S||A|}{(1-\gamma)^2 N^2}\right).$$

Finally for the error terms $e_{1,t} + e_{2,t}$, we use Lemma F.6

$$\frac{1}{T}\mathbb{E}\mathbb{E}_{s\sim\sigma}\max_x \sum_{t=1}^T e_{1,t} \leq \frac{\log|A|}{T} + \frac{1}{T}\sum_{t=1}^T 4\eta^2 \mathbb{E}\|\hat{\theta}_{t+1} - \hat{\theta}_{\star,t}\|_\infty^2.$$

Inserting these estimates and using $\hat{\lambda}_{\min}^\theta = \underline{\rho}\zeta/(|A| \vee |B|)$, $\hat{\lambda}_{\min}^\omega = \underline{\rho}$, $\zeta = \frac{\epsilon(1-\gamma)^2}{|A|\vee|B|}$, Remark F.3 give

$$\mathbb{E}\mathbb{E}_{s\sim\sigma}\left[\max_{x^s,y^s} x_{\text{out}}^s Q^s y^s - x^s Q^s y_{\text{out}}\right] \leq \tilde{\mathcal{O}}\left(\frac{1}{T(1-\gamma)}\right) + 8\epsilon + \tilde{\mathcal{O}}\left(\frac{|S|^2(|A|\vee|B|)^6}{(1-\gamma)^5\underline{\rho}^2\epsilon^2 N}\right),$$

which gives the final result. $\qquad\square$

**Stage 2 of Reflected NAC with a game etiquette and $\zeta$-greedy exploration in Algorithm 10**
Recall that in Appendix F.2, we extended the result of Lan (2021) on single agent setting to the stage 2 of our algorithm. Since in the stage 2, $x_k$ is fixed, it is part of the environment and therefore single agent analysis extends in a straightforward fashion. In this section, similarly we use our results in Appendix G.2.1 for stage 2 of the algorithm. Note that our results in Appendix G.2.1 are similar to single agent results of Lan (2021), with the difference that we use $\zeta$-greedy to avoid Assumption 2. Also recall that the single agent analysis in Lan (2021) requires Assumption 2 (see (Lan, 2021, Remark 1)).

For brevity, we do not repeat the arguments in Appendix F.2 to extend Appendix G.2.1 for stage 2 in this case. We summarize the result in the next corollary.

**Corollary G.21.** *Let Assumption 1 hold. Let $\zeta = \frac{(1-\gamma)^3\epsilon}{|A|\vee|B|}$ and step size from Lemma F.7 with $\hat{\lambda}_{\min}^\nu = (1-\gamma)\underline{\rho}\zeta/|B|$ (see also Corollary G.16). The overall sample complexity for stage 2 of Algorithm 10 is $\tilde{\mathcal{O}}((|A|\vee|B|)^4|S|(1-\gamma)^{-13}\underline{\rho}^{-2}\epsilon^{-4})$.*

*Proof.* It is straightforward to use the arguments in Appendix F.2 to extend Appendix G.2.1 for this result.

$\qquad\square$

# H  ADDITIONAL EXPERIMENTS

## H.1  ENVIRONMENTS DESCRIPTION

**Bandit environment.** We first consider a two player bandits problem with 100 arms. All the arms give zero reward except $a^*$ that has $r(a^*) = -1$ and $b^*$ with $r(b^*) = 1$. The reward of the two player game is $r(a,b) = r(a) + r(b)$. In this environment there exists a pure Nash Equilibrium given by $x = \mathbf{1}\{a = a^*\}$, $y = \mathbf{1}\{b = b^*\}$.

**Alesia environment.** We also test our algorithms on Alesia($L$, $C$) (Perolat et al., 2015). Alesia is a simultaneous move game where the two opponents try to move the *wrestler* to their extremity of the board of length $L$. At each turn, the "wrestler" moves one step in the direction of the player that bets the higher amount of coins from their budget of size $C$. The game ends when either one of the wrestler reaches extremity of the board or both players finish their budget.

## H.2  HYPERPARAMETERS SELECTION.

In the following we report the hyperparameters chosen for the experiments to ensure reproducibility. In addition, we include an additional experiment on Alesia($L$,$C$) with $L = 3$ and $C = 7$ in Figure 2. Since the eigenvalues $\lambda_{\min}^\theta, \lambda_{\min}^\omega, \lambda_{\min}^\nu$ are unknown, we cannot compute the values

of learning rates $\beta_n^\theta$, $\beta_n^\omega$ used in the theoretical results. Therefore, we simply set the learning rate as $\frac{C}{n}$ suggested by the lemmas and fine tune the hyperparameter $C$ by grid search. Similarly for OGDA, we use grid search to replace the optimal learning rate given by (Wei et al., 2021, Theorem 2). Appendix H reports the chosen parameters. Table 2, Table 3, Table 4 and Table 5 summarize the best hyperparameters found for Reflected NAC and OGDA in the two environments: bandit and Alesia.

REINFORCE (Daskalakis et al., 2020) only needs two hyperparameters. We chose the exploration parameter $\epsilon = 0.1$ and the step size $\eta = 0.0001$. We observe that it was critical to choose small step size to tackle the high variance coming from the REINFORCE estimator.

| Hyperparameter | Value |
|---|---|
| $\beta_n^\omega$ | $0.01/n$ |
| $\beta_n^{\theta^x}$ | $0.8/n$ |
| $\beta_n^{\theta^y}$ | $0.1/n$ |
| $\eta$ | $0.023$ |
| $\beta_n^\nu$ | $0.01/n$ |
| $K$ | $100$ |
| $N$ | $10$ |
| $T$ | $10$ |
| $\zeta$ | $0$ |

Table 2: Hyperparametets for Reflected NAC in the two players bandit environment

| Hyperparameter | Value |
|---|---|
| $\eta$ | $0.005$ |
| $\alpha_t$ | $0.01/t$ |
| $T$ | $1100$ |
| $L$ | $50$ |
| $\epsilon$ | $0.4$ |

Table 3: Hyperparametets for OGDA in the two players bandit environment. The notation used for the hyperparameters matches the original OGDA formulation in Wei et al. (2021)

| Hyperparameter | Value |
|---|---|
| $\beta_n^\omega$ | $0.01/n$ |
| $\beta_n^\theta$ | $0.1/n$ |
| $\eta$ | $0.05$ |
| $\beta_n^\nu$ | $0.01/n$ |
| $K$ | $500$ |
| $N$ | $70$ |
| $T$ | $10$ |
| $\zeta$ | $0$ for $y$ and $0.2$ for $x$ |

Table 4: Hyperparametets for Reflected NAC in Alesia($L, C$) with $L = 3$ and $C = 6, 7$.

| Hyperparameter | Value |
|:---:|:---:|
| $\eta$ | 0.005 |
| $\alpha_t$ | $0.01/t$ |
| $T$ | 1000 |
| $L$ | 3500 |
| $\epsilon$ | 0.4 |

Table 5: Hyperparametets for OGDA in Alesia. The notation used for the hyperparameters matches the original OGDA formulation in Wei et al. (2021)

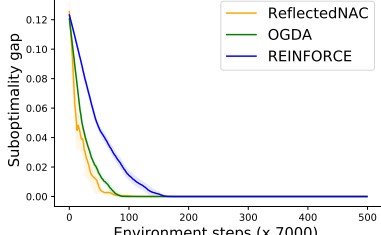 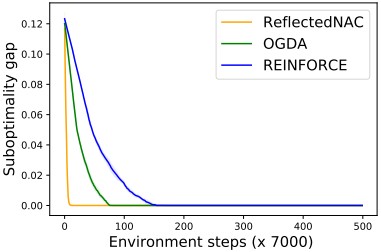

Figure 2: left: $x$ player, right: $y$ player. Experiments in a Alesia with length $L = 3$ and coin budget $C = 7$. The suboptimality gap on the y-axis is $\max_y V^{x,y}(s_0) - V^*(s_0)$ for $x$ and as $|\min_x V^{x,y}(s_0) - V^*(s_0)|$ for $y$ where $s_0$ is the initial state that is deterministic in Alesia. Results are averaged over 5 seeds.

