# OpenReview forum: "Sample-efficient actor-critic algorithms with an etiquette for zero-sum Markov games"
_ICLR.cc/2022/Conference — ICLR 2022 Submitted_

### Official Review · Reviewer_HqK9 · 2021-10-28

**Correctness:** 4
**Technical Novelty And Significance:** 2
**Empirical Novelty And Significance:** 2
**Recommendation:** 5
**Confidence:** 4

**Main Review:**

The paper develops a new NAC algorithm for solving zero-sum Markov games with finite-time convergence guarantees. Below are some major comments and questions.

1. The proposed algorithm is stochastic (sample-based), and preserves the private information of both agents, but it requires both players to play asymmetric roles/updates. This would restrict its application to practical scenarios where two competing players are not willing to negotiate.

2. After reading the paper for a while, I feel that the paper needs to be further polished. Dense notations are introduced and many of them are not clear from the context. For example, the parameters $\beta_n^{\omega}$, $\beta_n^{\theta}$, $\beta_n^{\nu}$ in Algorithm 1 is not clear until I read the Algorithm 2. It seems that these are just indicator parameters. In Algorithm 2, $\phi_n$ is confusing. It seems to be a state function, a state-action function and a vector in line 4, lines 6 & 9 and line 11, respectively. I'm not sure which entries of $\phi_n(\cdot)$ or $\phi_n(\cdot,\cdot)$ are updated in line 11. Moreover, $\lambda_{\min}^{\theta}$, $\lambda_{\min}^{\nu}$, $\lambda_{\min}^{\omega}$ are not clearly defined.

3. The authors presented the order of the complexity results that only emphasize the dependence on $\epsilon$. While there are many other important problem-related parameters, I suggest the authors also explicitly mention the dependence of the complexities on, e.g., $|S|, |A|, (1-\gamma)$. In particular, I am curious about the dependence of $\lambda_{\min}^{\theta}$, $\lambda_{\min}^{\nu}$, $\lambda_{\min}^{\omega}$ on these parameters. For example, the complexity result established in Theorem 3.2 in the appendix seems to has dependence on $|S|^2, |A|^2, (1-\gamma)^8, \lambda_{\min}^2$.

4. The comparison to the complexity results of the existing literature may not be fair. For example, the authors claim that the sample complexity $\tilde{\mathcal{O}}(\epsilon^{-4})$ in Theorem 3.3 matches the best-known result established in Corollary 5 of Wei et al. 2021. However, Wei 2021 adopts the policy distance measure while this paper adopts the one-sided function value gap. Also, Wei's result establishes a high-probability convergence guarantee, while this paper is in expectation. I am not sure if these results are directly comparable given these differences.

5. The technical proof is largely based on the existing literature. Can the authors highlight their technical novelties?

Some other comments/questions

1. The introduction mentions that the value-based methods offer near-optimal guarantees which are lacking in policy-based methods. To motivate this study, it is better to also state the advantages of policy-based methods over value-based methods.

2. In the introduction, the authors write "policy gradient (PG) methods, including actor-critic (AC) and their natural counterparts natural PG (NPG) (Kakade, 2001) and natural AC (NAC) (Peters \& Schaal, 2008), only have limited guarantees". Could you further elaborate on the limited guarantees (e.g. lack of non-asymptotic complexity result, very high complexity, restrictive assumptions, etc)?

3. What does the $x$ label "Environment steps" mean in Figures 1 & 2? Is it sample complexity?

4. At the beginning of Section 2, do we need a lower bound on the reward $r(s,a,b)$?

5. At the beginning of Algorithm 1, $\mathcal{P}_{KL}$ is actually defined in Section 2.

6. It is better to also provide a convergence rate for Theorem 3.3.

7. The $y$ label of Figure 1 should be "probability of optimal action".

**Summary Of The Paper:**

This work proposes a natural actor-critic (NAC) algorithm for two-player zero-sum Markov game in the tabular case. The algorithm is model-free, private, but asymmetric, and the convergence guarantee is established in terms of the expected one-sided Nash-equilibrium duality gap. Among the existing policy-based algorithms for the two-player zero-sum Markov game, this is the first work that claims to achieve the best-known sample complexities of policy gradient algorithms for single-agent RL.

**Summary Of The Review:**

Overall, I think this paper has the potential to deliver a solid theoretical contribution to the literature of zero-sum Markov games. It needs to be further polished before being seriously considered for publication.

---

> ### Author Response · Authors · 2021-11-22
> **Response to Reviewer HqK9 (Part 1/2)**
>
> We thank the reviewer for their constructive comments. We improved the presentation of our results in the revision and addressed the concerns.
>
> **Can the reviewer acknowledge our changes?**
>
>
> > *"it requires both players to play asymmetric roles/updates. This would restrict its application to practical scenarios where two competing players are not willing to negotiate."*
>
> - This is a valid point. We emphasize that our work focuses on the case when the players are willing to negotiate and follow the “etiquette”, which can be readily attained in self-play.
>
> - We also argue that this is not a weakness specific of our work. In particular, the existing algorithms (Daskalakis et al., 2020, Wei et al., 2021) rely on the players using a specified set of parameters and step sizes. If the players are not willing to negotiate, they might not use these parameters. As a result, none of these existing results would be applicable.
>
> > *"The comparison to the complexity results of the existing literature may not be fair. "*
>
> - It is true that policy distance measure and gap are not comparable in general.
>
> - Cor. 5 of Wei et al. 2021 is not applicable for natural policy gradient (NPG), which is our focus in this paper. For NPG, Wei et al., 2021 proves $O(\epsilon^{-8})$ for the duality gap in Cor. 4.
>
> - Our optimality measure is different from the duality gap in Wei et al., 2021 since it is a one sided duality gap, but we can prove that it is a valid optimality measure (Lem. B.6)
>
> - We can run our algorithm twice by exchanging the roles of players to get approximate Nash policies for both players (also done in Daskalakis et al., 2020).
>
> - Rates in expectation vs. high probability: One can use Markov’s inequality to convert any rate in expectation to a rate with high probability.
>
> > *"Can the authors highlight their technical novelties?"*
>
> We thank the reviewer for this comment, which prompts us to one more time highlight the novelties here. In particular, you can find these novelties explicitly highlighted in the text as "insights."
>
> - Our result relies on a careful analysis of the bias between different stages of the algorithm and an algorithmic design for making this possible. For example, notice that in Stage 2 of algorithm 1 we do not output a value function estimate as usually done in the policy evaluation step (Perolat 2015) but the best response policy. The value function is then estimated at each iteration of Stage 1.  This is important to achieve the fast rate as highlighted in Insight 4.
>
> - For solving the matrix game, we introduce an analysis for the stochastic primal-dual method that we use, which is different from the existing results in Malitsky, Tam 2020 or followup works such as Bohm et al., 2020.
>
> - To prove Theorem 3.3 we also extend (Lan, 2021)’s result in the single agent case, which only obtains $O(\epsilon^{-4})$ without Assumption 2 (see Appendix G.2.1).
>
> - We believe that our results are non-trivial as our complexity results significantly improve the existing ones for natural policy gradient. Note that in this case, only the $O(\epsilon^{-8})$ result from Wei et al., 2021 applies, which is improved by our $O(\epsilon^{-4})$ complexity under similar assumptions. Moreover, our $O(\epsilon^{-2})$ complexity matches the best-known result in the single agent setting.
>
> > *“For example, the parameters $\beta^w$, $\beta^\theta$,$\beta^\nu$ in Algorithm 1 is not clear until I read the Algorithm 2. It seems that these are just indicator parameters. In Algorithm 2, $\phi$ is confusing. It seems to be a state function, a state-action function and a vector in line 4, lines 6 & 9 and line 11, respectively. Moreover $\lambda_\min^\theta$, $\lambda_\min^\nu$, $\lambda_\min^\omega$ are not clearly defined”*
>
> Thanks for the remark: We addressed these concerns in our revision. Please see Alg. 2, 3, 4. Let us know how we can further improve the presentation.
>
> We replaced $\lambda_\min^\theta$, $\lambda_\min^\nu$, $\lambda_\min^\omega$ with their values depending on $\underline \rho, \underline x, \underline y$ from Assumptions 1, 2.
>
> > *"I am not sure which entries of \phi_n are updated in line 11"*
>
> For this update, we use an indicator vector $e(s_n)$ or $e(s_n, a_n)$. That is the only entry updated in Algs, 2, 3, 4 since all other entries are zeroed due to the indicator vector. We hope that the new version is clearer.
>
> > *"I suggest the authors also explicitly mention the dependence of the complexities"*
>
> We included the dependence of the constants in our revision. Please see the new statements in Thm 3.2, 3.3.
>
> We also refer to Remark G.17 for the comparison of the bounds in terms of different parameters with Wei et al., 2021.

---

> > ### Author Response · Authors · 2021-11-22
> > **Response to Reviewer HqK9 (Part 2/2)**
> >
> > > *"To motivate this study, it is better to also state the advantages of policy-based methods over value-based methods"*
> >
> > Thanks for this suggestion. Right now, we briefly mention their advantages in second paragraph of the introduction: *"(easy-to-implement structure, flexibility)"*. We can expand on these if the reviewer thinks it is necessary.
> >
> > > *"Could you further elaborate on the limited guarantees (e.g. lack of non-asymptotic complexity result, very high complexity, restrictive assumptions, etc)?"*
> >
> > By limited guarantees we mean that it achieves worse convergence rates than value based methods like Q-learning endowed with UCB exploration. In particular, Daskalakis et al., 2020 and Wei et al., 2021 only obtains $O(\epsilon^{-12.5})$ and $O(\epsilon^{-8})$ for NPG, respectively, and Wei et al., 2021 obtains $O(\epsilon^{-4})$ by using Euclidean projections, rather than NPG. Moreover, the result in Zhao et al., 2021 has stronger assumptions such as unbiased value function estimators and $O(\epsilon^{-6})$ complexity.
> >
> > > *"What does the X label "Environment steps" mean in Figures 1 & 2? Is it sample complexity?"*
> >
> > Environment steps stand for the number of observed tuples $(s_t, a_t, b_t, s_{t+1})$
> >
> > > *"At the beginning of Algorithm 1, PKL is actually defined in Section 2"*
> >
> > We fixed the typo in our revision.
> >
> > > *"At the beginning of Section 2, do we need a lower bound on the reward r(s,a,b)?"*
> >
> > This was a typo we fixed in our revision. The upper bound is on the absolute value.
> >
> > > *"It is better to also provide a convergence rate for Theorem 3.3"*
> >
> > Please see updated Thm. 3.3.

---

> ### Author Response · Authors · 2021-11-27
> **Discussion**
>
> Dear reviewer HqK9, could you please let us know if our responses are satisfactory to you to reconsider your score? We would appreciate the chance to interact before the deadline of the discussion period on monday, to address any other concerns you may have. Thank you.

---

### Official Review · Reviewer_jDaF · 2021-10-31

**Correctness:** 4
**Technical Novelty And Significance:** 3
**Empirical Novelty And Significance:** Not applicable
**Recommendation:** 6
**Confidence:** 4

**Main Review:**

Major Comments:

My major concern is about the correctness of the main result (Theorem 3.2). The authors claim that they match the $O(\epsilon^{-2})$ sample complexity of NAC in (Lan, 2021; Khodadadian et al., 2021b; Hong et al., 2020; Xu et al., 2020b). (Khodadadian et al., 2021b) has $O(\epsilon^{-3})$ sample complexity. (Hong et al., 2020) has $O(\epsilon^{-4})$ sample complexity. (Xu et al., 2020b) has $O(\epsilon^{-3})$ sample complexity (see their updated arXiv version). To my knowledge, the only paper that has $O(\epsilon^{-2})$ sample complexity of NAC is (Lan, 2021).

The reason that  (Lan, 2021) was able to obtain the $O(\epsilon^{-2})$ sample complexity of NAC is the following. By using mirror descent and carefully designed step sizes (in fact, exponentially increasing step sizes), the actor enjoys an exponential convergence rate, resulting in $O(\log(1/\epsilon))$ sample complexity of the actor. This also makes intuitive sense in that when using rapidly increasing step sizes, the natural policy gradient algorithm becomes close to policy iteration, which has geometric convergence. As for the critic, it is well-known that the optimal sample complexity to achieve $E[ \|V_k-V^\pi \| ]<\epsilon$ is $O(\epsilon^{-2})$. Therefore, the overall sample complexity of NAC in  (Lan, 2021) is $O(\epsilon^{-2})$, which is essentially the sample complexity of the critic.

In this paper, the reason for obtaining the $O(\epsilon^{-2})$ sample complexity is completely different from (Lan, 2021). First of all, the authors did not establish the geometric convergence of the actor, as evidenced by the $O(1/T)$ term in Theorem 3.2. As for the critic, it seems that the authors derive $O(\epsilon^{-1})$ sample complexity. Unless I miss something important, I do not think that is possible. Consider an MDP with a single state and a single action, and the discount factor is zero. The reward is i.i.d. In this case, the policy evaluation problem essentially reduces to estimating the mean of a random variable with access to i.i.d. samples. Cramér–Rao bound implies that the sample complexity cannot be better than $O(\epsilon^{-2})$. Without establishing the geometric convergence of the actor, I do not think one can achieve $O(\epsilon^{-2})$ sample complexity of NAC.

Minor Comments:

(1) Assumption 2 cannot hold. Softmax policies cover all policies, including deterministic policies. Therefore, there is no a lower bound for all policies.

(2) As a follow-up comment to (1), suppose that there is a unique deterministic optimal policy. Under Assumption 2, the agent can never find the optimal policy. This means that even when $K$, $T$, and $N$ go to infinity, there should still be some constant error term on the right hand side of Theorem 3,2, because of the fact that the agent is trying to find a deterministic policy under Assumption 2. Why is it not the case in Theorem 3.2?

(3) In equation (2), it should be $y(b|s)$ instead of $y(a|s)$.

----------After Author Feedback-----------

I am now convinced about the correctness of the result, and have increased my score.


**Summary Of The Paper:**

This paper considers the problem of solving zero sum Markov games using natural actor-critic algorithms. The authors derive $O(\epsilon^{-2})$ overall sample complexity of the proposed algorithm, and conduct numerical experiments to verify the convergence of the algorithm.

**Summary Of The Review:**

As mentioned in my main review, my major concern is about the correctness of the result. Therefore I vote for rejection.

---

> ### Author Response · Authors · 2021-11-10
> **Clarification on correctness**
>
> **In this paper, the reason for obtaining the  sample complexity is completely different from (Lan, 2021)... Unless I miss something important, I do not think that is possible ....Without establishing the geometric convergence of the actor, I do not think one can achieve $O(\epsilon^{-2})$ sample complexity of NAC**
>
> We are afraid that the reviewer indeed missed important points in (Lan 2021) and ours, which led to the misjudgement on the correctness of our work.
>
> (Lan 2021, v5) provided two different ways to establish the $O(\epsilon^{-2})$ sample complexity to solve RL with general convex regularizer.
>
> - SAPMD with increasing stepsize, geometric convergence of the actor in the outer loop, $O(\epsilon^{-2})$ samples for critic update (as reviewer mentioned, see e.g., their Prop 3 and Prop 5)
>
>
> - SPMD with decaying (or constant) stepsize, $O(1/k)$ convergence of the actor in the outer loop, $O(\epsilon^{-1})$ samples for critic update (which we adopt in our paper, see e.g., their Thm 6, Prop 2 (and compare with Prop 3) and remark before Prop 4)
>
>
> Note that the second result hinges on an improved bound on the bias term for the TD update,  $\| \mathbb{E} \theta_{t+1} - \theta_\star \|^2 \leq O(1/t^3)$, as given in their Lemma 18. Combining the bias term and the variance term $\mathbb{E} \| \theta_{t+1} -\theta_\star \|^2 \leq O(1/t)$ (Lemma 17) into Theorem 6, this immediately indicates that selecting $k=O(\epsilon^{-1})$ and $t=O(\epsilon^{-1})$ makes the RHS of Theorem 6  smaller than $O(\epsilon)$; as a result, the overall number of samples is $tk = O(\epsilon^{-2})$.
>
>
> The reviewer is right on the sufficiency side of establishing geometric convergence of the actor, but it is by no means necessary as the reviewer speculated.  Moreover, the fact that the bias term achieves a faster rate does not contradict with the Cramer-Rao lower bound on the variance of an estimator. We hope that we have clarified the reviewer's confusion on these very important points.
>
> **Assumption 2 cannot hold. Softmax policies cover all policies, including deterministic policies. Therefore, there is no a lower bound for all policies**
>
> We refer to Section 5.2 in (Lan, 2021) to show that the exact same assumption is made to obtain the $O(\epsilon^{-2})$ complexity even for single agent settings (please see fifth line in page 21 in Lan, 2021 and Remark 1 therein). Our second main result Theorem 3.3 does not require Assumption 2. Moreover, to prove Theorem 3.3 we also extend (Lan, 2021)’s result in the single agent case, which only obtains $O(\epsilon^{-4})$ without Assumption 2.
>
> **As a follow-up comment to (1), suppose that there is a unique deterministic optimal policy. Under Assumption 2, the agent can never find the optimal policy. This means that even when $K, T, N$ go to infinity, there should still be some constant error term on the right hand side of Theorem 3,2, because of the fact that the agent is trying to find a deterministic policy under Assumption 2. Why is it not the case in Theorem 3.2?**
>
> First, Theorem 3.2 assumes that Assumption 2 holds (we should also mention that we require Assumption 2 to hold in the iterates of the algorithm). In the example of the reviewer, Assumption 2 does not hold. Then, of course, the result of the theorem does not follow. Nevertheless, our paper covers the case mentioned by the reviewer in Theorem 3.3 with rate O(1/\epsilon^4).
>
> To see what goes wrong in the bound of Thm. 3.2 without Assumption 2, we note the following: We mention right after Assumption 2 that $\lambda_{\min}^{\theta}, \lambda_{\min}^\nu$ will be separated from $0$ only when Assumption 2 holds (their values are constant multiples of $\underline x, \underline y$). If not, these parameters will tend to $0$ and hence the second term in Thm. 3.2 will have these terms in the denominator, resulting in the non-convergence of this term.
>
> Can the reviewer please let us know if they follow our arguments or if they would require further explanations, regarding their doubts on the correctness of our analysis?

---

> > ### Author Response · Authors · 2021-11-26
> > **Discussion**
> >
> > Dear reviewer, could you please let us know if our explanation is clear to you? We would appreciate the chance to interact before the deadline of the discussion period, to address any other misunderstandings. Thank you.

---

> > ### Comment · Reviewer_jDaF · 2021-11-27
> > **Clarification on correctness**
> >
> > Thank the authors for their detailed feedback. I went back and carefully checked the SPMD method (in particular Theorem 6) of (Lan 2021) and also this paper. I am now convinced about the correctness of the result.
> >
> >
> > I want to add to the authors’ feedback that that the reason that the SPMD method in (Lan 2021) achieves $O(\epsilon^{-2})$ sample complexity is not completely because of the faster convergence of the bias. A main reason is that the critic error appears in the bound as mean-square $E[|\theta_t-\theta^*|^2]$ rather than mean $E[|\theta_t-\theta^*|]$ (the $\sigma^2$ in Theorem 6 of their result). This is because the author of (Lan 2021) uses Young’s inequality to trade-off the convergence rate of the policy and the convergence rate of the critic in their Eq. (4.8). In many existing literature, the critic error appears as mean error in the bound, such as Theorem 6.1 of (Agarwal et al., 2020), where the $\epsilon_{stat}$ appears under the square root. In that approach, it seems that it is not possible to establish $O(\epsilon^{-2})$ sample complexity without establishing geometric convergence of the actor first.
> >
> > Regarding my first paragraph in the review, none of the other papers except (Lan 2021) mentioned in the contributions paragraph of this work have $O(\epsilon^{-2})$ sample complexity. Please update the paper to avoid confusion.

---

> > > ### Author Response · Authors · 2021-11-27
> > > **Thanks for following up**
> > >
> > > We are grateful to the reviewer for the time and effort to carefully check the technical arguments and verify our explanation. According to the suggestion, we will highlight that Lan, 2021 is the only work with $O(\epsilon^{-2})$ complexity (to our knowledge).

---

### Official Review · Reviewer_JFQa · 2021-11-02

**Correctness:** 3
**Technical Novelty And Significance:** 3
**Empirical Novelty And Significance:** Not applicable
**Recommendation:** 6
**Confidence:** 3

**Main Review:**

Strengths: The theoretical RL community will find the results presented in this paper interesting. Assumption 1 about stationary state distribution is common in many previous works on finite-time convergence bounds for RL algorithms. Although Assumption 2 is less common than 1, I think it is reasonable, and one of the results in this paper does not rely on Assumption 2. It is good to see the order of sample complexity can be improved much based on these assumptions. The authors also explain the intuitions behind their proofs well in Section 3.2.

Weakness: Compared with the improvement on complexity bounds, the novelty of the algorithm design is relatively weak. The algorithm is designed based on the two-stage framework used in [Perolat et al., 2015], and NPG approach used in the second stage is standard because it is a single agent problem. There are multiple existing results that can be used to solve the equilibrium of a matrix game in the first stage, and the authors adopt FoRB algorithm [Malitsky & Tam, 2020]. If I understand correctly, the major challenge is that the theoretical results in [Malitsky & Tam, 2020] cannot be directly applied here.

There are also some minor presentation issues. The definition of "Policy-Eval" algorithm may cause confusion because it can take either 5 or 4 arguments, and the argument $\beta_n^\omega, \beta_n^\theta, \beta_n^\nu$ are only used to distinguish different functions of this algorithm. To improve readability, I suggest to separate "Policy-Eval" to three small procedures. Besides, the curve in the second subfigure of Figure 1 looks weird because the shaded area exceeds the lower bound 0 and upper bound 1 for probabilities. The implementation of this simulation needs to be double-checked.

**Summary Of The Paper:**

This paper studies the sample complexity of learning algorithms in two-player zero-sum tabular Markov games. The authors propose a two-stage algorithm that requires agents to agree on an etiquette, which means the agents have behave differently in two stages. Based on two assumptions on stationary state distribution and action exploration that are different from previous works, the authors show two finite-time convergence bounds, which improve much compared with existing results. The most challenging part of their proof is in bounding the approximation error in the first stage, which approximates the equilibrium of a matrix game. The authors also present some numerical results that can verify the performance of the proposed algorithm.

**Summary Of The Review:**

The theoretical analysis of this work is interesting and new, but the framework and the two major components of the algorithms have been studied in prior work. This is a theoretical work, so the empirical novelty and significance does not apply.

---

> ### Author Response · Authors · 2021-11-22
> **Response to Reviewer JFQa**
>
> We thank the reviewer for the constructive feedback.
>
> > *" There are multiple existing results that can be used to solve the equilibrium of a matrix game in the first stage, and the authors adopt FoRB algorithm [Malitsky & Tam, 2020]. If I understand correctly, the major challenge is that the theoretical results in [Malitsky & Tam, 2020] cannot be directly applied here."*
>
> We thank the reviewer for this comment, which prompts us to one more time highlight the novelties here. In particular, you can find these novelties explicitly highlighted in the text as "insights."
>
> - Our result relies on a careful analysis of the bias between different stages of the algorithm and an algorithmic design for making this possible. For example, notice that in Stage 2 of algorithm 1 we do not output a value function estimate as usually done in the policy evaluation step (Perolat 2015) but the best response policy. The value function is then estimated at each iteration of Stage 1.  This is important to achieve the fast rate as highlighted in Insight 4.
>
> - For solving the matrix game, we introduce an analysis for the stochastic primal-dual method that we use, which is different from the existing results in Malitsky, Tam 2020 or followup works such as Bohm et al., 2020.
>
> - To prove Theorem 3.3 we also extend (Lan, 2021)’s result in the single agent case, which only obtains $O(\epsilon^{-4})$ without Assumption 2 (see Appendix G.2.1).
>
> - We believe that our results are non-trivial as our complexity results significantly improve the existing ones for natural policy gradient. Note that in this case, only the $O(\epsilon^{-8})$ result from Wei et al., 2021 applies, which is improved by our $O(\epsilon^{-4})$ complexity under similar assumptions. Moreover, our $O(\epsilon^{-2})$ complexity matches the best-known result in the single agent setting.
>
> > *"To improve readability, I suggest to separate "Policy-Eval" to three small procedures"*
>
> We addressed these concerns on readability in our revision. Please see Alg. 2, 3, 4.
>
> > *"Besides, the curve in the second subfigure of Figure 1 looks weird because the shaded area exceeds the lower bound 0 and upper bound 1 for probabilities. The implementation of this simulation needs to be double-checked."*
>
> We are plotting the average probability with one standard deviation for the shaded area. The value obtained as mean + std can fall outside the domain of the random variable as it happens in this case.

---

> > ### Comment · Reviewer_JFQa · 2021-11-28
> > **Thanks for the response and the revision!**
> >
> > I want to thank the authors for the detailed response and the paper revision. I agree with the technical novelties the authors mentioned here, and are willing to vote for acceptance. I feel my current score is appropriate after reading other reviewer's comments.

---

### Official Review · Reviewer_TPgg · 2021-11-03

**Correctness:** 4
**Technical Novelty And Significance:** 3
**Empirical Novelty And Significance:** Not applicable
**Recommendation:** 6
**Confidence:** 2

**Main Review:**

This paper analyzes an important multi-agent RL problem and obtains interesting theoretical results. As claimed by the authors, the results are novel for policy gradient methods in this game setup. While the paper is overall technically dense, the authors manage to clearly convey the main idea through well articulated insights/intuitions. The assumptions and results are explained, which is helpful for understanding the context without referring too much to the technical details. I didn't check the proofs, but the theoretical results seem solid. I agree that this paper primarily focuses on theoretical contributions and that the numerical verification is enough.

One small comment: could the authors add some remarks regarding the feasibility/difficulty of extending the results to the case of function approximation? For example, it was mentioned in page 3 that a sample complexity of epsilon^-6 was obtained with function approximation, under access to unbiased samples of the value functions. How would the proposed algorithm perform if under the similar oracle assumptions?

**Summary Of The Paper:**

This paper considers algorithms based on natural actor-critic for solving two player zero-sum Markov games in the tabular case. In particular,  the authors focus on the analysis of the sample complexity for a two-stage algorithm that solves a matrix game and a single agent problem, alternatively and iteratively. By combining and refining recent results for policy gradient methods, this paper manages to match the best-known results for global convergence of policy gradient algorithms for single agent RL.

**Summary Of The Review:**

I think the paper is well written and the technical contributions are important and solid.

---

> ### Author Response · Authors · 2021-11-22
> **Response to Reviewer TPgg**
>
> We thank the reviewer for the constructive feedback.
>
> > *"Could the authors add some remarks regarding the feasibility/difficulty of extending the results to the case of function approximation? For example, it was mentioned in page 3 that a sample complexity of epsilon^-6 was obtained with function approximation, under access to unbiased samples of the value functions. How would the proposed algorithm perform if under the similar oracle assumptions?"*
>
> With unbiased samples of value functions, we can run standard SGD and if the variance of the estimator is bounded, one can get $O(\epsilon^{-2})$ by using standard SGD analysis and diminishing step sizes. With such an oracle, one would not need Assumptions 1 and 2 in this case and still obtain O(\epsilon^{-2}). We believe this shows the strength of such an oracle. Here, the important point is that we do not need the unbiased samples in the tabular case, whereas the current theory for NPG with function approximation, even with single agent case, needs unbiased samples of value functions. We refer to Agarwal et al., 2020 for this result.
>
> Finally, for the extension of our techniques to the function approximation case, there are a couple of directions. We believe it is straightforward to extend our results to the case when the critic uses linear function approximation (by using the techniques of Bhandari et al., 2018). On the other hand, analyzing a more complicated function approximation for critic and/or actor is still mostly open, to our knowledge, even for the single agent setting. The best complexities for single agent problems, to our knowledge, are from Lan, 2021 and this result only applies without function approximation.

---

### Author Response · Authors · 2021-11-22
**Main updates on the revision/common response**

We thank the reviewers for the constructive feedback which allowed us to improve the presentation of our paper significantly. We marked the changes in blue in the revised version. Main changes are:

- We removed $\lambda_{\min}^\theta,\lambda_{\min}^\nu,\lambda_{\min}^\omega$ and replaced them with their values based on the parameters in Assumptions 1, 2.

- We split Alg. 2 to three different algorithms to clarify their similarities and differences.

- We edited Assumptions 1, 2 to clarify that we need them to hold on the iterates of the algorithm.

- We added explicit convergence rates and complexities in Thm. 3.2, 3.3 to highlight the dependence of our bounds in terms of $\gamma, |S|, |A|, |B|, \underline \rho, \underline x, \underline y$.

- We clarified in Table 1 that $O(\epsilon^{-4})$ result of Wei et al., 2021 does not apply to NPG algorithm, which is the focus of our paper.

Moreover, we clarified the main concern of Reviewer jDaF on the correctness of our result, by explaining relevant results in the previous work Lan, 2021 and why the results therein are consistent with ours.

---

> ### Author Response · Authors · 2021-11-27
> **Further discussion**
>
> We thank all the reviewers for their constructive criticism which helped us improve several aspects of our paper. As Reviewer jDaF acknowledged and updated their score, we addressed their main concern regarding the correctness of our result.
>
> Therefore now all the reviewers seem to agree on the significance and correctness of our results.
>
> On this front, could the reviewers please let us know if there is anything else we can do to improve our paper, before the discussion deadline on monday?
>
> Thank you.

---

### Decision · Program_Chairs · 2022-01-20

**Decision:**

Reject

**Comment:**

This paper proposed algorithms (based on natural actor-critic methods) to solve two-player zero-sum Markov games. The authors established theoretical support for the convergence properties--and hence sample complexity---of the proposed methods. The authors claimed, based on their theoretical results, that the proposed methods are sample-efficient.

As the reviewers pointed out, the original submission focused on the dependency on epsilon without explicit dependency on other important parameters like S, A, B, etc. The revised version has made explicit the dependencies on all these problem parameters, which I appreciated. However, the sample complexity presented in the new version scales as either S^3 max{A,B} or S^4 * \max{A,B}^6 on the sizes of state space and action spaces, which are all huge. What is more, the sample complexities also rely on additional parameters like rho, x, y, which could all depend on S,A,B, etc. As a result, the resulting sample complexity bounds do not seem to imply sample efficiency. In addition, Assumptions 1 and 2 are somewhat unnatural to make.